# FOXQ1 recruits the MLL complex to activate transcription of EMT and promote breast cancer metastasis

Allison V. Mitchell[1], Ling Wu [1], C. James Block[1], Mu Zhang[1], Justin Hackett [1], Douglas B. Craig[1], Wei Chen [1], Yongzhong Zhao[2], Bin Zhang[2], Yongjun Dang [3], Xiaohong Zhang[1], Shengping Zhang[4], Chuangui Wang [4], Heather Gibson[1], Lori A. Pile[5], Benjamin Kidder [1], Larry Matherly[1], Zhe Yang [6], Yali Dou [7,8] & Guojun Wu [1] ✉

Aberrant expression of the Forkhead box transcription factor, FOXQ1, is a prevalent mechanism of epithelial-mesenchymal transition (EMT) and metastasis in multiple carcinoma types. However, it remains unknown how FOXQ1 regulates gene expression. Here, we report that FOXQ1 initiates EMT by recruiting the MLL/KMT2 histone methyltransferase complex as a transcriptional coactivator. We first establish that FOXQ1 promoter recognition precedes MLL complex assembly and histone-3 lysine-4 trimethylation within the promoter regions of critical genes in the EMT program. Mechanistically, we identify that the Forkhead box in FOXQ1 functions as a transactivation domain directly binding the MLL core complex subunit RbBP5 without interrupting FOXQ1 DNA binding activity. Moreover, genetic disruption of the FOXQ1-RbBP5 interaction or pharmacologic targeting of KMT2/MLL recruitment inhibits FOXQ1-dependent gene expression, EMT, and in vivo tumor progression. Our study suggests that targeting the FOXQ1-MLL epigenetic axis could be a promising strategy to combat triple-negative breast cancer metastatic progression.

The epithelial-to-mesenchymal transition (EMT) is a fundamental process during organ development and tissue repair[1]. Aberrant activation of EMT has been observed as a critical step in cancer progression, facilitating distant metastasis and drug resistance[2]. At the pinnacle of EMT regulation is the induction of a core set of EMT transcription factors (TFs) that drive cell plasticity[3].

FOXQ1, a Forkhead box TF, plays a canonical role during development and hair follicle differentiation[4,5]. Aberrant expression of

FOXQ1 has been observed to cause increased metastatic competence and drug resistance through triggering EMT in carcinoma cells[6–10]. In breast cancer, FOXQ1 is linked to the aggressive triple-negative subtype (TNBC)[9,10]. Epidemiologically, FOXQ1 expression is predictive of a worse prognosis in six solid tumor types[11].

The mechanisms that regulate FOXQ1 expression are largely unknown. Both tumor-associated fibroblasts and macrophages have been found to increase the expression of FOXQ1 in cancer cells[12,13].

[1]Barbara Ann Karmanos Cancer Institute, Department of Oncology, Wayne State University School of Medicine, 4100 John R, Detroit, MI 48201, USA. [2]Department of Genetics and Genomic Sciences, Icahn Institute of Genomics and Multiscale Biology, Icahn Mount Sinai School of Medicine, New York, NY 10029, USA. [3]Key Laboratory of Metabolism and Molecular Medicine, the Ministry of Education, Department of Biochemistry and Molecular Biology, School of Basic Medical Sciences, Fudan University, Shanghai 200032, China. [4]Institute of Translational Medicine, Shanghai General Hospital, Shanghai Jiao Tong University School of Medicine, No. 650 Xinsongjiang Road, Songjiang District, Shanghai 201620, China. [5]The Department of Biological Sciences, Wayne State University, Detroit, MI 48202, USA. [6]Department of Biochemistry, Microbiology, and Immunology, Wayne State University School of Medicine, Detroit, MI 48201, USA. [7]Department of Pathology, University of Michigan Medical School, Ann Arbor, MI 48109, USA. [8]Department of Biological Chemistry, University of Michigan Medical School, Ann Arbor, MI 48109, USA. ✉e-mail: wugu@karmanos.org

FOXQ1 was found to be upregulated downstream of the transforming growth factor-beta (TGF-β) and canonical Wnt signaling pathways[7,14,15]. Numerous miRNAs have also been found to downregulate FOXQ1 and suppress EMT and metastasis[16–21]. Disruption of FOXQ1 expression has been shown to decrease metastasis, acquisition of stemness, and enhanced chemotherapeutic sensitivity in vivo[9,10]. These data suggest that disruption of the FOXQ1 transcription program could provide therapeutic benefits. In contrast with the previous findings[8,10,11], FOXQ1 was recently reported to suppress metastasis in melanoma models, which derive from the neural crest lineage[22]. These cell type-dependent observations suggest a critical need to understand the epigenetic contexts governing FOXQ1 function in cancer.

The histone methyltransferase 2 or mixed-lineage leukemia (KMT2/MLL) complex is a histone-3 lysine-4(H3K4) histone methyltransferase (HMT) complex[23–26]. KMT2/MLL HMTs share a standard composition of four core subunits (WDR5, ASH2L, RbBP5, and DPY30) bound to one of six catalytic subunits: KMT2A/MLL1, KMT2B/MLL2, KMT2C/MLL3, KMT2D/MLL4, KMT2F/SET1A, KMT2G/SET1B. Unlike other HMTs, KMT2/MLL HMTs have low intrinsic activity and require binding of the four core subunits for maximal catalytic activity[24,27–29]. H3K4 methylation is tightly regulated and can be found in mono-, di-, or tri-methylated forms (me1, me2, me3). H3K4me1 marks active and poised enhancers, H3K4me2 is found throughout the gene bodies of actively expressed genes, and H3K4me3 marks promoters near transcription start sites (TSS) and are correlated with active gene transcription[30–32]. The KMT2/MLL enzymes and the core subunits have been well recognized as prominent regulators of H3K4 methylation and the activation of gene transcription throughout development and in cancer[23]. In mammals, KMT2C/MLL3 and KMT2D/MLL4 primarily regulate H3K4me1 within putative enhancer regions[33]. KMT2F/SET1A and KMT2G/SET1B are responsible for the bulk of H3K4me2/3, whereas KMT2A/MLL1 and KMT2B/MLL2 are responsible for gene-specific H3K4me3 within target promoters[33].

In this work, we sought to determine the mechanism by which FOXQ1 regulates the activation of an EMT transcription program. By tandem affinity purification and mass spectrometry proteomics (TAP-MS), we found that FOXQ1 physically associates with KMT2A/MLL1-containing HMT complex through directly interacting with RbBP5. Direct interaction of FOXQ1 and RbBP5 was further validated in TNBC cell lines and various human tumor samples, including TNBC. We demonstrated that FOXQ1 acts as a sequence-specific TF that recruits the MLL complex to promoter regulatory regions of EMT target genes in human mammary epithelial cells (HMLE) and TNBC cell lines. We found that RbBP5 and MLL1, and consequently H3K4me3, are necessary for the transcriptional activation of FOXQ1 target genes. Disruption of the FOXQ1-MLL1 axis, with pharmacologic or genetic approaches, resulted in a perturbation of the EMT and oncogenic phenotypes. Targeting MLL1 in TNBC cells dramatically reduced metastasis in vivo.

## Results

### FOXQ1 interacts with the KMT2/MLL family core complex
We sought to identify FOXQ1-interacting proteins by TAP-MS from HEK293 cells alongside LacZ vector control[34]. We identified 98 FOXQ1-complexing proteins, including subunits of the KMT2/MLL methyltransferase core complex (RbBP5, WDR5, and ASH2L) responsible for the regulation of H3K4me3 at regulatory elements of actively transcribed genes[23,25] (Fig. 1a, and Supplementary data 1). We confirmed that FOXQ1 binds the MLL core complex (RbBP5, WDR5, and ASH2L) in human mammary epithelial cells with ectopic FOXQ1 expression (HMLE/FOXQ1) (Fig. 1b). These cells overexpress FOXQ1 to a similar level to what is observed in TNBC and is a validated model of FOXQ1-driven EMT[8,10] (Supplementary Fig. 1a, b). We performed pairwise binding assays to identify the MLL complex component(s) directly bound by FOXQ1. Recombinant RbBP5, ASH2L, and WDR5 were individually incubated with resin conjugated to GST-FOXQ1 or GST-alone

(Supplementary Fig. 1c, d). Coprecipitated proteins were detected by western blot. We determined that only purified RbBP5 bound to recombinant GST-FOXQ1, establishing RbBP5 is responsible for directly binding FOXQ1 to the MLL complex (Fig. 1c). We further confirmed that endogenous FOXQ1 and RbBP5 interact in multiple TNBC cells (MDA-MB-231, SUM1315, MDA-MB-436) utilizing an anti-FOXQ1 antibody developed by the lab (Fig. 1d–f, Supplementary Fig. 1e). The specificity of the FOXQ1 antibody was further validated by RbBP5-IP in MDA-MB231 cells with FOXQ1 knockdown (Supplementary Fig. 1f). In contrast, the EMT TFs, FOXC2, and SNAIL1 were unable to coprecipitate the members of the MLL complex, suggesting that the FOXQ1-MLL complex association harbors biochemical and functional specificity (Supplementary Fig. 1g, h). We also confirmed endogenous interaction between FOXQ1 and RbBP5 in various tumor types, including TNBC (Fig. 1g). Our mass spectrometry proteomics failed to identify any KMT2/MLL enzymatic subunits in the FOXQ1 complex, likely due to the large protein sizes of these proteins. Therefore, we performed endogenous immunoprecipitation (IP) of each KMT2/MLL enzymatic subunit in HMLE/FOXQ1 cells and tested their binding ability to FOXQ1. Results from these co-IP experiments identified that KMT2A/MLL1 is the FOXQ1 binding subunit (Fig. 1h–m).

### FOXQ1 and the MLL core complex co-regulate gene targets with critical EMT functions
To assess the interplay between FOXQ1 and the MLL complex in regulating the EMT transcription program, we performed chromatin immunoprecipitation sequencing (ChIP-seq) for FOXQ1, RbBP5, and H3K4me3 in HMLE/FOXQ1 cells. Several FOX family TFs are pioneer factors and predominantly bind within distal enhancers[35–37]. In HMLE cells, we observed FOXQ1 bound to proximal promoter regions (<10 kb upstream; 8.1% of 13,513 total peaks, 4.6% above the genome average of 3.5%) and distal intergenic regions (58.5% of the total, 8.5% above the genome average of 50%) (Fig. 2a). RbBP5 binding peaks also fell within proximal promoter regions (10.8% of 25,866 total peaks, 7.3% above the genome average) and in distal intergenic regions (51.9% of the total, <2% above the genome average) (Fig. 2a). A substantial subset (32%) of FOXQ1 peaks overlapped with RbBP5 peaks (4294/13,513), and 15% of FOXQ1-bound regions were marked by H3K4me3 (2073/13,513 peaks) (Fig. 2b). Moreover, the FOXQ1 and RbBP5 binding profiles correlated with transcription start sites (TSS) containing high levels of H3K4me3, supporting a role for MLL in the activation of FOXQ1 target genes[38] (Fig. 2c, Supplementary Fig. 2a). Analysis of the enriched DNA recognition motifs within FOXQ1-RbBP5 overlapping regions confirmed the presence of the Forkhead motif (MEME[39], $E$-value = 2.3e−300) (Supplementary Fig. 2b).

To examine the transcriptional consequence of FOXQ1-RbBP5 chromatin binding events, we analyzed differential expression between HMLE/FOXQ1 and HMLE/LacZ cells by RNA-seq (Supplementary data 2). We identified 2499 upregulated genes downstream of FOXQ1, enriched for EMT functions such as hypoxia signaling, angiogenesis, and extracellular matrix organization (Supplementary Fig. 2c). Of the 2499 upregulated transcripts in the FOXQ1-driven EMT model (HMLE/FOXQ1), 791 genes (~32%) displayed FOXQ1 binding within the associated promoter (Fig. 2d). Conversely, only 7.3% of differentially downregulated genes were related to FOXQ1 promoter binding (182/2500 genes), suggesting that FOXQ1 plays a much more prominent role as a transcriptional activator of EMT (Fig. 2d and Supplementary Fig. 2d). Strikingly, 92% of FOXQ1-bound transcriptionally active promoters (622/791 genes) were also occupied by RbBP5 (<10 kb upstream TSS), and 73% were marked by H3K4me3 (Fig. 2e).

The 622 FOXQ1-RbBP5 co-activated genes were significantly enriched for pathways critical to the EMT program, including Wnt, PDGF, TGF-β, and cadherin signaling (PANTHER[40]) (Fig. 2f). FOXQ1-RbBP5 targets were also enriched for oncogenic functions such as focal adhesion regulation and pro-survival signaling, and several metabolic

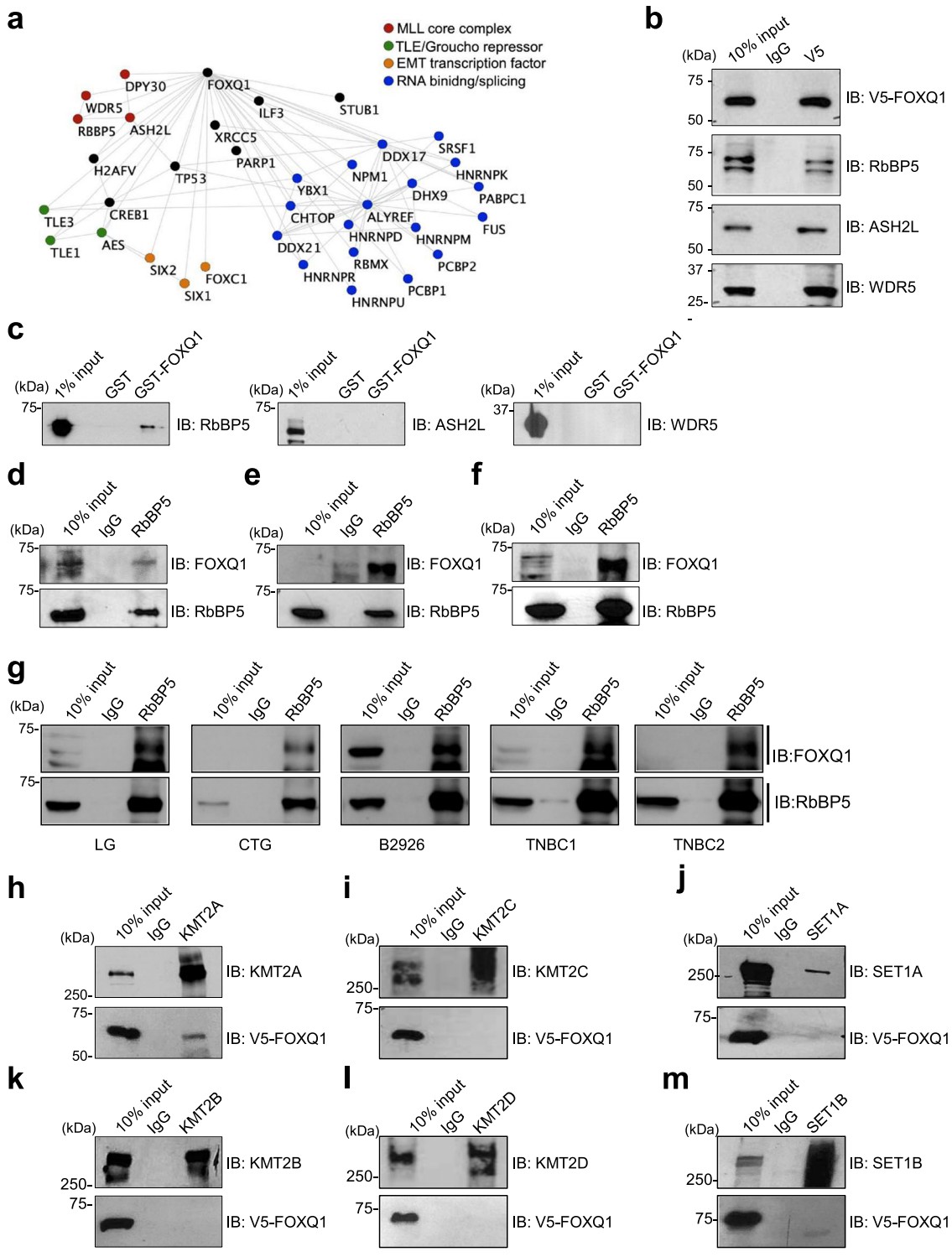

processes such as lipid metabolism, insulin signaling, and inflammatory processes (KEGG[41], GO[42]) (Fig. 2f). Furthermore, four EMT TFs (FOXC2, TWIST1, ZEB1, SIX2) were identified as direct FOXQ1-RbBP5 downstream targets (Fig. 2g). The co-localization of FOXQ1 and RbBP5 within EMT promoters was confirmed by ChIP-qPCR (Supplementary Fig. 2e). The clinical significance of the FOXQ1-RbBP5 epi-regulome was examined by correlating the expression of the 622 genes across breast cancer samples (TCGA). A subset of 109 genes exhibited a significant Spearman correlation ($R > 0.2$) with FOXQ1 across breast cancer samples (Supplementary Fig. 2f). Further analysis demonstrated these genes were enriched for EMT- and stemness-associated

pathways, including Insulin resistance, Focal adhesion, PDGFR, and DDR2 signaling (Supplementary Fig. 2g). Moreover, 7/109 exhibited significant multivariate association with worse overall survival, including WNT5B and LAMA4 (Supplementary Fig. 2h). Interestingly, LAMA4 and FOXQ1 have been identified as molecular determinants of metastatic tumor re-initiation in breast cancer[43].

**MLL core complex is required within gene promoters for FOXQ1 target gene expression**

The involvement of MLL1 in the FOXQ1 complex, the significant co-occupation (62%) of FOXQ1, RbBP5, and H3K4me3 in the

**Fig. 1 | FOXQ1 interacts with the KMT2A/MLL1 complex in EMT. a** Network analysis (Cytoscape v 3.7.1) of the 100 most significant proteins bound to FOXQ1 identified through tandem affinity purification (TAP) and mass spectrometry proteomics. Fischer's Exact T-test was conducted on candidate proteins' normalized, mapped peptide counts. We highlighted several complexes, including four subunits that comprise the KMT2/MLL core complex (RbBP5, ASH2L, WDR5, and DPY30). FOXQ1 and LACZ TAP were performed in duplicate. **b** Co-immunoprecipitation of FOXQ1 from stable HMLE/FOXQ1 cells (with antibody against a C-terminal V5-epitope tag) validates protein interaction with the endogenous MLL core complex proteins (RbBP5, ASH2L, WDR5) within the context of EMT induction. **c** The binary interacting partner of FOXQ1 was assessed by GST pull-down. Purified GST-FOXQ1, or GST alone, was incubated at equimolar concentration with individual recombinant MLL core complex proteins (RbBP5, ASH2L, WDR5). Western blots were probed for with an antibody that recognizes the native protein as indicated. **d**–**f** Endogenous Co-IP of RbBP5, or IgG control, was performed in triple-negative breast cancer cell lines. IPs were probed with an antibody recognizing endogenous FOXQ1 to detect a protein interaction. **d** MDA-MB-231, **e** SUM1315, **f** MDA-MB-436. **g** Protein lysates from patient tumor samples of various types were used to test the endogenous interaction of FOXQ1 and RbBP5. Tumor lysates were subject to IP anti-RbBP5 antibody or IgG control. Samples were analyzed by western blot and probed with anti-FOXQ1 antibody. LG is a lung tumor tissue. CTG is an ovarian tumor tissue. B2926, TNBC1, and TNBC2 are all triple-negative breast cancer tumors. **h**–**m** HMLE/FOXQ1 cell lysate was utilized for immunoprecipitation toward identifying the KMT2/MLL enzymatic subunit(s) associated with FOXQ1. HMLE/FOXQ1 lysate was subject to IP with antibody against each of the six KMT2/MLL family members, alongside an IgG control, and assessed for the presence of FOXQ1 binding by western blots probed with anti-V5 antibody. An interaction was detected between FOXQ1 and KMT2A/MLL1 (**h**) and was absent in IP samples for other KMT2/MLL members (**i**–**m**). For panels **b**–**m**, representative images are presented (*n* = 3). Unprocessed western blot images are provided in Source data 2.

promoter region of targeted genes, and the enrichment of co-activated genes in the EMT program promoted us to investigate how FOXQ1 recruits the MLL complex to the promoter region in regulating the EMT program, which could potentially contribute to tumor progression. We employed a panel of FOXQ1-RbBP5 direct targets with annotated EMT functions as makers for this mechanistic study (Fig. 3a). Two FOXQ1 target genes (*CTSB, IL17RA*), which lack RbBP5 promoter binding ("RbBP5 independent genes"), were included as negative controls. We first assessed the impact of RbBP5 silencing on the expression of FOXQ1 target genes in HMLE/FOXQ1 cells (Supplementary Fig. 3a, b). RbBP5 silencing caused a decrease in the expression of gene targets bound by FOXQ1 and RbBP5 in the associated promoter. In contrast, the mRNA levels of FOXQ1 and negative control genes (*CTSB, IL17RA*) were largely unaffected (Fig. 3b and Supplementary Fig. 3a).

We sought to determine if MLL1 recruitment, containing H3K4 methyltransferase activity, was required for transcriptional activation by FOXQ1. HMLE/FOXQ1 cells were subject to shRNA knockdown of MLL1 or treated with OICR-9429, which disrupts the protein interaction between WDR5 and MLL and prevents H3K4me3[44,45] (Supplementary Fig. 3c, d). Upon MLL1 knockdown, we observed a decrease in expression of the FOXQ1-RbBP5 target genes (Fig. 3c). We also observed a dose-dependent reduction in the transcript abundance of FOXQ1-RbBP5 co-activated targets upon OICR-9429 treatment (Fig. 3d). RbBP5-independent genes, *CTSB* and *IL17RA*, displayed no significant change in expression with OICR-9429 treatment or MLL1 knockdown compared to the respective controls (Fig. 3c, d).

We confirmed our findings in the MDA-MB-231 and MDA-MB-468 TNBC cell lines. Knockdown of FOXQ1 in MDA-MB-231 or MDA-MB-468 cells decreased the expression of FOXQ1 targets identified by ChIP-seq, confirming their status as bonafide FOXQ1-target genes (Supplementary Fig. 3e–j). Further, the knockdown of RbBP5 in MDA-MB-231 or MDA-MB-468 cells caused a decrease in FOXQ1-RbBP5 target genes without a significant change in the expression of the RbBP5-independent targets (CTSB, IL17RA) (Supplementary Fig. 3k–p). We observed a similar reduction in FOXQ1-RbBP5 EMT target gene expression in MDA-MB-231 cells upon MLL1 KD or treatment with OICR-9429 (Supplementary Fig. 3q–s). In line with this, we showed that MLL1 knockdown decreased H3K4me3 levels within the FOXQ1-targeted promoters (Supplementary Fig. 3t). These results support that both RbBP5 and MLL1 recruitment are essential for the transcriptional activation activity of FOXQ1 at EMT gene loci.

### FOXQ1 recruits the MLL complex to EMT target promoters
We further examined the interdependence of FOXQ1 and MLL complex chromatin localization at selected loci in HMLE/FOXQ1 cells. We found that FOXQ1 promoter recognition was not impacted by RbBP5 depletion (Fig. 3e). However, RbBP5 knockdown caused a significant decrease of H3K4me3 at the promoters of FOXQ1-RbBP5 gene targets, confirming the critical role of the MLL complex in H3K4me3 deposition at the target loci (Fig. 3f). Conversely, we found that FOXQ1 functions upstream of both RbBP5 and MLL recruitment since FOXQ1 knockdown decreased RbBP5 promoter localization and H3K4me3 abundance within the panel of EMT promoters without affecting RbBP5 expression (Fig. 3g, h, and Supplementary Fig. 3u, v). Furthermore, RbBP5 was not observed to bind the promoters of FOXQ1-targets in the epithelial, control HMLE/LacZ cells (Supplementary Fig. 3w). These observations were confirmed in MDA-MB-231 cells, in which FOXQ1 depletion caused a decrease in RbBP5 binding within EMT promoters (Supplementary Fig. 3x). These results demonstrate that FOXQ1 recruits the MLL complex to FOXQ1 target gene promoters as a transcriptional coactivator.

### The N-terminal region of the Forkhead domain of FOXQ1 mediates the interaction with RbBP5
To map the domain of FOXQ1 responsible for RbBP5 binding, we cloned a series of FOXQ1 sub-domains and interrogated endogenous RbBP5 binding in HEK293 cells. The central domain of FOXQ1 contains the Forkhead box DNA binding domain (FHD)[46–48]. The amino and carboxy-terminal of FOXQ1 are predominately unstructured[4,47]. We first interrogated the role of these three major regions of FOXQ1 (Fig. 4a). Both the FOXQ1 N-terminal (1-142) and FOXQ1-FHD (105-227) fragments, but not the C-terminal (204-403) fragments, retained RbBP5 binding. (Fig. 4b). Subsequently, only constructs containing the N-terminal region of the FOXQ1 FHD (105-142) retained the ability to interact with RbBP5 (Fig. 4c). Truncation of the FOXQ1 N-terminal region to residues 75-142 retained the ability to bind RBBP5, indicating that this region is minimally required to bind FOXQ1 to the MLL core complex (Fig. 4d). The nuclear localization of these FOXQ1 sub-cloned fragments remained unchanged (Supplementary Fig. 4a).

### The FOXQ1 H1-S1 residues are critical for binding to RbBP5
The FOX superfamily shares a conserved FHD with three α-helices, three β-sheets, and two wing regions[49]. The identified RbBP5 binding region (residues 75-142 of FOXQ1) resides in the FOXQ1 FHD containing helix-1 and sheet-1 (H1-S1), which is highly conserved across species (Fig. 4e). However, compared to FHD sequences across the human FOX family, several residues of human FOXQ1 H1-S1 exhibit sequence divergence[49] (Supplementary Fig. 4b). We hypothesized the amino acids within the FOXQ1 H1-S1 could mediate the interaction with RbBP5 as an evolutionarily conserved mechanism. We performed site-directed mutagenesis within the H1-S1 FHD region and included residues outside the H1-S1 (K112E, R164E, R168E) as controls (Fig. 4f). These mutations did not alter FOXQ1 nuclear localization (Supplementary

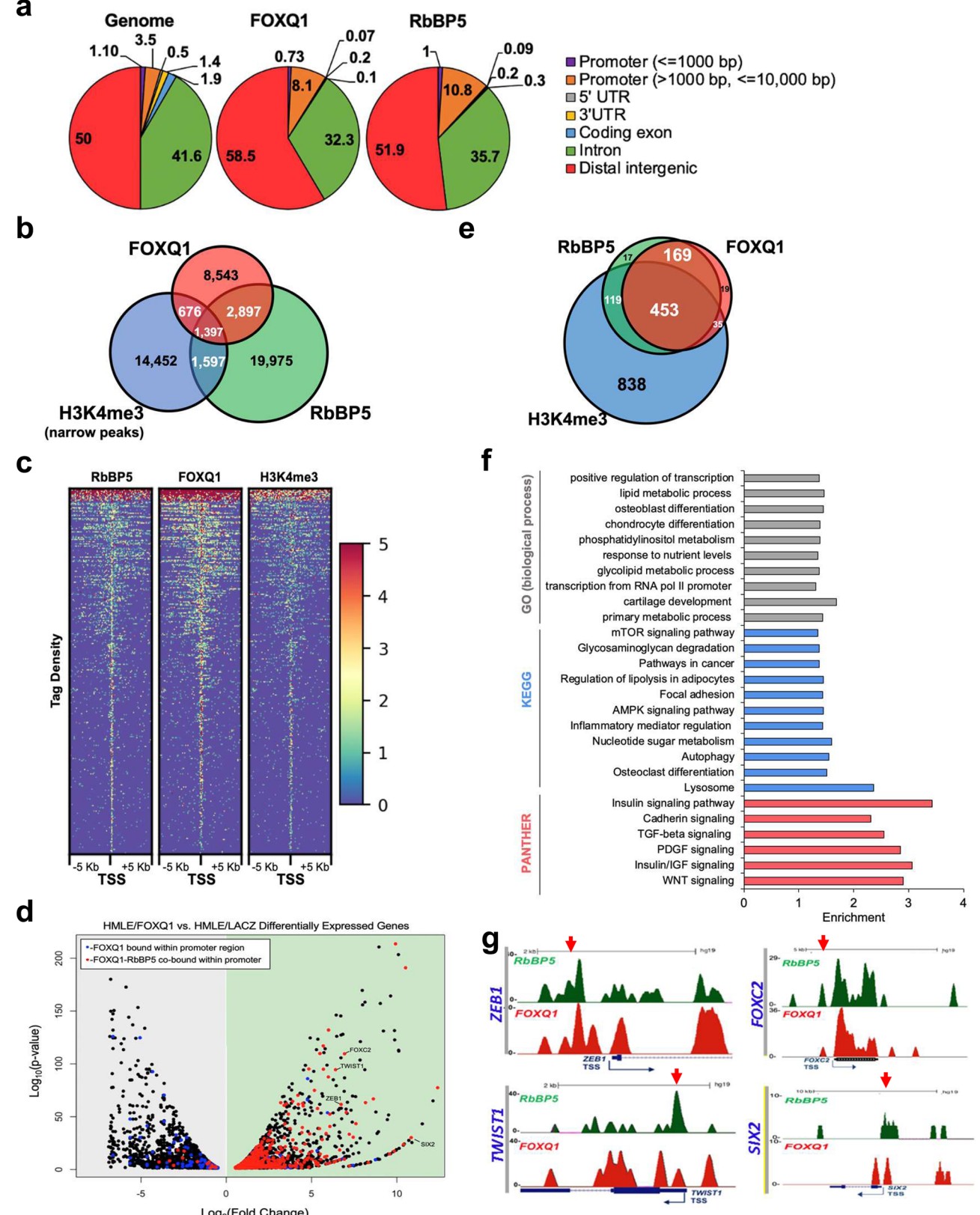

Fig. 4c). Mutation of helix-1 residues from nonpolar to polar (A126S, A129S, I132S) or amino acids in sheet-1 to a homologous residue (A136P, G137E, G138E) diminished the ability of FOXQ1 to bind RbBP5 (Fig. 4g and Supplementary Fig. 4d). Mutation of residues A129S and I132S resulted in the most dramatic decrease in RbBP5 binding (Fig. 4g and Supplementary Fig. 4d). Altogether, these data demonstrate that

the FOXQ1 FHD houses the RbBP5 binding domain. Disruption of this region perturbs the protein interaction.

## RbBP5 interacts with FOXQ1 through the hinge region

The N-terminus of RbBP5 contains a β-propeller domain consisting of WD40 repeats (1–338). The C-terminus of RbBP5 is unstructured and

**Fig. 2 | FOXQ1 and RbBP5 regulate a common set of EMT genes and display global chromatin co-localization within the EMT program. a** The cis-regulatory element annotation (CEAS) was utilized to identify the genomic regions occupied by FOXQ1 and RbBP5 binding peaks in HMLE/FOXQ1 cells. The genomic distribution of each factor is shown alongside whole-genome distribution (background control values). **b** Venn diagram of the overlapping genomic occupancy of FOXQ1, RbBP5, and H3K4me3 modified regions from HMLE/FOXQ1 ChIP-seq data. **c** Heatmap of the binding peak signal intensity centered around the TSS (+/−5 kb) within the FOXQ1-RbBP5 co-bound regions. Binding is ranked from strongest (red) to weakest intensity (blue). **d** Volcano plot of the differentially expressed genes between HMLE/FOXQ1 and HMLE/LACZ cells identified by EdgeR. Axes show $\log_2$ fold-change versus $\log_{10}$ p-value. We identified FOXQ1 and RbBP5 promoter localization by ChIP-seq. Red dots indicate differentially expressed genes that displayed

FOXQ1-RbBP5 co-localization within the promoter (TSS +/− 5 kb). Blue dots highlight gene promoters bound by FOXQ1 alone ("RbBP5-independent"). **e** Venn diagram of the overlap of ChIP-seq binding peaks for FOXQ1, RbBP5, and H3K4me3 within the promoter regions of differentially upregulated transcripts. **f** Enrichment analysis was performed on the 622 FOXQ1-RbBP5 upregulated genes using Enrichr. Annotation terms and gene sets used for comparison were derived from the indicated analytical tools. Input gene sets were compared to the annotated gene sets by Fisher's exact t-test with the Benjamini-Hochberg method for correction for multiple hypotheses testing. Enrichment was calculated as the $\log_{10}$ adjusted p-value and plotted as a bar graph. **g** ChIP-seq binding peaks for FOXQ1 and RbBP5 within the TSS of several EMT transcription factors (UCSC genome browser). Red arrows indicated genomic localization of the ChIP-PCR amplicon.

contains a central hinge region with binding motifs for activating MLL (AS), ASH2L binding (ABM), and WDR5 binding (WBM)[50,51]. We generated RbBP5 deletion constructs based on these features and co-expressed them individually with FLAG-FOXQ1 in HEK293 cells (Supplementary Fig. 4e). The RbBP5 C-terminal region (330–538) was able to mediate an interaction with FOXQ1. Truncation of the RbBP5 C-terminus to amino acids 360-538 disrupted FOXQ1 binding (Supplementary Fig. 4f). Therefore, RbBP5 depends on an intact hinge region to bind FOXQ1.

## Disrupting the FOXQ1-RbBP5 protein interaction attenuates the EMT program

To characterize how disruption of RbBP5 recruitment impacted FOXQ1 transcription, we performed RNA-seq of HMLE cells with stable expression of RbBP5-binding defective mutant FOXQ1-I132S (Supplementary Fig. 5a, and Supplementary data 3). The principal component analysis demonstrated that HMLE/FOXQ1-I132S cells exhibited an expression profile similar to HMLE/LacZ cells over HMLE/FOXQ1 wild-type (WT) (Supplementary Fig. 5b). HMLE/FOXQ1-I132S mutant cells differentially expressed 1,736 genes (911 upregulated, 825 downregulated) relative to HMLE/LacZ. Only a fraction (~12%, 350 genes) of differentially expressed genes were shared between HMLE/FOXQ1-WT and FOXQ1-I132S mutant cells relative to HMLE/LacZ (Fig. 5a and Supplementary Fig. 5c). Genes upregulated in HMLE/FOXQ1-I132S mutant cells were functionally enriched for epithelial cell functions, DNA damage response, and various metabolic processes (Supplementary Fig. 5d). Conversely, the HMLE/FOXQ1-I132S downregulated gene set displayed enrichment of functions associated with EMT such as cell migration, angiogenesis, and TGF-β signaling (Fig. 5b), demonstrating that the FOXQ1-I132S mutant has a loss of function for activation of an EMT transcription program.

We proposed that the above dysfunction was caused by an inability of FOXQ1 to recruit the MLL core complex to target EMT promoters. To examine this more closely, we compared the expression of the previously identified FOXQ1-RbBP5 target genes between HMLE/FOXQ1-I132S mutant and -WT cells. We found that 521 of the 622 FOXQ1-RbBP5 target genes were differentially expressed in the HMLE/FOXQ1-I132S mutant model relative to HMLE/LacZ. Of the differentially expressed FOXQ1-RbBP5 gene targets, ~93% (484/521 genes) displayed lower expression in HMLE/FOXQ1-I132S mutant cells than in HMLE/FOXQ1-WT cells (Fig. 5c). Furthermore, ~50% of the FOXQ1-RbBP5 gene targets were downregulated in HMLE/FOXQ1-I132S mutant cells relative to the epithelial HMLE/LacZ cells, suggesting the loss-of-function FOXQ1 mutant may have a dominant-negative effect on transcriptional activation of FOXQ1 targets.

Consistent with the protein interaction results, HMLE cells expressing RbBP5-binding defective FOXQ1 mutants (A129S or I132S) had decreased transcription of genes dependent upon the recruitment of RbBP5 (Fig. 5d), with sustained gene expression levels of RbBP5-independent targets (Fig. 5e). In addition, neither FOXQ1 mutant displayed an alteration in promoter localization by ChIP-qPCR,

demonstrating that the point mutations do not disrupt FOXQ1 DNA-binding (Fig. 5f). We also infected MDA-MB-231/shFOXQ1 cells with lentivirus expressing FOXQ1-I132S, FOXQ1-A129S, or FOXQ1-WT and assessed their ability to restore the regulation of FOXQ1 target genes. Neither FOXQ1-A129S nor FOXQ1-I132S mutant expression restored FOXQ1-RbBP5 target gene expression to wild-type levels in TNBC cells (Supplementary Fig. 5e). Altogether, these results demonstrate that disruption of the FOXQ1 FHD transactivation activity through loss-of-functional RbBP5 binding obstructs the ability of FOXQ1 to act as an EMT TF.

## The FOXQ1-MLL axis is critical for EMT and in vivo tumor progression

To further examine the biological requirements of the MLL core complex for FOXQ1-induced EMT, we performed several in vitro assays. Knockdown of RbBP5 in HMLE/FOXQ1 cells led to a significant morphological change and a decrease in cell migration and invasion (Fig. 6a, b, and Supplementary Fig. 6a, b) with no significant effect on cell proliferation relative to nontarget (NT) control (Supplementary Fig. 6c). There was no observed effect of RbBP5 knockdown on migration or invasion in the HMLE/LacZ control cell line (data not shown). Further, knockdown of RbBP5 resulted in a decrease in mesenchymal marker expression and increased epithelial claudin-1 levels (Fig. 6c). There was a decrease in the mammosphere formation and a reduction in the CD44$^+$/CD24$^-$ stem-like population in HMLE/FOXQ1 shRbBP5 models (Fig. 6d, e, and Supplementary Fig. 6d). Similarly, OICR-9429 treated HMLE/FOXQ1 cells exhibited a prominent epithelial-like morphology (Supplementary Fig. 6e) and a decrease in migration and invasion (Fig. 6f, g and Supplementary Fig. 6f), without change in cell proliferation (Supplementary Fig. 6g). HMLE/FOXQ1 OICR-9429 treatment decreased the expression of mesenchymal proteins and increased the expression of claudin-1 in a dose-dependent manner (Fig. 6h). OICR-9429 treatment also caused a reduction in stem-like features, including mammosphere formation and the CD44$^+$/CD24$^-$ population (Fig. 6i, j, and Supplementary Fig. 6h). We observed similar effects in MDA-MB-231 cells, in which knockdown of FOXQ1 or RbBP5 significantly decreased mammosphere formation (Supplementary Fig. 6i–l).

Next, we compared the effects of WT and mutant (A129S or I132S) FOXQ1 on EMT and stemness phenotypes. HMLE cells expressing either mutant FOXQ1(A129S or I132S) did not undergo EMT, as shown by inefficient induction of vimentin and N-cadherin and retention of E-cadherin (Fig. 7a). Both mutant cells maintained an epithelial-like morphology, with intact E-cadherin and β-catenin membrane localization, compared to WT counterparts (Supplementary Fig. 7a). Further, HMLE cells expressing either mutant FOXQ1 had a reduced ability to migrate and invade than the WT group (Fig. 7b, c, and Supplementary Fig. 7b). However, there was no observed effect on cell proliferation between the groups (Supplementary Fig. 7c). HMLE/FOXQ1 mutant cells lacked stem-like characteristics shown by their deficient mammosphere formation capability and decreased CD44$^+$/CD24$^-$

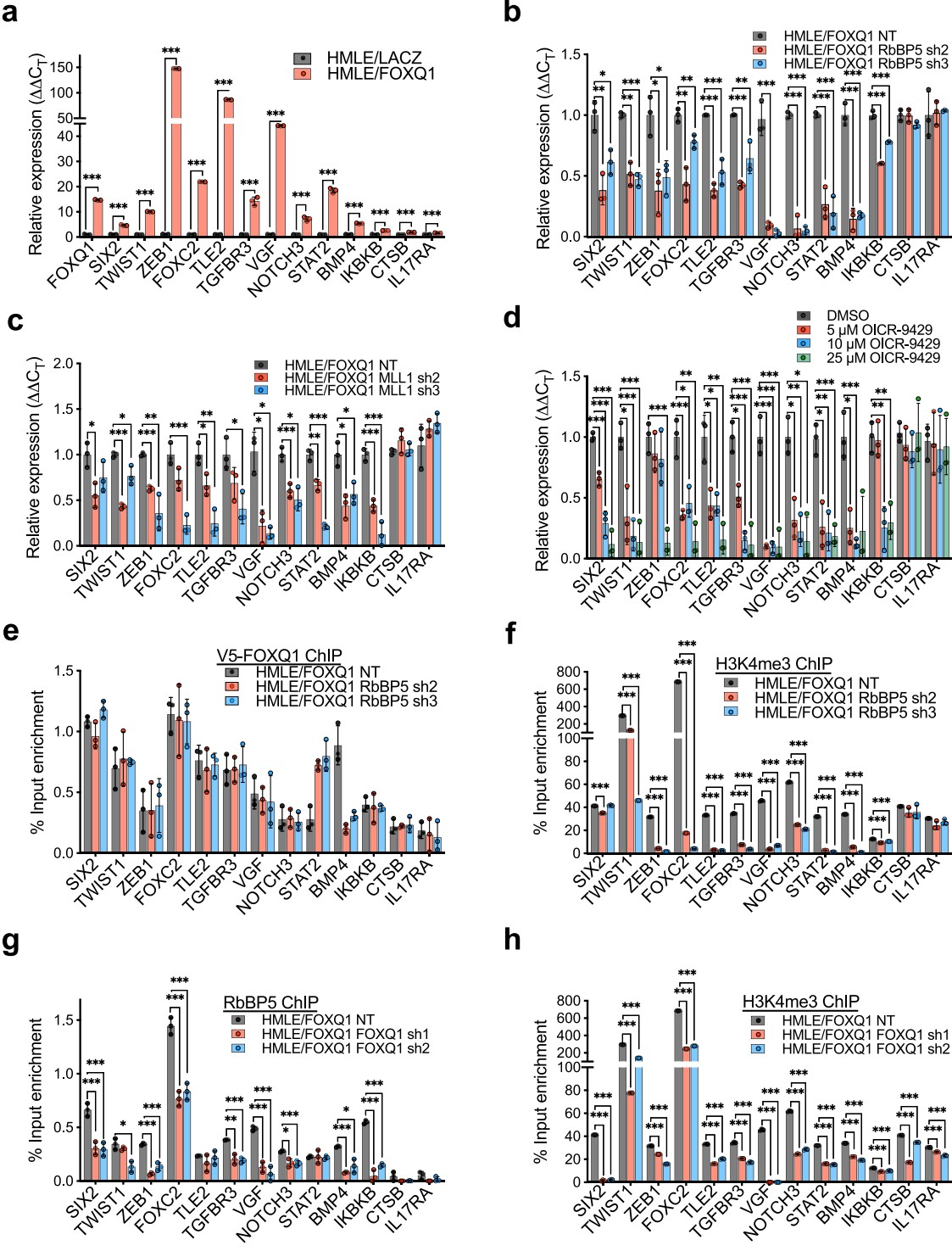

population (Fig. 7d, e and Supplementary Fig. 7d). MDA-MB-231/shFOXQ1 cells with restored expression of mutant-FOXQ1 (A129S or I132S) displayed a decreased expression of mesenchymal marker proteins compared to the WT group, while the epithelial marker occludin remained unchanged (Fig. 7f). Moreover, MDA-MB-231/shFOXQ1 cells with mutant-FOXQ1 expression (A129S, I132S) exhibited decreased invasion, migration, mammosphere formation, and more prominent epithelial-like morphology, compared to WT-FOXQ1 rescued cells (Fig. 7g–i, and Supplementary Fig. 7e–g). Finally, MDA-MB-231/shFOXQ1 with mutant-FOXQ1 expression displayed a decrease in CD44 expression and an increase in CD24 expression relative to WT-FOXQ1 cells, supporting a reduction in stem-like character (Fig. 7j).

The effects of disrupting the FOXQ1-RbBP5 interaction were assessed in vivo through orthotopic xenografts of HRas-transformed HMLE cells (HMLER) expressing either WT- or mutant-FOXQ1 (I132S, A129S) in NSG mice. The HMLER/FOXQ1-WT group showed a median onset of palpable tumor on day 15. In contrast, mice implanted with mutant HMLER/FOXQ1-A129S or HMLER/FOXQ1-I132S had a median tumor onset on day 42 or 57, suggesting a significant delay in tumor initiation (Supplementary Fig. 8a). Concomitantly, the HMLER/FOXQ1-WT group had a median time to the endpoint of 8 days (23-15), while FOXQ1-A129S and FOXQ1-I132S groups had a median time to the endpoint of 10 (52-42) days and 18 (75-57) days (Supplementary Fig. 8a–c), indicating the FOXQ1-mutant groups had a delay in tumor

**Fig. 3 | The MLL core complex is required for transcriptional activation by FOXQ1. a** Validation of a panel of FOXQ1 upregulated genes in HMLE/FOXQ1cells. *CTSB* and *IL17RA* serve as RbBP5-independent controls. Data were analyzed by $2^{-\Delta\Delta CT}$ method with internal sample normalization against β-actin and compared to the HMLE/LACZ control cells ($n = 3$). **b** The effects of RbBP5 knockdown (KD) on the expression of a panel of FOXQ1-RbBP5 EMT targets in HMLE/FOXQ1 cells. Results are relative to HMLE/FOXQ1 nontarget (NT) control cells normalized to β-actin. **c** The effects of KMT2A/MLL1 silencing on the expression of EMT target genes were assessed by qRT-PCR of HMLE/FOXQ1 shMLL1 cells. Results are relative to HMLE/FOXQ1 nontarget (NT) control cells normalized to β-actin. **d** The effects of OICR-9429 treatment on the expression of the FOXQ1-RbBP5 EMT gene panel were evaluated by qRT-PCR in HMLE/FOXQ1 cells. Results are relative to DMSO mock treatment and normalized to β-actin. **e** The impact of RbBP5 KD on FOXQ1 target promoter occupancy was assessed using V5-FOXQ1 ChIP-qPCR from HMLE/FOXQ1 cells with shRbBP5 or NT control. Signals were normalized to the input chromatin sample. **f** The impact of RbBP5 KD on H3K4me3 levels within the promoters of FOXQ1 target genes was evaluated via H3K4me3 ChIP-qPCR from HMLE/FOXQ1 cells with shRbBP5 or NT control. Results were evaluated by qPCR, and the data were normalized to a matched DNA sample from 1% input chromatin. **g** The impact of FOXQ1 on RbBP5 occupancy within EMT promoter regions was evaluated via RbBP5 ChIP-qPCR in HMLE/FOXQ1 cells with shFOXQ1 or NT control. Results were analyzed using the same approach as panel **f**. **h** The effects of FOXQ1 depletion on H3K4me3 levels within EMT promoter regions were assessed by H3K4me3 ChIP-qPCR in HMLE/FOXQ1 cells with FOXQ1 shRNA or NT control. Results were analyzed using the same approach as panel **f**. For **a–h** panels, bars indicate the mean ± SD ($n = 3$). Dots indicate the individual replicate values normalized to the control group's mean. Results were analyzed by unpaired, two-tailed *t*-test with Bonferroni multiple comparison adjustment. *$p < 0.05$, **$p < 0.01$, ***$p < 0.001$. Source data are provided in Source data 1.

progression. In addition, tumors formed from HMLER/FOXQ1 mutant cell lines (A129S, I132S) had decreased vimentin expression than HMLER/FOXQ1-WT tumors, reflecting a difference in EMT character between these tumors (Supplementary Fig. 8d).

Lastly, we evaluated the effect of targeting the FOXQ1-MLL1 axis on TNBC oncogenic properties and metastatic tumor progression. We found that shMLL1 significantly impaired the EMT phenotype in MDA-MB-231 cells, demonstrated by an apparent mesenchymal to epithelial transition and a reduction in mesenchymal marker Fibronectin expression (Supplementary Fig. 8e, f). Interestingly, we observed a significant decrease in cell migration and invasion (Fig. 8a, b), but not in cell proliferation in MLL1 knockdown cells than in control cells (Fig. 8c). MDA-MB-231 cells with shMLL1 also displayed a decrease in the CD44+/CD24− population compared to NT control cells (Fig. 8d). We then implanted MDA-MB231 cells with and without MLL1 knockdown into the mammary fat pads of female NSG mice. The MDA-MB231NT group showed a median tumor onset at day 15, whereas mice implanted with MDA-MB231 sh1 and sh3 had a median tumor onset at day 18 (Fig. 8e). Moreover, tumor progression from initiation to endpoint spent 31 days for the NT group and about 32 days for two MLL1 knockdown groups (Fig. 8e). These results suggest no significant difference in tumor initiation and progression among MLL1 knockdown and control groups. With H&E staining, intratumoral cell morphology appeared to be more epithelial-like (Fig. 8f, top panels), with lower mesenchymal marker Fibronectin expression in MLL1 knockdown tumors than those tumors with nontarget control (Fig. 8f, bottom panels). Consistent with the in vitro results, the tumor proliferation was not significantly changed upon MLL1 knockdown, as determined by Ki-67 staining (Fig. 8f, middle panels). Finally, the lungs from each mouse were removed, sectioned, H&E stained, and examined via microscopy. We observed a significant decrease of metastatic lesions in the lungs of mice engrafted with MLL1 knockdown cell lines relative to that in mice engrafted with the NT control (Fig. 8g, h).

## Discussion

The FOX family members possess diverse functions in homeostasis and cancer, but the distinct mechanisms are not fully understood[52–55]. Here, we propose a model in which FOXQ1 directly binds the RbBP5 subunit, utilizing the H1-S1 region of the FHD, and further recruits the MLL core complex to establish a local H3K4me3 histone-code within the promoter regions of EMT genes to facilitate transcriptional activation (Fig. 9). FHD is well-recognized as a conserved DNA binding domain within the FOX family that recognizes a similar motif[56]. Our current study identified a FOX FHD harboring another essential function as a protein interaction and transactivation domain[57]. Interestingly, one of the mutations (I132S) identified as critical for FOXQ1 to bind RbBP5 was reported to produce a "satin hair" phenotype in mice[4]. However, a biological mechanism has yet to be reported. Here, we show that both the human FOXQ1-I132S and nearby

A129S mutations confer loss-of-function in FOXQ1 recruitment of RbBP5 as a transcriptional coactivator without altering FOXQ1 nuclear localization or promoter binding capability. Therefore, this "satin hair" region may also be critical for FOXQ1 binding to cofactors in other biological contexts.

Additionally, we demonstrate that RbBP5 binding to FOXQ1 is dependent upon the RbBP5 "hinge region", which also houses binding sites of MLLs, ASH2L, and WDR5[50,51]. However, our results suggest that the FOXQ1-RbBP5 interaction does not interrupt the formation of the MLL core complex. Our data has repeatedly identified FOXQ1 bound to intact MLL core complex as shown through FOXQ1 TAP-MS in HEK293 cells and FOXQ1 or RbBP5 IP from the mammary cell model (HMLE/FOXQ1), TNBC cell lines, and tumor samples (Fig. 1). We also observe FOXQ1 and RbBP5 chromatin co-localization associated with regions of high levels of H3K4me3, suggesting FOXQ1 does not compete with MLL enzymatic activity. Therefore, our data support a two-step model in which FOXQ1 recognizes and binds to specific DNA sequences (a step independent of the MLL core complex) but has minimal effect on the transcriptional activation of these target genes. FOXQ1 then recruits RbBP5, which facilitates MLL complex assembly and deposition of H3K4 trimethylation. Determining the exact amino acids responsible for FOXQ1 binding within the small hinge region (containing MLLs, ASH2L, and WDR5 binding sites) warrants further work deciphering the 3D structure of the FOXQ1 FHD/RbBP5 interacting interface.

Mounting evidence showed that H3K4me3 enrichment correlated with EMT. McDonald et al.[58] investigated epigenetic modifications during EMT mediated by TGF-β. They found a global reduction in the heterochromatin mark H3K9me2, an increase in the euchromatin mark H3K4me3 and the transcriptional mark H3K36me3. However, DNA methylation was unchanged during EMT. In the process of TGFβ1 induced EMT in the prostate cancer cell line DU145, H3K4me3 enrichment and RbBP5 binding increased in the vicinity of the Snail (SNAI1) transcription start site. Knockdown of RbBP5 notably decreased Snail expression and EMT. Recruitment of RbBP5 and formation of H3K4me3 at Snail TSS during EMT depend on the binding of SMAD2/3 and CBP at Snail TSS[59]. More recently, SETD1A was shown to promote lung cancer progression via several critical oncogenes, which exhibited enhanced H3K4me3 levels around transcriptional start sites[60]. Our current study provided mechanistic evidence of the involvement of H3K4me3 in EMT promotion through a direct physical interaction of an EMT-driving gene FOXQ1 and an MLL core complex subunit RbBP5 in both normal epithelial cells and TNBC cancers, suggesting the epigenetic machinery could be commonly implicated in tumor progression.

In addition to the EMT-related signaling pathways such as WNT, PDGF, TGFβ, and cadherin, FOXQ1-MLL interaction also directly activate other signaling pathways, including lysosome, nucleotide sugar metabolism, and inflammatory mediator regulation. Moreover, there

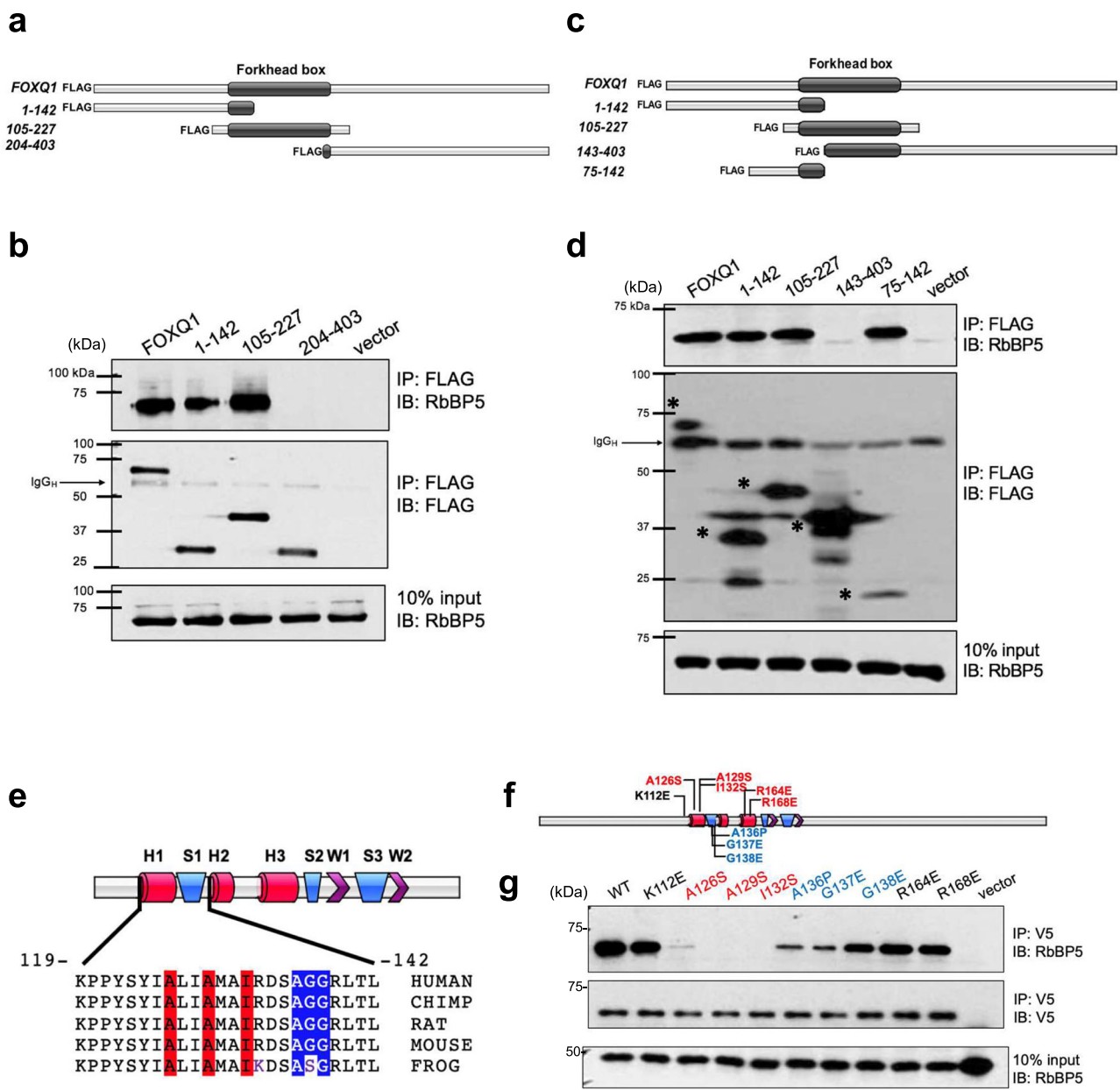

**Fig. 4 | FOXQ1 binds to RbBP5 within the Forkhead box domain. a** Schematic of the FOXQ1 fragment subcloning. N-terminal (1-142), Forkhead box (105-227), and C-terminal (204-403) were fused to FLAG-tag. **b** Fragments were transfected into HEK293 cells with full-length FOXQ1 and vector control. Lysates were subject to IP with anti-FLAG resin and analyzed by western blot with an antibody against endogenous RbBP5. **c** Schematic of constructs designed to narrow the domain of interaction within FOXQ1., Additional constructs of the FOXQ1 N-terminal fragments (1–142) and the Forkhead box fragment (105–227) were transfected alongside C-terminal residues 142–403 and N-terminal 75–142 to refine the region responsible for the RbBP5 interaction. **d** IP western blot analysis with membranes probed with

the RbBP5 antibody. *Indicates predicted band. **e** Analysis of the evolutionary conservation of the H1-S1 region of the FOXQ1 Forkhead box. Red cylinders indicate α-helix domains, blue rectangles indicate β-sheets, and purple triangles indicate winged regions. The highlighted residues alter RbBP5 binding upon mutation. **f** Schematic of site-directed mutagenesis within the FOXQ1 Forkhead box. **g** WT and mutant FOXQ1 vectors were transfected into HEK293 cells. Lysates were subject to anti-V5 IP, and the effects of the point mutation of FOXQ1 on RbBP5 binding were analyzed by western blot. For panels **b**, **d**, **g**, the image serves as a representative result from experiments performed in triplicate (*n* = 3). Unprocessed western blot images are provided in Source data 2.

are 2499 genes upregulated by FOXQ1, and 791 (32%) genes display a FOXQ1 binding in the promoter. The other 68% of the upregulated genes enriched diverse functions such as hypoxia signaling, angiogenesis, and extracellular matrix organization. In line with this, many EMT-related TFs, including ZEB2, TWIST2, SIX1, and FOXF1, did not show FOXQ1 binding in their promoters. This observation can be partially explained by FOXQ1 binding to the possible enhancer rather than promoter regions of these gene targets. As shown in Fig. 2a, FOXQ1 binding peaks are enriched in both the proximal promoter regions and the distal intergenic regions. Whether the transcriptional

regulation of EMT master genes, including ZEBs and TWISTs, is the primary mechanism of FOXQ1 mediated EMT, how FOXQ1 binding to the enhancer of downstream genes, and how these transcription activities contribute to FOXQ1 promoted cancer progression will be interesting research questions for us to follow.

Individual components of the MLL core complex have been implicated in solid tumor progression. WDR5 overexpression was found to contribute to tumorigenesis[61,62] and to promote metastasis in prostate and colon cancer[63,64]. ASH2L was shown to cooperate with HRAS in fibroblast transformation and tumorigenesis[65]. MYC was

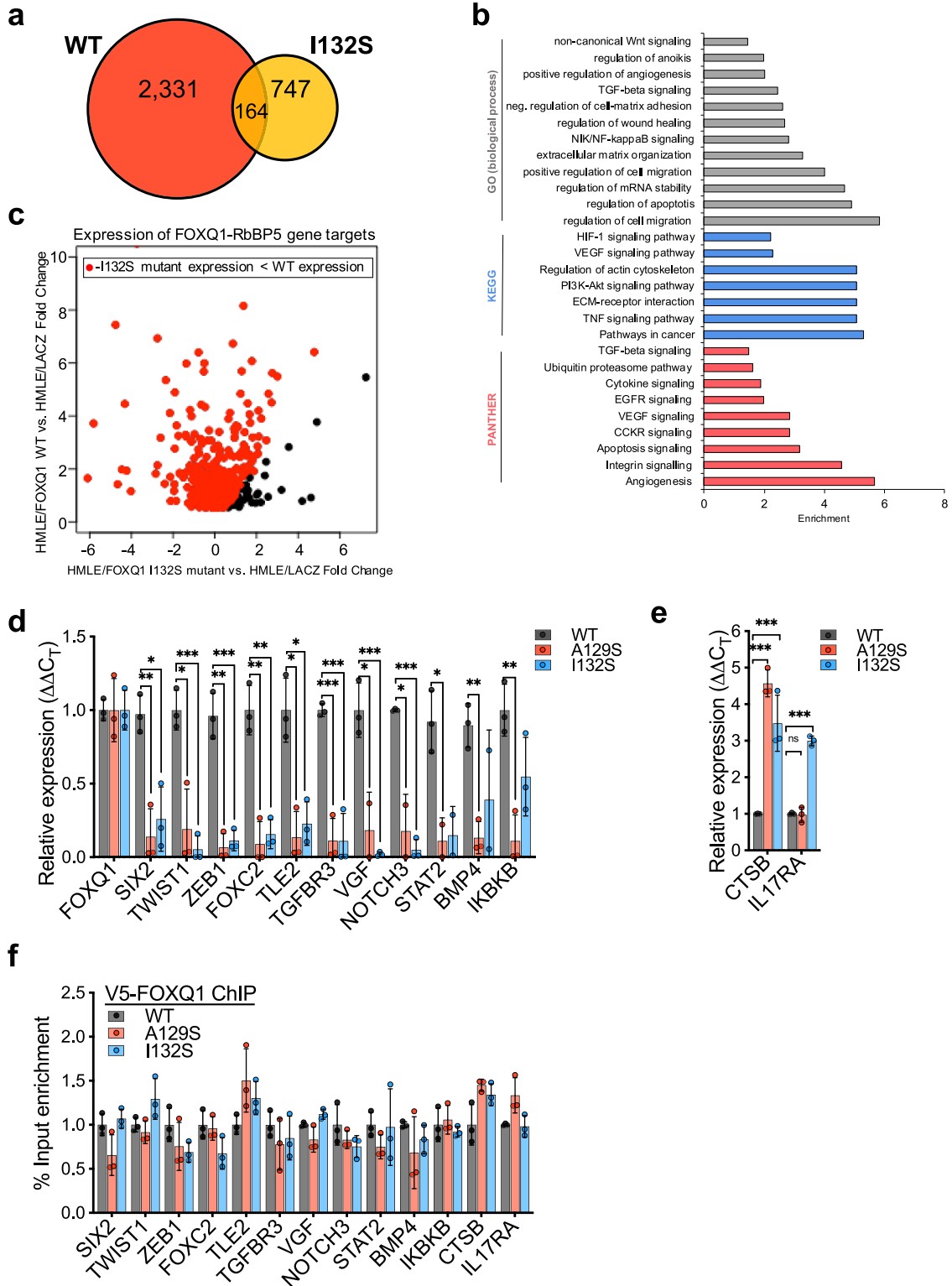

independently complexed with ASH2L or WDR5 to regulate MYC-target gene expression through MLL-independent mechanisms[31,32,66,67]. Specifically, MYC/ASH2L axis was not associated with H3K4me3 but was related to transcriptional activation by H3K27 acetylation through p300/CBP[67]. Pioneer factor FOXA1, a recognized EMT suppressor, was reported to recruit MLL3 to deposit H3K4me1 within FOXA1-bound enhancers in ER-positive breast cancer cells[68,69], but whether the core complex is involved in this process remains unknown. In contrast to these reports, our study demonstrates that the entire MLL core

complex is an essential cofactor for the FOXQ1-activated EMT program and metastatic progression in breast cancer. Although targeting MLL enzymatic activity and global H3K4me3 is an active research area in leukemia, the role of sequence-specific oncogenic TFs in directing MLL activity in different cancers warrants further investigation.

Our results suggest the FOXQ1-RbBP5 interaction may be a promising therapeutic target for targeting TNBC progression or metastatic tumor recurrence. Moreover, two viable strategies can be applied to this kind of therapy. First, inhibitors targeting MLL activity

**Fig. 5 | Disrupting the FOXQ1-RbBP5 protein interaction attenuates the EMT program. a** Venn diagram compares the sets of differentially upregulated genes in HMLE/FOXQ1-WT and HMLE/FOXQ1-I132S cell lines relative to HMLE/LacZ, identified by RNA-seq. **b** Enrichment analysis of the differentially upregulated genes in HMLE/FOXQ1 I132S mutant cells relative to HMLE/LacZ using Enrichr. The differentially regulated gene set was compared to the annotated sets from the indicated tool by Fisher's exact $t$-test with Benjamini-Hochberg multiple testing correction. Enrichment is plotted as the $\log_{10}$ adjusted $p$-value. **c** Scatter plot of the fold-change in gene expression of the 622 genes classified as direct FOXQ1-RbBP5 targets for HMLE/FOXQ1-WT and HMLE/FOXQ1-I132S cell lines, both relative to HMLE/LacZ. The genes that display decreased expression in the HMLE/FOXQ1-I132S mutant cells compared to HMLE/FOXQ1 WT are highlighted in red. **d** Expression of FOXQ1-RbBP5 EMT targets, assessed by qRT-PCR, in HMLE cells with stable overexpression of mutant FOXQ1 (A129S or I132S). Values for each target gene assessed were

normalized to respective β-actin and compared relative to HMLE cells with WT FOXQ1 expression. **e** Expression of RbBP5-independent FOXQ1 targets (*CTSB, IL17RA*) in HMLE/FOXQ1 mutant lines (A129S or I132S) relative to HMLE/FOXQ1 WT. **d, e** Data were normalized against β-actin and compared to the WT-FOXQ1 control group. Dots indicate the individual replicate values normalized to the mean of the WT control group. **f** The promoter binding activity of FOXQ1-WT or RbBP5-deficient mutants, FOXQ1-A192S or I132S, was assessed by V5 ChIP-qPCR from respective stable HMLE cells. Data were normalized to a matched DNA sample from 1% input chromatin. For **d–f**, bars indicate the mean ± SD. Experiments were performed in triplicate, and dots indicate each replicate value ($n = 3$). Results were analyzed by unpaired, two-tailed $t$-test with Bonferroni multiple comparison adjustment. For all experiments, $*p < 0.05$, $**p < 0.01$, $***p < 0.001$. Source data for panels d-f were provided in Source data 1.

could be used to reverse FOXQ1-driven tumor progression. The bottleneck of this approach is that most current available MLL inhibitors have not been tested in vivo and their specificity and pharmacokinetics or pharmacodynamics need to be determined and improved. Second, specific small molecular inhibitors can be developed to interrupt FOXQ1-RbBP5 interaction to nullify FOXQ1 function in cancer progression. However, a drug screen is required to identify a small molecule that is bioavailable with minimal toxicity in vivo. This type of inhibitor could help overcome significant challenges of targeting transcription factors, including drug efficacy and specificity, since it specifically targets the protein-protein interaction necessary to regulate a tumor-promoting transcription program[70].

## Methods

### Cell culture

HEK293T (CRL-11268), MDA-MB-231 (HTB-26), MDA MB 468 (HTB-132), and MDA MB-436 (HTB-130) cell lines were obtained from and characterized by cytogenetic analysis by American Type Culture Collection (ATCC, Manassas, VA). Cells were authenticated by comparing them to the original morphological and growth characteristics and were verified using the GenomeLab short tandem repeat (STR) profiling (Beckman Coulter) with >90% match. DAPI stain and Immunofluorescence microscopy tested all cell lines for mycoplasma negative. HEK293T, MDA-MB-231, and MDA-MB-468 cell lines were cultured in DMEM with 10% FBS and 1% penicillin/streptomycin at 37 °C with 5% $CO_2$. MDA MB-436 cells were cultured in Leibovitz's L-15 medium with ten μg/ml insulin, 16 μg/ml glutathione, and 10% fetal bovine serum at 37 °C with 100% air. SUM1315 cells were obtained from Dr. Stephan P. Ethier at MUSC and were grown in Ham's F12 (HyClone), 5% FBS, 1% pen/strep, EGF (10 μg/L), Insulin (5 μg/L). HMLE and HMLE-Ras cells were obtained from Dr. Robert A. Weinberg's laboratory at MIT and were cultured in 50:50 DMEM/F12 media (HyClone) containing 10% FBS, 1% pen/strep, EGF (10 μg/L), Insulin (10 μg/L) and hydrocortisone (500 ng/L). The presence of the SV40 large T antigen and a catalytic subunit of telomerase in the HMLE cell line was confirmed by PCR. HMLE cells with stable overexpression of WT- or mutant-FOXQ1 were selected and maintained in media with 10 μg/mL blasticidin (InvivoGen, ant-bl-1). HMLE cells with stable shRNA knockdown were established and maintained in media with puromycin (InvivoGen, ant-pr-1) at 1 μg/mL.

### Plasmids and cloning

For the generation of FOXQ1 fragment constructs, wild-type pENTR-FOXQ1 was used as the DNA template. FOXQ1 was subcloned into GST vector, pGEx-6p2 (Amersham Biosciences, 18-1157-58), with GST fused to the N-terminal by PCR with 5′ BamHI and 3′ XhoI restriction sequences. FOXQ1 fragments were cloned into P3XFLAG-CMV 7.1 vector (Sigma-Aldrich, E7533) with N-terminal FLAG-epitope tag with 5′ HindIII and 3′ KpnI endonuclease sequences. RbBP5 plasmid was purchased from Addgene (#15550). RbBP5 regions were cloned into

pCMV-MYC (BD Biosciences, K6003-1) by PCR with N-terminal MYC-epitope tag with 5′ EcoRI and 3′ KpnI sequences. PCR products were purified by gel purification (Qiagen) and subjected to double digestion overnight at 37 °C. The double digestion product was purified by gel purification. The ligation reaction was performed with a 5:1 insert to vector molar ratio with T4 DNA ligase and incubated at 16 °C overnight. Five μL ligation reaction was heat shock transformed into 25 μL DH5α competent cells. Clones were picked from single colonies. DNA plasmids were purified by Qiagen mini and midi preps according to the manufacturer's protocol. Plasmids were validated through double digestion and Sanger sequencing (GenScript).

### Site-directed mutagenesis

Mutagenesis was conducted using QuikChange II XL Site-Directed Mutagenesis Kit (Aligent, cat #200521-5) according to the manufacturer's protocol. The pENTR-FOXQ1 plasmid was used as a template for mutagenesis. PCR reaction volume was 20 μL with 1X buffer, dNTPs (200 μM each), 0.5 μM forward primer, 0.5 μM reverse primer, 0.02 U/μL PfuUltra HF DNA Polymerase, and five ng template DNA. Reactions were initially denatured at 95 °C for 2 min and cycled at 95 °C denaturing (1 min), 58 °C annealing (1 min), 68 °C extension (4 min) for 18 cycles, followed by a final extension (7 min). Parental template plasmid was subject to DpnI digestion with one μL enzyme added to the PCR mixture and incubated at 37 °C for 1 h. 5 μL of the resultant PCR mixture was used for ligation reaction in 1X Rapid Ligation Buffer, and 0.5 μL T4 DNA ligase for 5 min at room temperature. Five μL of ligation mixture was used for heat shock transformation of 25 μL competent cells (Aligent, XL 10-Gold Ultacompentent Cells, cat# 200315). Clones were selected and validated by Sanger sequencing (Genscript). The resultant FOXQ1 mutant inserts were transferred from pENTR to plenti6/UbC/V5-DEST (Invitrogen) by recombination. 150 ng of pENTR vector was combined with 150 ng of plenti6 vector in TE buffer and two μL LR ClonaseTM II enzyme mix with a final reaction volume of 10 μL. The recombination reaction was incubated for 4 h at 35 °C. Then, 1 μL of proteinase K solution (20 mg/mL) was added to the recombination mixture and incubated at 37 °C for 30 min. 5 μL of recombination reaction was added to 25 μL OneShot Stbl3 competent cells for heat shock transformation at 42 °C. Clones were selected from overnight colonies and validated by restriction digest with BamHI and XhoI enzymes. Plasmids were purified with Qiagen mini and midi prep kits according to manufacturer guidelines. plenti6 construct expression was validated with transient transfection in HEK293T cells and western blot analysis before viral production.

### Viral transduction

**plenti viral overexpression.** Systems for Gateway cloning into the plenti6/V5-DEST expression vector and lentiviral production were obtained from Invitrogen. To generate the virus, HEK293T cells were seeded overnight at a confluence of 90%. Cells were transfected in

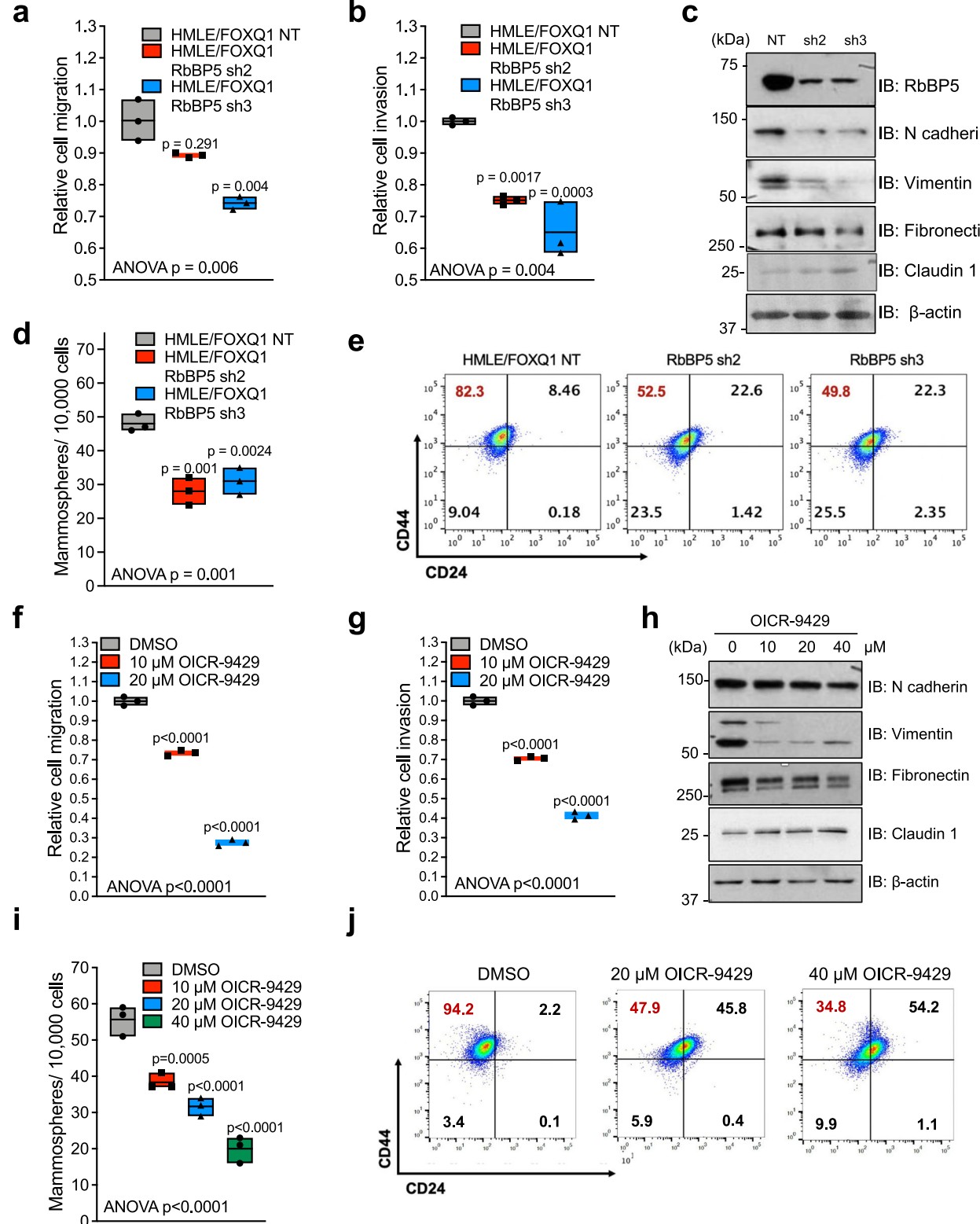

serum-free, antibiotic-free media (DMEM). ViraPower Packaging Mix (Invitrogen) was combined with a plenti6 expression vector at a 3:1 ratio diluted in OPTI-MEM media. In a separate tube, the Lipofectamine 2000 reagent was diluted (3:1 Lipo: DNA ratio) in OPTI-MEM. Mixtures were incubated separately for 5 min at room temperature. DNA and Lipofectamine mixtures were combined and incubated for 20 min at room temperature. The transfection reagent was added dropwise to the HEK293 cells and incubated overnight in the culture incubator. The following day, the media was replaced with antibiotic-free media (DMEM/10% FBS). Virus-containing supernatants were harvested 48- and 72 h post-transfection. For viral transduction, cells were seeded in 6 well plates to 25% confluence overnight. Viral transduction was conducted in a total volume of 1 mL complete culture media with 10 μg/mL polybrene for 24 h. Media was changed to complete media. At 48 h post-transduction, the selection was begun with blasticidin to a concentration of 10 μg/mL. Surviving cells were diluted to colonies,

**Fig. 6 | Targeting the MLL core complex inhibits FOXQ1-driven EMT and oncogenesis in vitro. a, b** RbBP5 regulates the migratory and invasive capabilities of HMLE/FOXQ1 cells. Transwell migration (**a**) and invasion (**b**) assays ($n = 3$) were conducted in cells with RbBP5 knockdown or NT control. The floating bar graph displays the mean, minimum and maximum values. Individual dots depict the cell numbers relative to the control group average for each replicate. Results were analyzed by one-way ANOVA with Bonferroni correction for multiple comparisons (*p* values for panels **a** and **b** are 0.006 and 0.004, respectively). **c** Western blot analysis of the EMT markers expression in HMLE/FOXQ1 cells with shRbBP5 KD or NT control. The β-actin loading control was probed on the same membrane. Representative images, $n = 3$. **d** Mammosphere formation assay for HMLE/FOXQ1 RbBP5 KD or NT control cells ($n = 3$). Spheres ≥50 μm in diameter were counted as mammospheres. Presentation and statistical analyses are the same as panels **a, b** (*p* = 0.001). **e** The abundance of a CD44hi/CD24low stem-like population was of HMLE/FOXQ1 cells with RbBP5 stable KD or NT control by flow cytometry. **f, g** Quantification of migration (**f**) and invasion (**g**) assays for HMLE/FOXQ1 cells treated with 10 and 20 μM OICR-9429 or DMSO control for 72 h. Cells were plated in triplicate ($n = 3$). The dots indicate replicate measures normalized to the mean of the DMSO control. Presentation and statistical analyses are the same as panels **a, b** (*p* < 0.0001 for both panels). **h** Expression of mesenchymal and epithelial markers in HMLE/FOXQ1 cells after 72-hour treatment with indicated doses of OICR-9429 or DMSO control by western blot. The β-actin loading control was probed on the same membrane. Representative images, $n = 3$. **i** Mammosphere formation of HMLE/FOXQ1 cells with OICR-9429 treatment for 72 h ($n = 3$). Presentation and statistical analyses are the same as panels **a, b** (*p* < 0.0001). **j** Flow cytometry quantification of the CD44+/CD24- population in HMLE/FOXQ1 cells following 72 h of OICR-9429 treatment with indicated doses. For all experiments, Source data are provided in Source data 1.

and clones were selected and monitored for target gene overexpression by qPCR and V5-tagged western blot.

**short-hairpin RNA (shRNA) knockdown.** shRNA targeting the knockdown of RbBP5 was purchased from Dharmacon. To generate the virus, HEK293T cells were seeded overnight at a confluence of 90%. The following day, 1 μg shRNA (pLKO.1 plasmid) was combined with 250 ng pMD2.G, 350 ng pMDLg/pRRE and 350 ng pRsv/Rev packaging plasmids (addgene, cat#12259, #12253, #12251) in OPTI-MEM. The subsequent transfection into HEK293, viral production, and transduction steps are the same as above. At 48 h post-transduction, the selection was begun with puromycin to a concentration of 1 μg/mL. Surviving cells were expanded and monitored for target gene knockdown by western blot. The information on all shRNA used in this project was provided in Supplementary Table 1.

## Transfections
HEK293 cells were plated overnight to approximately 70% confluence in a 6 cm plate. 2 μg plasmid DNA was diluted in 250 μL of OPTI-MEM. 8 μg of polyethyleneimine (PEI) was diluted in 250 μL of OPTI-MEM. Diluted mixtures were incubated for 5 min at room temperature. DNA and PEI solutions were combined and mixed by pipetting, and the resultant DNA: PEI solution was incubated for 15 min at room temperature. The transfection mixture was added dropwise to HEK293 cells in 2 mL of fresh, complete medium. Transfections were allowed to proceed for 48 h before further experimental analysis.

## RT-qPCR and ChIP-qPCR
Adherent cells were washed with 1× PBS prior to the addition of TRIZOL reagent directly to the cell plate and collected by scraping. RNA was isolated by TRIZOL, chloroform reaction, and isopropanol precipitation. Samples were washed with 70% ethanol and allowed to air-dry prior to resuspension in DEPC water. cDNA was generated using the SuperScript III First-Strand Synthesis System (ThermoFischer, 108080-051) according to manufacturer protocol with 1 μg isolated RNA. The resultant cDNA was diluted to 100 μL with DEPC water. qPCR reactions were conducted using PowerUP SYBR Green MasterMix (Applied Biosystems A25742). The sample volume was 2 μL for both cDNA or ChIP DNA samples. Reactions were run in triplicate. Primers were added to a final concentration of 500 nM. RT-qPCR was analyzed by standard $\Delta\Delta C_T$ method with β-actin as the loading control. ChIP-qPCR enrichment was calculated relative to 1% input DNA ($\Delta C_T$). The data were analyzed in Microsoft Excel (Version 16.40) and Prism 8 (Version 8.4.3). The information on all primers, including cloning primers, mutagenesis primers, and PCR primers, was provided in Supplementary Table 2.

## Western blot analysis
Protein was harvested by scrapping cells in 1× ice-cold PBS and centrifugation. Cells were washed three times with 1× PBS. Pellets were lysed in Cell Extraction Buffer (Life Technologies, FNN0011) with protease inhibitors (Halt Protease/Phosphatase Inhibitor, Thermo Scientific 1861289) by incubation on ice for 1 h with intermittent vortexing. Protein was quantified by Bradford assay (BioRad Quick Start Bradford, Cat #500-0205). 20–50 μg of protein was loaded per gel for SDS-PAGE and transferred to the nitrocellulose membrane. Membranes were blocked with 5% milk/TBST for 1 h and incubated in primary antibody diluted in 5% BSA/TBST at 4 °C overnight. See the extended table for antibody specifications. Membranes were washed with TBST and incubated in a secondary antibody diluted in 5% milk/TBST for 1 h at room temperature. Blots were developed with Pierce ECL substrate (Thermo Fischer, 32106) and visualized by autoradiograph film exposure (HyBlot Cl). β-actin was used as the loading control and was probed on the same membrane. Panels constructed from multiple membranes will have multiple, distinct β-actin images in the panel. The information on all antibodies used in this project was provided in Supplementary Table 3. All uncropped blots were presented in Source data 2.

## Co-immunoprecipitation
For immunoprecipitations, cells were lysed in Co-IP buffer (20 mM Tris-HCl, 140 mM NaCl, 10% glycerol, 1% NP-40, 2 mM EDTA, and protease inhibitor cocktail) for 1 h on ice with frequent vortex. Lysates were incubated with protein A or protein G agarose (Abcam, Ab193258) for 2 h with rotation at 4 °C for pre-clearing. Pre-cleared samples were incubated with the indicated primary antibodies and protein A or protein G agarose beads overnight at 4 °C on a rotator. IPs for overexpressed proteins with epitope tags were incubated with antibody-conjugated beads. Immunocomplexes were collected and washed in Co-IP buffer three times, and samples were boiled at 95 °C for 5 min with protein loading buffer. Beads were centrifuged, and the supernatant was analyzed, alongside input lysate, by western blot.

For validation of the interaction of FOXQ1 and RbBP5 in tumor samples, we obtained the lung and ovarian tumor samples from the AMTEC core facility in KCI and three TNBC tumors from Dr. Huiping Liu at Northwestern University. These tumor samples are cryopreserved and without patient identity. Each tumor sample tissue (around 2 mg) was disintegrated in the frozen state by ball mill grinding, followed by extraction and solubilization in 2% SDS for 10 min at 70 °C in a volume corresponding to ten times the wet tissue weight with shaking. The resulting protein extracts were then used for further IP and western blotting analysis.

## GST pull down
**MLL core complex protein expression and purification.** All MLL complex subunits, i.e., RbBP5, WDR5, and ASH2L, were expressed using the pET-28a expression vector with N-terminal 6-histidine and SUMO tag. All proteins were expressed in BL21(DE3) *E. coli* strain in LB media. Cells were grown initially at 37 °C until $OD_{600}$ reached 0.6–0.8 and shifted to 20 °C after Isopropyl β-D-1-thiogalactopyranoside

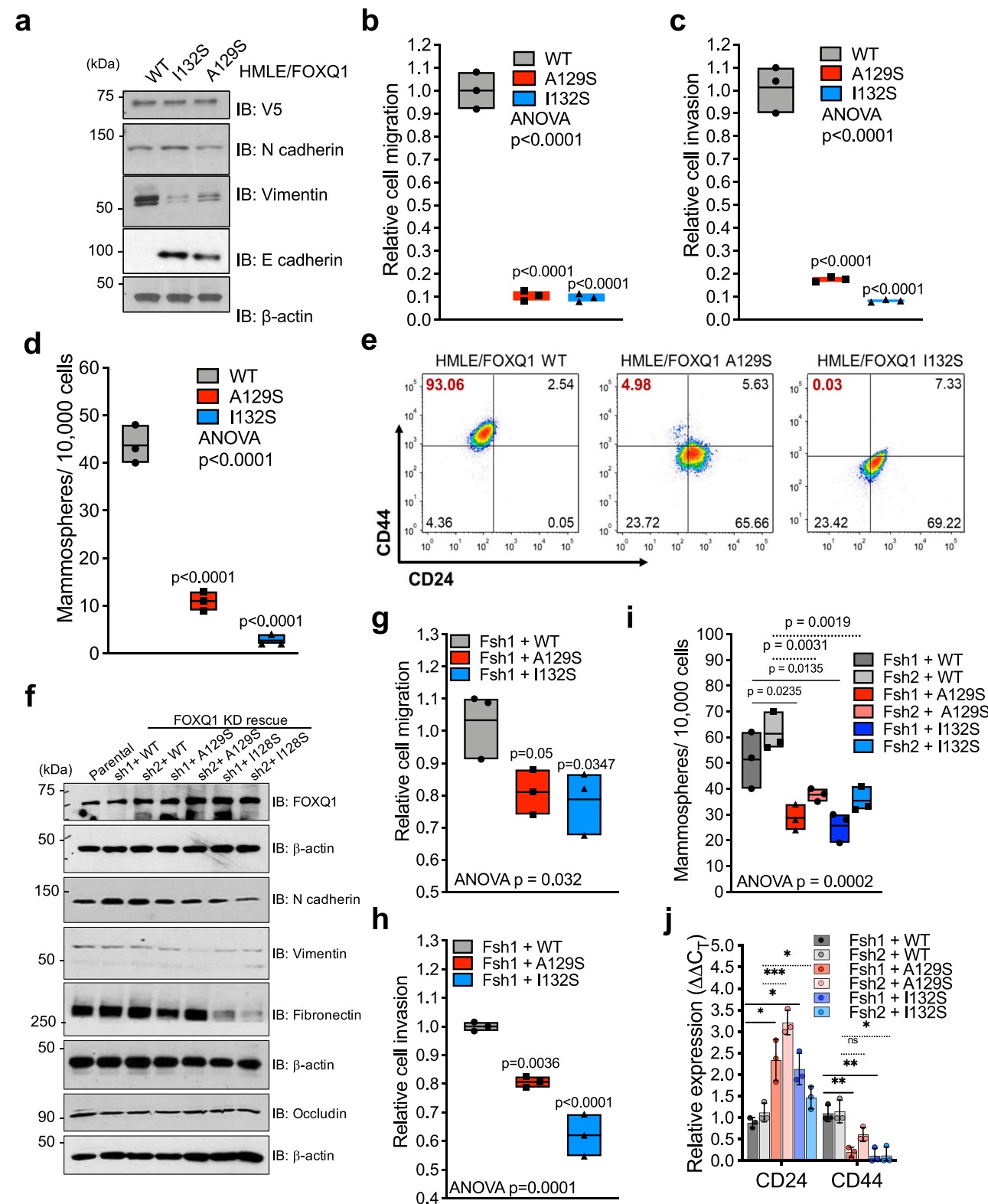

(IPTG) was added at a final concentration of 0.2-0.4 mM. Cells were lysed by sonication, and lysates were collected after centrifugation at 32,000 × *g* at 4 °C. The supernatant was filtered through a 0.45 μm syringe filter and purified through a Ni-NTA metal-affinity column (Qiagen and Goldbio). After extensive washing with 20 mM Tris (pH 8.0), 300–500 mM NaCl, 10 mM β-mercaptoethanol and 10 mM imidazole (washing buffer), protein was eluted stepwise at 30, 60, 90, 120, 150, 210, and 300 mM imidazole. SUMO protease was added to the

pooled fractions during dialysis at 4 °C overnight. Ni-NTA purification was repeated to remove the 6-histidine tag and other bacterial impurities. Proteins were further purified on a HiLoad 16/60 Superdex 75PG or 200PG columns (GE Healthcare).

**GST pull-down.** BL21 cells harboring the GST or GST-FOXQ1 were grown to an $OD_{600}$ of 0.6 at 37 °C and then induced with 0.5 mM IPTG at room temperature overnight (-18 h). Bacteria were

**Fig. 7 | Disrupting the FOXQ1-RbBP5 protein interaction attenuates the EMT program. a** Western blot analysis of mesenchymal and epithelial protein expression in HMLE cells with stable expression of mutant FOXQ1 (A129S, I132S) or WT FOXQ1. The β-actin loading control was probed on the same membrane. Representative images, $n = 3$. **b–d** Effects of FOXQ1 mutant (A129S, I132S) overexpression on cell migration (**b**), invasion (**c**), and mammosphere formation (**d**) relative to WT counterpart in HMLE cells ($n = 3$). Floating bar displays the mean, minimum and maximum values. Dots indicate the value from each replicate normalized to the mean control value. The bar graph depicts the sample mean ± SD. Significance was determined by one-way ANOVA with Bonferroni correction for multiple comparisons ($p < 0.0001$ for all panels). **e** Flow cytometry quantification of the CD44$^+$/CD24$^-$ population in HMLE cells with expression of mutant FOXQ1 (A129S, I132S) or WT FOXQ1. **f** The expression of EMT markers was evaluated by Western blot analysis in MDA-MB-231shFOXQ1 cells transduced with FOXQ1 mutants (A129S, I132S) or WT-FOXQ1. Representative images, $n = 3$. **g–i** Migration (**g**), invasion (**h**), and Mammosphere formation (**i**) assays of MDA-MB-231 FOXQ1sh1 cells with stable expression of WT FOXQ1 or mutant FOXQ1 (A129S, I132S). Cells were plated in triplicate ($n = 3$). floating bar and dots were presented as panels b-d. Statistical analysis is the same as panels **b–d** (panel **g**, $p = 0.032$; panel **h**, $p = 0.0001$; and panel **I**, $p = 0.0002$). **i** Of MDA-MB-231shFOXQ1 cells with WT or mutant (A129S, I132S) FOXQ1 rescue. The assay was performed in triplicate ($n = 3$). Individual dots depict replicate values normalized to the control mean. **j** The CD24 and CD44 expression levels in MDA-MB-231shFOXQ1 cells with rescued WT-, A129S-, or I132S-FOXQ1 expression were determined by qPCR normalized to β-actin and relative to the WT-FOXQ1 group mean. Individual replicate values are points on the graph. Results were analyzed by unpaired, two-tailed $t$-test with Bonferroni multiple comparison adjustment. *$p < 0.05$, **$p < 0.01$, ***$p < 0.001$. For all panels, source data are provided in Source data 1.

collected and lysed with BC100 buffer (20 mM Tris-HCl pH 7.8, 100 mM NaCl, 10% glycerol, 0.2 mM EDTA), containing 5 mM DTT and 1% sarcosyl (w/v, final concentration). Samples were sonicated at 10-second intervals for 1 min on wet ice. After sonication, Triton-X was added to a final concentration of 1%. The solubilized proteins were recovered by centrifugation and incubated with glutathione-agarose beads (Thermo) for 1 h at 4 °C. Beads were collected and washed five times with ice-cold PBS. The resulting bead-bound proteins were then incubated with 100 µg of purified MLL core complex protein (RbBP5, ASH2L, or WDR5) in BC100 with 0.1% BSA overnight at 4 °C with rotation. The glutathione-agarose beads were washed five times with BC100 buffer with 1% Triton-X before sample boiling and loading onto SDS-PAGE.

### FOXQ1 antibody production
Anti-FOXQ1 antibody was generated with the antigen: KLEVFV-PRAAHGDKQGSDLEGAGGSDAPSPL cloned into a modified pcDNA3.1-FLAG vector. The epitope was FLAG-tag affinity purified and immunized into two rabbits. Reactive serum was run over a FLAG column to remove any antibodies which reacted with the fusion partner. Flow-through was run through a second column containing a FLAG column conjugated with FOXQ1 epitope. Antibodies were eluted with 2 mL of 0.2 M glycine pH 2.6 collected in 200 µL per fraction into 1.5 mL tubes containing 60 µL of 1 M Tris pH 8.0. Bradford assay was used to check for protein peaks, and peak fractions were combined. Antibodies were dialyzed against PBS with 40% glycerol at 4 °C overnight. The antibody was collected and stored at −20 °C.

### Immunofluorescence
Cells were plated overnight into 6-well plates with cover slips. Cells were fixed in 4% formaldehyde in PBS for 15 min at room temperature. Coverslips were washed three times with PBS for 5 min. Specimen was blocked (1× PBS/5% normal goat serum/0.3% TritonX-100) for 1 h at room temperature. A primary antibody was diluted at 1:800 in dilution buffer (1× PBS/1% BSA/0.3% TritonX-100) for 1 h at room temperature. Samples were washed three times with PBS for 5 min. A secondary antibody (Invitrogen) was diluted at 1:600 in dilution buffer for 1 h at room temperature in the dark. Samples were washed three times with PBS for 5 min. Coverslips were mounted onto slides with ProLong Gold antifade with DAPI (Invitrogen, P36941) and allowed to solidify overnight at room temperature in the dark.

### Cell proliferation assay
Sulforhodamine B colorimetric (SBC) assay was used for cell density quantification and monitored over time to monitor cell proliferation. Cells were plated at a density of 2500 cells per well in a 96-well plate with six replicates per group. Cells were fixed by the addition of cold trichloroacetic acid (TCA) to a final dilution of 5% (wt/v), and plates were incubated at 4 °C for 1 h. Plates were gently washed by submersion in water four times and allowed to dry at room temperature. Cells

were stained by adding 100 µL 0.057% SBC solution per well and incubated at room temperature for 30 min. The stain solution was aspirated and washed four times with 100 µL 1% acetic acid. Plates were allowed to dry at room temperature. SBC dye was solubilized with 200 µL 10 mM Tris base (pH 10.5) per well and incubated on a shaker at room temperature for 30 min. The absorbance of each well was measured with a multiple reader at 510 nm.

### Mammosphere formation
Cells were plated in triplicate on a 6-well ultra-low attachment plate (Corning Inc.) at 10,000 cells/well and were grown in serum-free DMEM growth medium (MEBM Basal Medium, Lonza) supplemented with 1X B27 (Invitrogen), 20 ng/mL EGF, 1 mg/mL hydrocortisone, and 5 mg/mL insulin. Media (0.5 mL) was added every other day for 7–10 days. Images of mammospheres were recorded, and the number of mammospheres (≥50 µm in diameter) was counted manually.

### Migration/invasion assay
Cell migration and invasion assays were performed using the 24-well control chamber and 24-well Matrigel invasion chamber according to the manufacturer's instructions (BD Biosciences, San Jose, CA). Cells were seeded at a density of $2.5 \times 10^4$/chamber with a DMEM medium. DMEM medium with 10% FBS was used as a chemoattractant. About 18 h after seeding, migrating, and invading cells were fixed and stained with a HEMA-3 stain set (Fisher Scientific). The experiments were performed in triplicate.

### Flow cytometry
Cells were harvested with trypsin and washed with PBS. $2.5 \times 10^5$ cells were resuspended in 400 µL PBS. Antibodies against CD44 (FITC, #555478, BD Pharmingen) and CD24 (PE, #555428, BD Pharmingen) were added at 1:200 dilution for 20 mins on ice. Unstained and single stain (CD44 or CD24 alone) samples were generated for compensation. Samples were spun down and washed three times with PBS. Samples were then analyzed using a BDLSRII (BD Biosciences) with BD FACS Diva 4.0 software. DAPI was added for live/dead analysis. Living singlet cells were utilized for CD44/CD24 population analysis. Flow data were analyzed on FlowJo v10 software.

### Tumor xenograft studies
All animal handling and procedures were approved by Wayne State University Institutional Animal Care and Use Committee (IACUC, 19-02-0971). The maximal tumor burden permitted by IACUC is 10% of mouse weight at the endpoint. To measure the tumor progression of wild-type and mutant (A129S or I132S) FOXQ1, we first generated these various FOXQ1 stable expression cell models based on HMLER cells (obtained from Dr. Robert A. Weinberg) by a lentivirus infection approach. The derivative HMLER cells were injected ($2 \times 10^6$ cells/0.2 mL 50:50 matrigel/PBS) into the mammary fat pads of female NSG (NOD.Cg-Prkdc Il2rg /SzJ) mice (8 mice per group). To study the effect

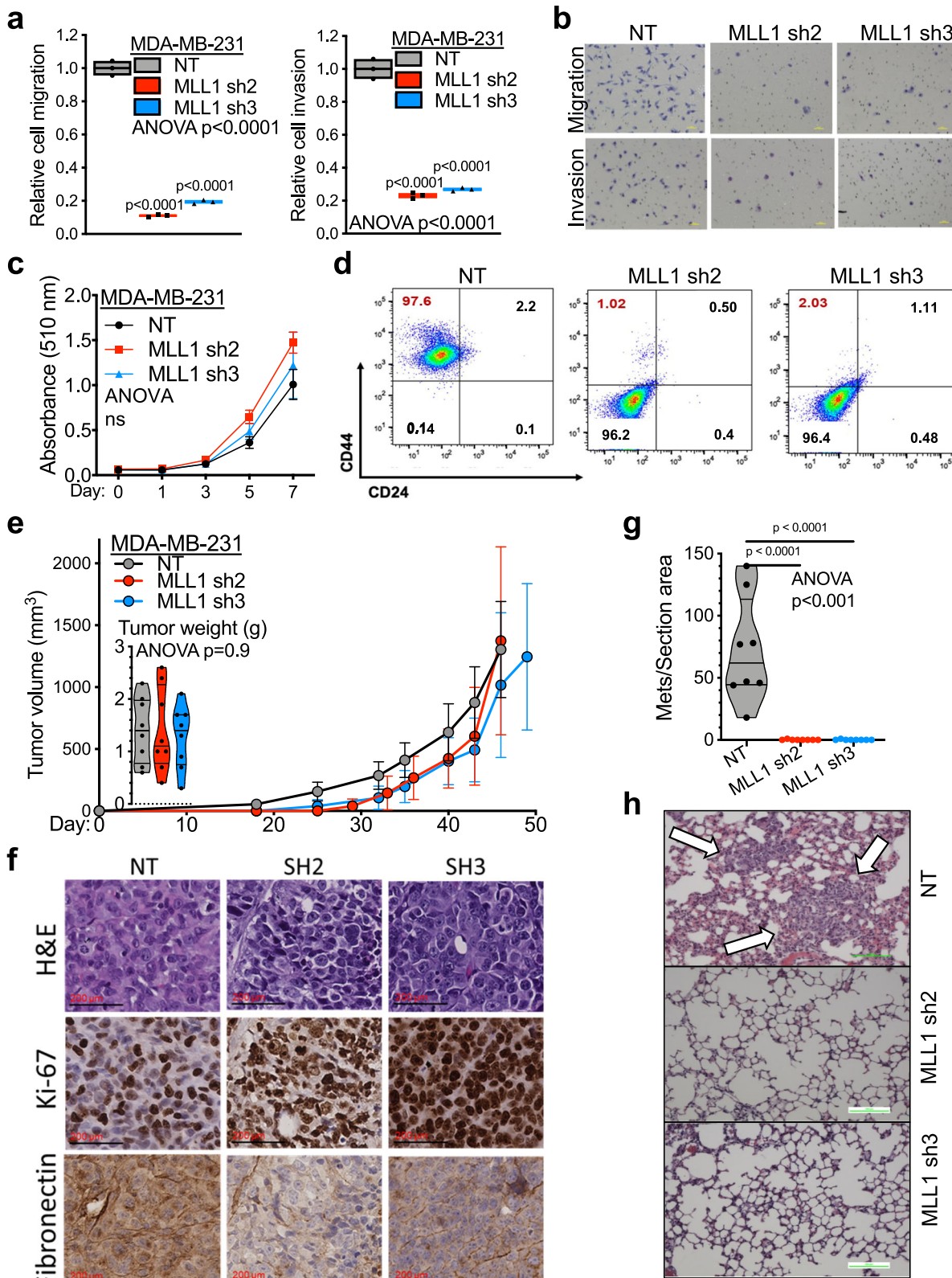

of MLL1 on tumor metastasis, MDA-MB231 cells ($2.5 \times 10^5$ cells/100 μl PBS) with NT control and MLL1 shRNAs were implanted in mammary fat pads of female NSG mice (8 mice per group). All NSG mice are purchased from Jackson Labs and are seven weeks old upon receiving.

Tumor formation and growth were monitored by biweekly palpitation, and tumor volume was estimated by length × width². Once largest tumor in one group reached an approximate weight of 2.5 g, the

group of mice were sacrificed, and the primary tumors and whole lungs were harvested. Paraffin blocks for primary tumors and lungs were made immediately for further H&E and IHC analysis. IHC lung sections were visualized and manually counted under the microscope to quantify metastatic lesions in the lungs. Slide images were taken, and the total area of the lung section analyzed was estimated by measuring the surface area with ImageJ software (v1.51). The number of

**Fig. 8 | Targeting MLL1 inhibits cancer cell motility in vitro and tumor metastasis progression in vivo. a** The effects of MLL1 silencing on cell migration and invasion in MDA-MB-231 cells. The migration (left panel) and invasion (right panel) were assessed with Boyden chamber assay ($n = 3$). Migrated or invaded cells were quantified and normalized to the mean of the NT control group. The floating bar graph depicts the sample mean, maximum and minimum values. Values from each replicate are indicated as dots on the plot. Statistical significance was assessed using a one-way ANOVA with Bonferroni multiple comparisons adjustment ($p < 0.0001$ for both panels). **b** Images of cell migration (top panels) and invasion (low panels) for MDA-MB231 cells with and without MLL1 knockdown. Scale bar: 100 μm. **c** Cell proliferation of MDA-MB-231 cells with NT control and MLL1 knockdown was measured by SBC assay ($n = 6$ per group). The line graph depicts the mean ± SD. Data were analyzed as panel a (No statistical significance was detected). **d** FACS measurement of CD44 + high/CD24-low cell population in MDA-

MB231 cells with and without MLL1 knockdown. **e** Results of tumor formation and growth of MDA-MB-231 NT and MDA-MB-231 shMLL1 cells in NSG mice ($n = 8$ per group). Tumor growth was analyzed using one-way ANOVA with Bonferroni correction for multiple comparisons ($p = 0.9$). The average tumor weights at the endpoints are also shown inside the panel as a violin plot and display points for values from each animal. **f** IHC staining of the primary tumors from the engraftment of MDA-MB-231 NT and MDA-MB-231 shMLL1 cells. Representative image, $n = 5$ for each tumor. Scale bar: 100 μm. **g** Quantification of metastatic lesions in two groups of tumors derived MDA-MB-231 NT or MDA-MB-231 shMLL1 cells implanted in NSG mice ($n = 8$ per group). Average metastatic lesion numbers from five serial sections in each tumor were normalized to the total surface area of the lung section. Data were analyzed as panel a ($p < 0.001$). **h** Representative H&E staining pictures of the lung metastatic lesions. Arrows point to metastatic lesions in the lungs. Scale bar: 100 μm. For all panels, source data were provided in Source data 1.

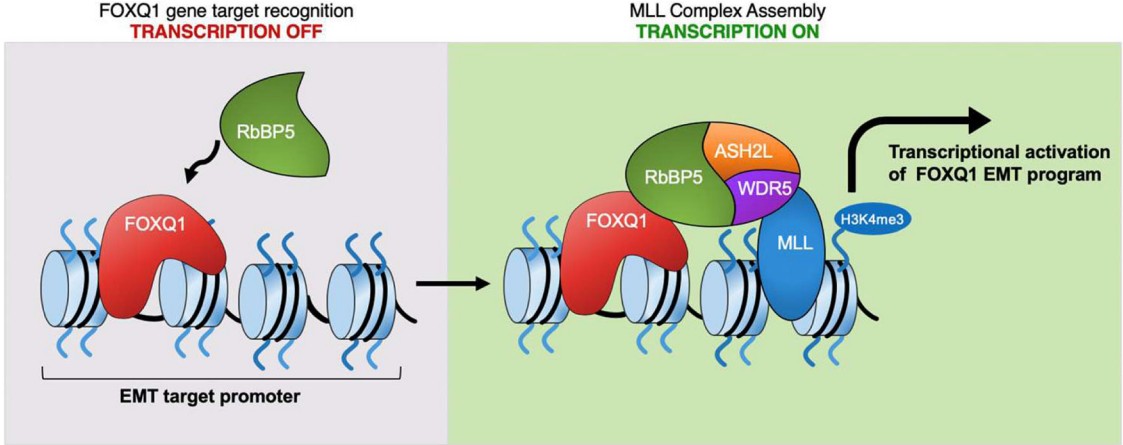

**Fig. 9 | Schematic illustration of the mechanism of FOXQ1 transcriptional regulation.** A proposed model illustrates that FOXQ1 directly recruits RbBP5 to the promoters of EMT target genes to facilitate MLL complex assembly, H3K4me3, and transcriptional activation.

micrometastatic lesions observed was normalized to the total surface area of the lung section analyzed. DNA was isolated from one representative primary tumor, and the human FOXQ1 gene was isolated and purified by PCR to confirm the group identity by Sanger sequencing.

### H&E staining and immunohistochemistry assay (IHC)

Paraffin sections were de-waxed in a xylene-ethanol series. For IHC, endogenous peroxides were removed by a TBS/3% hydrogen peroxide incubation at room temperature for 30 min. HIER antigen retrieval was done with citrate buffer (pH 6) and the BIOCARE Decloaking Chamber. Slides were blocked with Super Block Blocking buffer (Thermo Scientific) for 40 mins at room temperature. Anti-V5 (Abcam ab27671, 1:50), anti-Vimentin (Dako M0725, 1:30), Fibronectin (BD Transduction Laboratories, 610077, 1:50), and Ki67 (Cell signaling 9027, 1:200) were used to stain the sections, alongside an antibody-negative control. Secondary antibody incubation and detection were carried out using the GBI Labs DAB chromagen kit (GBI Labs, #D52-18). Sections were then dehydrated through ethanol to xylene washes and mounted with Permount. For H&E staining, slides were hydrated and stained with hematoxylin for one min. After rinsing, the slides were stained with eosin for one min, rinsed, and sealed with cover slips using Permount.

### Tandem affinity purification mass spectrometry

**TAP-TAG purification system.** Full-length wild-type human FOXQ1 gene was cloned into pcDNA3.1(C)-STAP vector. The resultant TAP-tag consists of streptavidin binding protein (SBP) and protein A, separated by a TEV protease site, fused to the C-terminus of FOXQ1. A construct expressing LacZ TAP-tag was used as the control.

HEK293T cells ($6 \times 10^7$) were seeded in 100 mm dishes (ten dishes each) and, 24 h later, transfected with 15 μg pcDNA3.1-FOXQ1-(C)-STAP constructs and vector control using Lipofectamine 2000 in a ratio of 1:2.5 (μg/μl), respectively. Cells were incubated at 37 °C for 6 h, washed with 1× PBS, and complete media was added for another 48 h. Cells were washed once with 1× PBS and suspended in lysis buffer (50 mM Tris-HCl pH 7.5, 125 mM NaCl, 0.2% NP-40, 5% glycerol, 1.5 mM MgCl$_2$, 25 mM NaF and 1 mM Na3VO4) containing protease/phosphatase inhibitors on ice for 5 min. The cell lysate was then sonicated on ice for 10 min. The cell lysate was centrifuged at 10,000 g at 40 °C for 15 min and the supernatant was collected for downstream processing. 30 μL IgG beads (Cat#A2909, Sigma) were added per mL of supernatant and rotated at 4 °C for 2 h. The IgG beads were then spun down and combined into one tube. Beads were washed three times with lysis buffer and once with SBP Buffer (10 mM Tris-HCL pH 7.5, 100 mM NaCl and 0.2 NP-40). For each bead sample, 100 μL SBP buffer and 1.5 μL AcTEV protease (Invitrogen, cat# 12575-015) were added and incubated at room temperature for 40 min and further at 16 °C overnight with rotation. 400 μL SBP buffer was added to the samples, and beads were pelleted by centrifugation. The supernatant was transferred to equilibrated streptavidin beads (Thermo Fischer, Cat # 20359) and incubated at 4 °C for 4 h. Streptavidin beads were spun down and washed four times with 1 mL SBP buffer. The SBP buffer was removed after washing, and 40 μl 2× SDS sample buffer was added. Samples were boiled for 6 mins at 100 °C. The denatured eluate was size-separated by precast gradient gel and visualized by Colloidal Blue Staining Kit (Invitrogen, Cat # LC6025), according to the manufacturer's protocol. The stained gel was cut into pieces based on protein size, with five slices per lane

**Fusion mass spectrometry analysis for protein identification.** Protein identification was performed by the proteomics core facility of Wayne State University. The gel pieces were first washed with water and 25 mM $NH_4HCO_3$, 50% ACN for 15 min each. The liquid was removed, and the gel pieces were dehydrated in 100% ACN for 5 min. Then, the gel pieces were rehydrated in 50 mM $NH_4HCO_3$ for 5 min, followed by incubation in an equal volume of 100% ACN for 15 min. All liquid was removed, and the gel pieces were dehydrated once again in 100% ACN for 5 min. The gel pieces were then speed vacuumed dry for 5 min. The following was then performed: reduction with five mM DTT, 50 mM $NH_4HCO_3$; alkylation with 15 mM IAA, 50 mM $NH_4HCO_3$; and overnight digestion with sequencing-grade trypsin (Promega) in 25 mM $NH_4HCO_3$, 10% ACN. Following digestion, peptides were extracted from the gel plugs using 50% ACN and 0.05% FA. The free peptides were then speed-vacuumed to dryness and solubilized in 0.1% FA. The peptides were separated by reverse-phase chromatography (Acclaim PepMap100 C18 column, Thermo Scientific), followed by ionization with the Nanospray Flex Ion Source (Thermo Scientific), and introduced into Orbitrap Fusion™ Tribrid mass spectrometer (Thermo Scientific). Abundant species were fragmented with high-energy collision-induced dissociation (HCID). Data analysis was performed using Proteome Discoverer 1.4 (Thermo), which incorporated the Mascot (Matrix Science) and Sequest algorithms (Thermo Fisher). The Uniprot_Hum_Compl_20150826 database (https://www.uniprot.org/uniparc?query=(dbid:20150826)) was searched for human protein sequences, and a reverse decoy protein database was run simultaneously for false discovery rate (FDR) determination. Secondary analysis was performed using Scaffold 4.4.5 (Proteome Software). The minimum protein identification probability was set at two unique peptides with ≤1.0% FDR. Mascot, Sequest, and X! Tandem were searched with a fragment ion mass tolerance of 0.6 Da and a parent ion tolerance of 10 PPM. Carbamidomethylation of cysteine was specified in Mascot, Sequest, and X! Tandem as a fixed modification. Deamidation of asparagine and glutamine and oxidation of methionine were specified in Mascot & Sequest as variable modifications. Glu→pyro-Glu of the N-terminus, ammonia-loss of the N-terminus, Gln→pyro-Glu of the N-terminus, deamidation of asparagine and glutamine, and oxidation of methionine were specified in X! Tandem as variable modifications. A total of 510 proteins were identified from over 2500 spectra. The resulting proteins identified in the FOXQ1 TAP samples and LacZ negative control were compared based on the number of peptides mapped to the protein ID, normalized to the overall protein molecular weight by Fischer's exact *T*-test

**Network analysis.** The top 100 most significantly identified FOXQ1-interacting proteins (Fischer's Exact) were subject to network analysis (STRING v11.0) to identify protein complexes. The resulting network was visualized by Cytoscape v.3.7.1.

## ChIP-seq

**Chromatin preparation, immunoprecipitation, and DNA purification.** HMLE/FOXQ1 cells were grown to ~70–80% confluency, and $10^7$ cells were harvested and fixed in 1% formaldehyde at a density of $2 \times 10^6$ cells/mL for 10 min at room temperature with agitation. Fixation was clenched with ice-cold glycine to a final concentration of 0.125 M and agitated at room temperature for 5 min. Pellet was washed three times with PBS. Fixed cells were resuspended in sonication buffer (10 mM Tris pH 8, 1 mM EDTA, 1× protease inhibitors). SDS was added to a concentration of 0.1% for H3K4me3 ChIP samples. No SDS was added for FOXQ1 and RbBP5 ChIP samples prior to sonication. Chromatin was sonicated using the Qsonica ultrasonic processor for 24 pulses (30 s ON, 1 min OFF, AMP 40) on wet ice. The Sonication pattern was visualized by gel electrophoresis to validate a chromatin smear around 1 kb–200 bp in size. Triton X-100 and sodium deoxycholate were added to sonicated

samples to a final concentration of 0.1% each. SDS was also added to FOXQ1 and RbBP5 samples to a final concentration of 0.1%. Samples were mixed well and centrifuged at $14,000 \times g$ for 10 min at 4 °C. Antibodies [4 µg H3K4me3 (Millipore), 4 µg RbBP5 (Bethyl) and 5 µg V5 (invitrogen)] were conjugated with 40 µg Protein G Dynabeads (Invitrogen) for 1 h at room temperature. Beads were washed in 1× PBS to remove unconjugated IgGs. The sheared chromatin ($4 \times 10^6$ cells per ChIP) was allowed to incubate with antibody-conjugated beads overnight at 4 °C with rotation. Beads were washed for 10 min at 4 °C with the following steps: twice with 1 mL RIPA (TE pH 8, 0.1% SDS, 0.1% sodium deoxycholate, 1% Triton X-100), twice 1 mL RIPA with 0.3 M NaCl, twice with LiCl buffer (0.25 M LiCl, 0.5% NP-40, 0.5% sodium deoxycholate), once TE with 0.2% Triton X-100, and once with TE buffer. Beads were resuspended in 100 µL 1X TE and 3 µL of 10% SDS, and 5 µL 20 mg/mL proteinase K was added. Samples were incubated overnight in a 65 °C water bath. The supernatant was collected, and beads were washed with 100 µL 1X TE with 0.5 M NaCl by vortexing. The two supernatants were combined. DNA was precipitated by phenol: chloroform extraction followed by 150 mM NaOAc and ethanol precipitation for 2 h at −80 °C incubation. DNA pellet was collected by max centrifugation at 4 °C for 30 mins. Pellet was washed with 70% ethanol and allowed to air dry. DNA pellet was resuspended in 40 µL 1X TE (50 µL ddH₂O for ChIP-qPCR). ChIP DNA was then subject to either library preparation for sequencing or used directly for qPCR. DNA isolates from two-independent replicate ChIP experiments and corresponding input chromatin DNA were used for downstream library preparation and sequencing.

**ChIP library preparation and sequencing.** DNA ends were repaired by the Epicentre End-IT DNA End-Repair Kit (Lucigen, ER0720) to generate blunt-ended DNA according to the manufacturer's protocol. Briefly, reaction set up with 34 µL DNA, 5 µL 10X end-repair buffer (300 mM Tris-acetate pH 7.8, 660 mM potassium acetate, 100 mM magnesium acetate, 5 mM DTT), 5 µL 2.5 mM dNTPs, 5 µL 10 mM ATP, 1 µL End-Repair enzyme mix (T4 DNA polymerase, T4 polynucleotide kinase). End-repair was incubated at room temperature for 45 min. The resulting DNA was purified by the Qiagen MinElute Reaction Cleanup Kit and eluted in 32 µL EB buffer. To add "A" overhang to the 3′ ends: 30 µL of the above-purified DNA, 5 µL 10X NEB buffer #2, 1 µL dATP (10 mM stock), 11 µL water, 3 µL 5 U/µL Klenow fragment (3′ → 5′ exo-) was incubated at 37 °C for 30 min. The DNA was then purified by the Qiagen MinElute Reaction Cleanup Kit and eluted in 25 µL EB buffer. Linked ligation was carried out as follows: 23 µL of the above DNA, 3 µL 10X T4 DNA ligase buffer, 1 µL Illumina Index adapter oligo mix, and 3 µL T4 DNA ligase. The adapter reaction was incubated for 30 min at room temperature. Size selection was carried out using a 2% E-Gel EX gel. DNA was run on the precast gel for 10 min, and DNA around 250–450 bp was excised. The DNA was purified using the Qiagen MinElute Gel Extraction Kit and eluted in 20 µL EB buffer. For PCR, 12 µL DNA, 12 µL molecular grade water, 25 µL master mix (2X Phusion HF, Finnzymes), 1 µL PCR primer (InPE 1.0, diluted 1:2), 1 µL PCR primer (InPE 2.0, diluted 1:2), 1 µL PCR primer Index #1-8 (diluted 1:2) were combined. PCR was cycled under the following conditions: initial denature at 98 °C for 30 s, 18 cycles of 98 °C for 10 s, 65 °C for 30 s, 72 °C for 30 s, and final elongation at 72 °C for 5 min. Size selection of the library was performed on 2% E-Gel EX gel. DNA was run on the precast gel for 10 min, and DNA around 250–450 bp was excised. The DNA was purified using the Qiagen MinElute Gel Extraction Kit and eluted in 20 µL EB buffer. Samples were sequenced on the Illumina HiSeq400 platform, with 50 bp reads single-end, according to the manufacturer's protocol.

**ChIP data analysis.** ChIP-seq FASTQ reads were mapped to the hg19 reference genome using Bowtie2 (Galaxy Version 2.3.2.2). Enriched

ChIP peaks were called using MACS2 (Galaxy Version 2.1.1.20160309.0) with the mapped input DNA reads as the background control. FDR (*q*-value) was set to 0.001. Peaks that were detected in both the replicate ChIP samples were utilized for downstream analysis. ChIP-seq peaks were analyzed by the Cis-regulation element annotation system (CEAS, v1.0) to obtain the genomic annotation for the chromatin regions (Peak BED intervals) bound by FOXQ1 and RbBP5 relative to the whole genome (the expected control values). FOXQ1 and RbBP5 co-bound regions were determined by intersecting the BED intervals of the peak files. Regions that had ≥1 bp overlap was considered co-bound regions. To analyze DNA binding motif enrichment, the top 50-scoring FOXQ1 summits (100 bp region) that fell within FOXQ1-RbBP5 overlapping regions were submitted to the MEME Suite (v5.4.4. meme-Suite.org).

### RNA-seq

**RNA isolation from cells.** HMLE/FOXQ1 and HMLE/LacZ cells were plated in biological duplicates at the same time and cultured for 48–72 h. When they reached 80% confluency, the cells were collected and processed for RNA extraction using RNeasy Plus Mini Kit (QIA-GEN). For each sample, 2 µg extracted RNA with a 260/280 above 2.0 were processed for library construction and sequencing.

**Library preparation, sequencing, and data processing.** Library preparation and sequencing, and data processing were performed at LC Sciences (Houston, TX). Succinctly, the RNA library was prepared (including Poly (A) selection, fragmentation, Adapter attachment, reverse transcription, PCR amplification, and Library size selection) using the TruSeq Stranded mRNA kit (Illumina) according to the manufacturer's protocol. Sequencing was performed using the Illumina HiSeq 2000 platform with 100 bp paired-end reads on-average min and of about 40–50 million reads per sample. Paired-end reads were mapped to the hg19 human genome using Bowtie2 v2.2.9. The abundance was estimated using RSEM, and genes with <10 counts were filtered. The differential expression analysis was done using EdgeR v3.12.1 in the Bioconductor package. The corresponding promoter regions (<10 kb upstream, <1 kb downstream from TSS) to the differentially expressed genes were selected for integrated analysis with ChIP-seq data. Promoter-bound regions were determined by intersecting ChIP-seq peak BED files with the promoter regions. Regions that had ≥1 bp overlap were considered factor-bound promoters.

### TCGA analysis

Normalized mRNA expression data for the 622 co-bound and differentially expressed genes from the provisional TCGA breast cancer cohort (*n* = 1093) were obtained from cBioPortal in the format of z-scores. A correlation matrix was generated using the R package, *ppcor*. The subset of genes with a Spearman correlation > 0.2 with FOXQ1 was selected and used to create a heatmap with the *corrplot* package (v 0.84).

### Survival analyses

Survival analysis was conducted with the *survival* package in R. Univariate Cox proportional hazards models were generated to test the significance of gene expression and clinical characteristics to test the significance of each variable in predicting overall survival. Subsequently, multivariate Cox proportional hazards models included patient stage, age at diagnosis, and estrogen receptor status, as these were all significant independent clinical predictors of overall survival. Forest plots (multivariate cox regression) of the association of gene expression with overall survival for eight FOXQ1-RbBP5 regulated targets were generated with ggplot2 in R.

The dots represent the hazard ratio, and the tails depict the 95% confidence interval.

### Functional enrichment analysis

Gene sets from high throughput sequencing experiments were subject to functional annotation using Enrichr (mayan.cloud/Enrichr). Input gene sets were compared to the annotated gene sets by Fisher's Exact *T*-test with *p*-value adjustment by the Benjamini-Hochberg method for correction for multiple hypotheses testing.

### Statistical analysis

The description of the statistical tests performed for each experiment can be found in the corresponding figure legend. The exact *p*-values and detailed statistical summaries can be found in the Source Data File. All bar graphs depict the sample mean ± standard deviation (SD). All floating bar graphs (box plots) represent the sample mean, minimum and maximum values. Dots on the bar and floating bar graphs represent the values from an individual replicate of the experiment.

Unless otherwise indicated, in experiments comparing one control group to multiple variable groups (>2 groups), a one-way ANOVA test was performed with the Bonferroni correction method for multiple comparisons concerning the control group. The ANOVA summary p-value is reported on the graph, and the stars above the sample data indicate the multiple comparisons adjusted *p*-value. For qPCR gene expression and ChIP data, each gene target/genomic locus was treated as an independent variable, and an independent, two-tailed *t*-test was performed with Bonferroni correction for the multiple comparisons to the control group. For cell proliferation and tumor growth experiments, data were analyzed by repeated measures ANOVA and mixed model analysis with Bonferroni correction for multiple comparisons to the control group. The results were graphed with a line graph depicting the sample mean absorbance ± SD over time. Cell viability data for treatment with MLL inhibitor (OICR-9429) was analyzed by normalizing data to the corresponding DMSO mock control. Line graphs depict the mean relative viability ± SD and the data were analyzed by two-way ANOVA with Bonferroni correction for multiple comparisons. Data processing and statistical tests for high-throughput assays can be found in the respective methods sections. For all graphs, *p < 0.05, **p < 0.01, ***p < 0.001. Qualitative experiments were performed in at least technical triplicate, and a representative image is shown. Detailed statistical results can be found in the Source Data File.

### Reporting summary

Further information on research design is available in the Nature Research Reporting Summary linked to this article.

## Data availability

All data are available within the Article, Supplementary Information, Source Data files or public repositories. Source data are provided as a Source Data file. The Uniprot_Hum_Compl_20150826 database (https://www.uniprot.org/uniparc?query=(dbid:20150826)) was searched for human protein sequences in this study. The RNA-sequencing and ChIP-sequencing data in this study have been deposited into the National Center for Biotechnology Information (NCBI) Gene Expression Omnibus (GEO) database with the accession code GSE141293. The raw spectral counts for protein identification and abundance are in Supplementary Data 1. A reporting summary for this article is available as a Supplementary Information file.

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

## Acknowledgements
We thank Dr. Robert A. Weinberg at MIT for providing the HMLE cell line. This work was supported in part by the NIH RO1 grant CA172480-01A1 (G. Wu); MT program pilot grant (25RU21) from the Karmanos Cancer Institute, and grant boost (I17768) from OVPR Wayne State University (G Wu); NIH training grant T32CA9009531 (A Mitchell and CJ Block); NCI individual predoctoral fellowship F31CA236245 (A.M.); and NCI R21 CA256423 (X.Z.). The Biostatistics Core, the Microscopy, Imaging, and Cytometry Resources Core, and the Biorepository Core of the Karmanos Cancer Institute are supported by NIH Center grant P30-CA022453-29. The Microscopy, Imaging, and Cytometry Resources Core are also supported, in part, by the Perinatology Research Branch of the National Institutes of Child Health and Development at Wayne State University. The Wayne State University Proteomics Core, and this work, are also supported through the NIH Center Grant P30 ES 020957, the NIH Cancer Center Support Grant P30 CA 022453, and the NIH Shared Instrumentation Grant S10 OD 010700.

## Author contributions
A.M., L.P., L.M., B.K., Z.Y., Y.Do, and G.W. conceived and designed all experiments. G.W. and Y.Da conducted TAP-TAG purification for proteomics. G.W. and L.W. prepared RNA for RNA sequencing and mutagenesis. Y.Do prepared the purified protein for the MLL core complex subunit. S.Z. and C.W. are responsible for the generation of FOXQ1 antibody. M.Z. and X.Z. help optimize methods for GST-protein purification and Co-IP assay. J.H. and H.G. performed IHC. A.M. and B.K. performed ChIP library preparation experiments. A.M., J.B., D.C., W.C., Y.Z., and B.Z. performed all the statistical and bioinformatics analyses in this study. A.M. performed RT-PCR, Western blot, immunofluorescence analysis, and other experiments. A.M., L.W., J.B., L.P., L.M., B.K., Z.Y., Y.Do, and G.W. discussed the data. G.W. and A.M. supervised and wrote the final draft of the manuscript.

## Competing interests
The authors declare no competing interests.
