## [Peer Review File · Nature Communications]

FOXQ1 recruits the MLL complex to activate transcription of EMT and promote breast cancer metastasisREVIEWER COMMENTS

Reviewer #1 (Remarks to the Author):

Mitchell and colleagues present a new exciting paper that analyses the interactome of the transcription factor FOXQ1. A mass spectrometry-based screen identified components of the MLL histone methyltransferase complex as previously unknown interactors of FOXQ1. Among these proteins, direct interaction with RbBP5 is demonstrated and analysed in depth. The biochemical work analysing the FOXQ1 interaction with RbBP5 is extensive and in most parts convincing. Biologically, the complex between FOXQ1 and MLL, among other functions, is shown to regulate the epithelial-mesenchymal transition in breast epithelial and breast cancer cells. This biological part of the study is thorough and extensive, yet leaves several open questions in terms of the selectivity or specificity of the mechanism. The latter part, although well documented, leaves this reviewer with certain major questions being unanswered. The ambiguity I refer to is reflected even in the title of this paper, whereby the term "FOXQ1 hijacks the MLL complex to activate the EMT program" is not clear. What are the specific biological conditions that mediate this type of "hijacking"? Do the authors mean overexpression of FOXQ1 after transfection? Furthermore, the contribution of the FOXQ1-MLL complex to the regulation of the EMT program is not discussed at all in the otherwise very nice discussion of the paper that focuses on the biochemical/mechanistic findings, which are more robust.

The paper presents a plethora of experiments and makes use of many complementary controls, including rescue experiments that is very nice. My major concern is the selective mechanism by which FOXQ1 regulates the EMT. Multiple transcriptomic experiments are provided in this paper, and large numbers of genes are shown to be regulated by FOXQ1 and its interaction with RbBP5. Among these many genes, few belong to the so-called EMT program, and they are followed up, which is very nice. It would have been useful if the authors had provided some discussion on all the other FOXQ1 target genes, and their co-regulation by the RbBP5-MLL activity.

This brings to my major comment: if FOXQ1-RbBP5-MLL regulate a large number of genes in epithelial cells, what are the biological conditions that drive the function of FOXQ1 as an EMT inducer? Are there pathways or biological conditions during cancer development that turn on expression or activate the transcriptional activity of FOXQ1? This is relevant as the study uses cell models in which immortalized mammary epithelial cells HMLE overexpress FOXQ1, and then this cell model is further manipulated to downregulate its RbBP5 or replace wild type FOXQ1 with mutants that fail to form the complex with RbBP5. An important set of experiments addressing my concern, focus on the breast cancer cell model MDA-MB-231, where endogenous FOXQ1 is silenced by shRNA and then wild type or mutant forms of FOXQ1 are co-transfected as rescuers. The paper presents these experiments briefly and emphasizes the effects of rescuing, overexpressed forms of FOXQ1 (wild type or mutant). Is the silencing of endogenous FOXQ1 in MDA-MB-231 cells turning these cells more epithelial, counteracting their mesenchymal identity? Data show that the migration of these cells is affected. Does such knockdown of endogenous FOXQ1 affect primarily the expression of EMT-like genes, or also the other genes identified in the HMLE-FOXQ1 overexpression transcriptomic screens?

As a second major comment, I try to understand how the FOXQ1-MLL complex regulates the EMT program. Clearly FOXQ1 is directed to its target genes via its binding domain. And the paper proves convincingly that the transcriptional activity of FOXQ1 depends on RbBP5 and MLL activity. Since the EMT field is very rich in many studies that have analysed gene expression and chromatin modification across a variety of breast cancer cell models but even in breast cancer patient material, is histone-3K4me3 a modification frequently found in the promoters of EMT genes in these other models and more importantly in patient samples? If yes, can the authors identify a correlation between the histone-3K4me3 profile of the EMT genes and FOXQ1, RbBP5 mRNA expression? Such evidence would strengthen the case that this paper attempts to make, regarding the functional role of FOXQ1-MLL and histone-3K4me3 in EMT and these findings would be nice to discuss also in the discussion of the paper.

This reviewer also provides some specific comments that could aim at clarifying or enhancing the accuracy of the data.

Specific (and minor) comments:

1. Fig. 1 and Suppl. Fig. 1: specificity of the FOXQ1 antibody is shown after overexpression of Flag-FOXQ1 and Flag-FOXC2. The antibody should also be tested in cells (e.g. MDA-231) where endogenous FOXQ1 is silenced. Does it cross-react with any other protein under such conditions? This is important because Fig. 1d-f show the endogenous co-IP of FOXQ1 and RbBP5. In some of these cells, endogenous FOXQ1 is undetectable at 10% input cell lysate, yet a strong protein band is detected upon immunoprecipitation of 90% of the lysate with the RbBP5 antibody. Since molecular size markers are missing (please add those in every immunoblot), and since the intense, sharp band of the co-IP does not reflect any of the very faint and fuzzy bands of the 10% input, the sh/siRNA experiment is necessary to prove that the FOXQ1 protein bands that co-IPs is actually FOXQ1. This is further important to cross-verify for the following reason: if we look at the data of Fig. 1c, where recombinant proteins are used we see that the direct interaction is very weak: compare the signal of 1% input to the signal after pull down of 99% protein. This signal of interaction is significantly enhanced upon endogenous co-IP (Fig. 1d-f). This means that either in cells, the proteins get post-translationally modified and their interaction is enhanced, or that cells provide an ancillary extra protein that enhances or stabilizes the interaction, or finally that the signal received in co-IP is not correct. For this reason, the requested specificity controls are important.
2. Fig. 3d, f: if we look at gene by gene presence of the H3K4me3 mark, it is obvious that for some genes the RbBP5 shRNA (e.g. ZEB1, FOXC2) shows stronger effect and for some other the FOXQ1 shRNA (e.g. SIX2). This can be discussed in more detail, as it shows that the impact of silencing one or the other protein of the same complex is not functionally equivalent.
3. All experiments with shRNA knockdown need to show the degree of silencing of the respective protein (Fig. 3, FOXQ1, RbBP5).
4. The MDA-MB-231 mammosphere morphology is rather inconsistent between experiments: compare the NT cells in Suppl. Fig. 6h, 6j and Suppl. Fig. 7f: aggregates in 6h, 7f, acinar-like organoids in 6j. Why is it so? Are the culture conditions the same in all mammosphere experiments?
5. Fig. 8b, c: it would be more appropriate if the three images (V5, vimentin, H&E, Fig. 8b) and (V5, H&E, Fig. 8c) were from serial sections so that the same area of the tumor was stained in order to make sure that the same tumor cells are being analysed.

Reviewer #2 (Remarks to the Author):

Authors of the manuscript provide a mechanistic study which shows FOXQ1's direct interaction with RbBP5, a core subunit of KMT2/MLL H3K4 methyltransferase complexes, follow by the recruitment of the KMT2/MLL complex to EMT promoters, which in turn facilitates the deposition of H3K4 tri-methylation (H3K4me3) within the target promoters and activates EMT gene expression programs. They found that FOXQ1 binds to RbBP5 through its Forkhead domain and further identified key amino acid residues for this interaction. Authors then utilized genetic manipulations and pharmacologic approaches to elucidate the molecular mechanism where the disruption of FOXQ1-RbBP5 interaction (OICR-9429 treatment) or the failure of the KMT2/MLL complex recruitment (FOXQ1-I132S/A129S mutants) attenuates FOXQ1-dependent gene expression, EMT, and cancer progression. Overall, the manuscript is well written, organized, and focused on an interesting topic with data presented support for authors' claims; however, few concerns remain and need to be addressed, thus, more studies and data would be required for publication in Nature Communications.

General comments:

1. Given that KMT2/MLL family members share the core subunits RbBP5, ASH2L and WDR5, and they play distinct or redundant roles in different cancer contexts, a major problem is that the present study failed to identify the specific KMT2 member(s) that contribute(s) to FOXQ1 transactivation activity in breast cancer. I wonder if the authors identified any KMT2 members and/or unique subunits of the KMT2/MLL complexes in the TAP-MS analysis. Specific knockdown experiments for individual members of the KMT2/MLL family, such as H3K4 trimethyltransferases KMT2A, 2B, 2F, and 2G, would need to be performed in the HMLE/FOXQ1 and/or MDA-MB-231 cells and check for the effects of specific KMT2 loss on EMT promoter-associated H3K4me3. Although KMT2C and KMT2D function as H3K4 monomethyltransferases and mainly exert their

function in enhancer regulation, they could also contribute to the regulation in promoter-associated H3K4me3 (Lee et al. PNAS 2008, Dhar et al. Genes & Development 2012, Kim et al. Cancer Research 2014). Thus, the effects of KMT2C/2D loss on EMT promoter-associated H3K4me3 would be worth to be checked as well.

2. As RbBP5 could also be shared by the KMT2C/2D complexes that catalyze H3K4 monomethylation on enhancer regions as well as facilitate enhancer activation, it is unclear whether the major role of FOXQ1 in regulating the activation of EMT transcription programs is exclusively through the promoter-associated H3K4me3. The ChIP-seq results of FOXQ1 and RbBP5 (Figure 2) also imply that the major occupancies of them both are not in the proximal promoter regions. Instead, the majority of FOXQ1 and RbBP5 seem to enrich in the distal intergenic and intron regions, where often contain enhancer DNA sequences. Thus, the authors should emphasize more clearly on the rationale to focus on the proximal promoter regions for the role of the FOXQ1-RbBP5-KMT2 axis in EMT gene regulation. Additional ChIP-Seq data for H3K4me1 may provide better distinctions between the FOXQ1 and RbBP5-targeted regions (whether promoters only or enhancers as well), and more clues for identifying the specific KMT2/MLL family member involved (mono- or trimethyltransferase).

3. The introduction is too short. There is a need to have a general introduction of the KMT2/MLL family of H3K4 methyltransferase complexes, especially on how they contribute to H3K4 methylation and target gene transcription.

Specific comments:

1. Figure 2-- The display for the genomic distribution and colocalization of FOXQ1 and RbBP5 is not clear. In lines 118-122, the authors mentioned "our data indicate that FOXQ1 displays equivalent enrichment within proximal promoter region (<10 kb upstream) and distal intergenic regions (8% enrichment each for 13,513 peaks) (Figure 2A). RbBP5 binding peaks were enriched within proximal promoter regions in HMLE/FOXQ1 cells (10.8% of 25,866 peaks), and a fraction were enriched in distal intergenic regions (<2%)(Figure 2A)". However, Figure 2A showed that FOXQ1 and RbBP5 occupied 58.5% and 51.9%, respectively, of distal intergenic regions. If it is true that FOXQ1 displays equivalent enrichment within proximal promoter region (<10 kb upstream) and distal intergenic regions (8% enrichment each for 13,513 peaks), with the enrichment of FOXQ1 in proximal promoter and intergenic regions only occupying 16% of the total FOXQ1 peaks, where would be the major distribution of FOXQ1 in the whole genome? Similarly, what is the percentage of FOXQ1-RbBP5-H3K4me3 co-targeted regions (1,397 peaks) that are within the promoter regions? A critical issue to address is how important the promoter-associated FOXQ1 function is in the scenario of EMT and breast cancer progression.

2. Figure 2C -- It is noticed that a significant number of FOXQ1 and RbBP5 co-targeted proximal promoters was covered by broad H3K4me3. Essentially, the major enrichment of FOXQ1 and RbBP5 within proximal promoter regions were broad and away from the target promoters (>1000 bp, ≤10,000 bp) (Figure 2A). Since broad H3K4me3 domains have been recently associated with maintenance of cell identity and cancer (Chen et al. Nature Genetics 2015, Cao et al. Scientific Reports 2017, Thibodeau et al. Scientific Reports 2017, Dhar et al. Molecular Cell 2018), it is unclear whether the FOXQ1-RbBP5-KMT2 axis is associated with the establishment of broad H3K4me3 domains in EMT-related genes. Additional discussion of this aspect in the manuscript is suggested. Furthermore, it would be better to indicate the genomic locations of qPCR primers for the target promoters in the study, such as SIX2, TWIST1, ZEB1, FOXC2 and so on.

3. Figure 5D and 5F -- Occupancies of RbBP5 and H3K4me3 in the FOXQ1/RbBP5 target promoter regions would need to be confirmed if the FOXQ1 mutants affect the recruitment of KMT2/MLL complex to these target promoters.

4. Figure 6E, 6J and 7E -- It is worth noting that RbBP5-knockdown or OICR-9429 treatment had modest effects on stem cell-like phenotype (CD44+/CD24-) when compared to the HMLE/FOXQ1 mutants. Please explain and discuss in the discussion section.

5. The representative pictures are missing for the morphological changes after knockdown of RbBP5 (Figure 6C), treatment of OICR-9429 (Figure 6H), introduction of FOXQ1 mutants (Figure 7A), and FOXQ1 manipulation in MDA-MB-231 cells (Figure 7F).

6. In lines 278-285, as the function of OICR-9429 is to disrupt the binding of RbBP5 to the KMT2/MLL complex, which should mimic the RbBP5 knockdown condition. It is not clear why the OICR-9429 treatment had effects on cell proliferation (Figure S6E) but RbBP5-knockdown showed no significant effects on cell proliferation (Figure S6B).

7. The basal level expression of FOXQ1 in HMLE/LacZ cells was missing. As this cell line serves as

a control for many experiments, this could impact many observable results. Thus, it is important to show the basal level of FOXQ1 in the parental and HMLE/LacZ cells.

8. The immunoblotting results for the knockdown of FOXQ1 and RbBP5, and overexpression of FOXQ1 variants in HMLE/FOXQ1 or MDA-MB-231 cells were missing. Moreover, it would need to be concerned if the knockdown of FOXQ1 affects the levels of RbBP5 in these cells.

Reviewer #3 (Remarks to the Author):

In this manuscript, Mitchel et al. investigated the requirement of FOXQ1 binding to RbBP5 subunit of KMT2/MLL histone methyltransferase to activate epithelial to mesenchymal transition (EMT) in breast cancer. The authors concluded that the disruption of RbBP5 binding to FOXQ1 causes decreased expression of EMT genes/proteins, inhibits the cell migration in vitro and prolongs survival of mice with xenograft tumors in vivo. While the in vitro mechanistic studies are interesting, most of the conclusions are limited to a one or two cell lines. The proposed molecular mechanism has very little relevance to triple-negative breast cancer in vivo. The in vivo orthotopic injection of the modified cell lines failed to demonstrate the convincing metastatic growth or changes in EMT that can support this mechanism. Clinical relevance of the conclusions to human breast cancer is also unclear.

Major points are outlined below.

Major concerns:

1. Although the main focus of the manuscript is the mechanisms of breast cancer metastasis, all the major conclusions are based on the data from a single mammary epithelial cell line with ectopic expression of FOXQ1 (HMLE/ FOXQ1). Mass spectrometry was done using embryonic kidney cells. The biological relevance of these two cell lines to the metastatic breast cancer is not clear.
2. All the conclusions are limited to analysis of cell lines, HMLE or a single breast cancer cell line MDA231, the latter of which was used to verify some of HMLE data. Authors should use transgenic mouse models of breast cancer to verify the results. Several transgenic mouse models of breast cancer are available and should be used in these studies.
3. The clinical relevance of the data to human breast cancer is unclear. Human tissue samples were not used in these studies. Is the proposed mechanism of FOXQ1-RbBP5 interaction important for all types of breast cancer or only for TNBC and why? For breast cancer patients, is there correlation between metastatic potential and FOXQ1/MLL expression?
4. All the main experiments are missing the HMLE/ LacZ control. What are the levels of FOXQ1 in the normal breast epithelial cells in culture and in vivo? How different is the ectopically expressed levels of FOXQ1 in cell lines compared to endogenous levels of FOXQ1 in human breast cancer tissues?
5. During assessment of migration/invasion of HMLE/FOXQ1 cells + shRbBP5, the authors did not include an important control of HMLE cells + shRbBP5.
6. Although, the authors intended to verify their data in vivo, their metastasis model is not very convincing. Based on histological images provided by the authors, the V5-antibody staining is non-specific and labels lung airway epithelial cells the most, making it difficult to quantify the metastases. To evaluate whether the distal metastases are developed in this model and to quantify them, the authors should label tumor cells with the reporter and track the development of the metastases with this reporter.
7. Some of the conclusions are not supported by the data. For example, using FOXQ1 ChIP-seq analysis, the authors state that FOXQ1 binds equally to proximal promoter regions (8%) and distal intergenic regions (Fig.2A). However the data in Fig.2A shows the majority of FOXQ1 binding peaks being in intron-distal intergenic regions (~90%). How the analysis was done is not explained in the manuscript. The other example is Fig.S5B, which uses the principle component analysis (PCA) for RNAseq. The authors state that overexpression of mutant FOXQ1 in HMLE/LacZ cells make the cells more similar to HMLE/LacZ than to HMLE/FOXQ1 cells. That is not what the PCA shows.
8. The figure legends are abbreviated. It is hard to understand how the data were generated and to assess the quality of the analysis.

Responses to reviewers' comments:

Reviewer #1 (Remarks to the Author):

Mitchell and colleagues present a new exciting paper that analyses the interactome of the transcription factor FOXQ1. A mass spectrometry-based screen identified components of the MLL histone methyltransferase complex as previously unknown interactors of FOXQ1. Among these proteins, direct interaction with RbBP5 is demonstrated and analysed in depth. The biochemical work analysing the FOXQ1 interaction with RbBP5 is extensive and in most parts convincing. Biologically, the complex between FOXQ1 and MLL, among other functions, is shown to regulate the epithelial-mesenchymal transition in breast epithelial and breast cancer cells. This biological part of the study is thorough and extensive, yet leaves several open questions in terms of the selectivity or specificity of the mechanism. The latter part, although well documented, leaves this reviewer with certain major questions being unanswered. The ambiguity I refer to is reflected even in the title of this paper, whereby the term “FOXQ1 hijacks the MLL complex to activate the EMT program” is not clear. What are the specific biological conditions that mediate this type of “hijacking”? Do the authors mean overexpression of FOXQ1 after transfection? Furthermore, the contribution of the FOXQ1-MLL complex to the regulation of the EMT program is not discussed at all in the otherwise very nice discussion of the paper that focuses on the biochemical/mechanistic findings, which are more robust.

The paper presents a plethora of experiments and makes use of many complementary controls, including rescue experiments that is very nice. My major concern is the selective mechanism by which FOXQ1 regulates the EMT. Multiple transcriptomic experiments are provided in this paper, and large numbers of genes are shown to be regulated by FOXQ1 and its interaction with RbBP5. Among these many genes, few belong to the so-called EMT program, and they are followed up, which is very nice. It would have been useful if the authors had provided some discussion on all the other FOXQ1 target genes, and their co-regulation by the RbBP5-MLL activity.

This brings to my major comment: if FOXQ1-RbBP5-MLL regulate a large number of genes in epithelial cells, what are the biological conditions that drive the function of FOXQ1 as an EMT inducer? Are there pathways or biological conditions during cancer development that turn on expression or activate the transcriptional activity of FOXQ1?

Response: We thank the reviewer for their comments about our biochemical work. We have addressed several questions concerning the biological significance by adding a more thorough background in our revision. We have incorporated new data from more TNBC cell lines and xenograft studies to highlight the relevance of our biochemical findings as features of cancer progression.

First, FOXQ1 has been demonstrated to be overexpressed in multiple human cancers and specifically in the TNBC subtype of breast cancer ^{1,2}. Biologically, FOXQ1 expression has been shown to be upregulated by the TGF- β and WNT signaling pathways, which are prominent pathways for EMT induction ²⁻⁴. These previous reports have been more thoroughly discussed in a new paragraph that has been added to the paper introduction (page 3, lines 74-81).

In this study, we provided further mechanistic insight into *how* FOXQ1 regulates the EMT program through the recruitment of the MLL complex as a transcriptional co-activator. The KMT2/MLL family of histone methyltransferases are crucial epigenetic regulators of gene expression, and the functional distinction amongst the members of this family has also been added to the introduction. While the family broadly regulates H3K4 methylation, KMT2/MLL gene chromatin distribution is thought to be dependent upon recruitment by sequence-specific TFs ⁵. Therefore, the biological *specificity* is dependent upon FOXQ1 and the contexts in which FOXQ1 is expressed.

Analysis of the breast TCGA data demonstrates that RbBP5 and KMT2A/MLL1 expression do not show differential expression between the PAM50 breast cancer subtypes. However, we do find that WDR5 and MLL4 have elevated expression in the basal-like subtype of breast cancer. While this is interesting, it is not explicitly implicated in our work based on our biochemical findings (**See attached Figure 1**).

Figure 1: Expression of FOXQ1 and KMT2/MLL complex components across the TCGA breast cancer subtypes. Gene expression data from the TCBA Breast Cancer Cohort (BRCA) (n=1247) was visualized using the UCSC Xeno browser. Heatmap displays log₂ transformed normalized RNA-seq count data for each sample. The PAM50 signature classification was annotated in the data set and determined according to the mRNA signatures established by the TCGA Atlas Network in *Nature* 2012.

Furthermore, we do not find that RbBP5 is bound to the promoters of EMT target genes in the absence of FOXQ1 through ChIP-qPCR in HMLE/LACZ epithelial control cells (Supplemental Figure 3v). Additionally, knockdown of FOXQ1 results in a loss of RbBP5 and H3K4me3 within the promoters of FOXQ1 EMT targets in our ectopic model and in TNBC cell lines (Figures 3g, h, and Supplementary Fig. 3w). Together, these data demonstrate our mechanism is relevant within the contexts in which FOXQ1 is *highly expressed*. Since FOXQ1 is highly expressed in the TNBC subtype of breast cancer and has been shown roles in EMT, this is the biological contexts that we focused on for this publication. We use an ectopic expression model based on HMLE cells to mimic the cancer cells to study the global binding of FOXQ1 and the MLL core complex. We also used multiple human TNBC cancer cells to confirm this effect. Moreover, the interaction was also validated in multiple human tumor samples, including lung cancer, Ovarian cancer and TNBC.

While we agree there is still much to learn about the biological contexts that involve the FOXQ1-MLL epigenetic axis of gene regulation, we believe we have demonstrated this mechanism occurs within the most well-known and significant contexts of FOXQ1 function thus far, and we present several possibilities for future work in this area in the discussion.

All the supporting literature and information have been added and clarified in the introduction and discussion of the current revision.

This is relevant as the study uses cell models in which immortalized mammary epithelial cells HMLE overexpress FOXQ1, and then this cell model is further manipulated to downregulate its RbBP5 or replace wild type FOXQ1 with mutants that fail to form the complex with RbBP5. An important set of experiments addressing my concern, focus on the breast cancer cell model MDA-MB-231, where endogenous FOXQ1 is silenced by shRNA and then wild type or mutant forms of FOXQ1 are co-transfected as rescuers. The paper presents these experiments briefly and emphasizes the effects of rescuing, overexpressed forms of FOXQ1 (wild type or mutant). Is the silencing of endogenous FOXQ1 in MDA-MB-231 cells turning these cells more epithelial, counteracting their mesenchymal identity? Data show that the migration of these cells is affected. Does such knockdown of endogenous FOXQ1 affect primarily the expression of EMT-like genes, or also the other genes identified in the HMLE-FOXQ1 overexpression transcriptomic screens?

Response: Many publications, including our previous work, have found that knockdown of FOXQ1 suppresses the EMT phenotype, including features such as gene expression, cell invasion, and distant metastasis *in vivo*.

In this study, we performed FOXQ1 KD in TNBC cell lines to confirm our mechanistic findings of the FOXQ1-MLL epigenetic axis of gene regulation and thereby regulation of the EMT phenotype. In several TNBC cell lines, including MDA-MB 231 and MDA-MB468 cells, we found knockdown of FOXQ1 affects the expression of both RbBP5-dependent and independent gene targets confirming their status as bonafide FOXQ1 target genes (Supp. figure 3g and 3j). We also identified FOXQ1 affects the expression of multiple signaling pathways, metabolic pathways, and inflammation in addition to the EMT program by ChIP and RNA sequencing experiments.

We performed the FOXQ1 knockdown rescue experiments with WT and mutant forms of FOXQ1 to specifically assess the significance of the FOXQ1-MLL epigenetic axis in regulating the EMT phenotype more globally. In the MDA-MB-231 rescue experiments, we did observe the morphology reverts to a mesenchymal morphology when WT FOXQ1 expression was restored, but not the mutant FOXQ1 (Supplementary Fig. 7e). Similarly, the migration and invasion capabilities were restored by WT FOXQ1 rescue but not the mutant FOXQ1 rescue (Fig. 7g and h).

In this manuscript, we focused on FOXQ1 regulation of the EMT program since there are still limited studies on the transcriptional mechanism of FOXQ1 on EMT. We will explore FOXQ1 function outside the context of EMT in future studies. This has been added to the discussion section (page 18, lines 445-455).

As a second major comment, I try to understand how the FOXQ1-MLL complex regulates the EMT program. Clearly FOXQ1 is directed to its target genes via its binding domain. And the paper proves convincingly that the transcriptional activity of FOXQ1 depends on RbBP5 and MLL activity. Since the EMT field is very rich in many studies that have analyzed gene expression and chromatin modification across a variety of breast cancer cell models but even in breast cancer patient material, is histone-3K4me3 a modification frequently found in the promoters of EMT genes in these other models and more importantly in patient samples? If yes, can the authors identify a correlation between the histone-3K4me3 profile of the EMT genes and FOXQ1, RbBP5 mRNA expression? Such evidence would strengthen the case that this paper attempts to make, regarding the functional role of FOXQ1-MLL and histone-3K4me3 in EMT and these findings would be nice to discuss also in the discussion of the paper.

Response: We thank the reviewer for their thoughtful questions. The chromatin landscape is highly altered during EMT as the epithelial genes undergo repression, and the mesenchymal genes become transcriptionally activated. For example, TGF- β stimulated EMT has been observed to result in chromatin decompaction due to a global loss in histone-3 lysine-9 dimethylation (H3K9me2) and an increase in histone-3 lysine-4 trimethylation (H3K4me3). Many histone-modifying complexes have been found to regulate the expression of, or directly interact with, EMT-TFs ⁶. Additionally, H3K4me3 is a prominent epigenetic mark located at the majority of promoters of actively transcribed genes. Therefore, it is challenging for us to assess the impact of the FOXQ1-MLL1 H3K4me3 signature through correlation. We have added one paragraph in the discussion section specifically addressing this issue (Page 17-18, lines 429-443).

We attempted to directly assess the involvement and contribution of the FOXQ1-MLL1 axis in regulating EMT in TNBC in our study using gene knockdown and mutation approaches since this would increase the specificity. Specifically, we found that knockdown of RbBP5 or MLL1 decreased the expression of FOXQ1 target EMT genes in TNBC cells (Supplementary Fig. 3g, 3j, 3m, 3p, and 3r). Since these knockdowns could also have off-target epigenetic consequences, we utilized the FOXQ1mutants that disrupt RbBP5-binding to demonstrate the contribution of FOXQ1-MLL1 to EMT-related features in HMLE and TNBC cells specifically (Fig. 7f-j and Supplementary Fig.7e-g).

Specific (and minor) comments:

1. Fig. 1 and Suppl. Fig. 1: specificity of the FOXQ1 antibody is shown after overexpression of Flag-FOXQ1 and Flag-FOXC2. The antibody should also be tested in cells (e.g. MDA-231) where endogenous FOXQ1 is silenced. Does it cross-react with any other protein under such conditions? This is important because Fig. 1d-f show the endogenous co-IP of FOXQ1 and RbBP5. In some of these cells, endogenous FOXQ1 is undetectable at 10% input cell lysate, yet a strong protein band is detected upon immunoprecipitation of 90% of the lysate with the RbBP5 antibody. Since molecular size markers are missing (please add those in every immunoblot), and since the intense, sharp band of the co-IP does not reflect any of the very faint and fuzzy bands of the 10% input, the sh/siRNA experiment is necessary to prove that the FOXQ1 protein bands that co-IPs is actually FOXQ1.

Response: Thanks for this valuable suggestion. We have added a western blot to show FOXQ1 in MDA-MB231/FOXQ1- KD cells (Supplementary Fig. 1f).

This is further important to cross-verify for the following reason: if we look at the data of Fig. 1c, where recombinant proteins are used we see that the direct interaction is very weak: compare the signal of 1% input to the signal after pull down of 99% protein. This signal of interaction is significantly enhanced upon endogenous co-IP (Fig. 1d-f). This means that either in cells, the proteins get post-translationally modified and their interaction is enhanced, or that cells provide an ancillary extra protein that enhances or stabilizes the interaction, or finally that the signal received in co-IP is not correct. For this reason, the requested specificity controls are important.

Response: We agree with the reviewer's comment. We have added protein markers on all blots.

2. Fig. 3d, f: if we look at gene by gene presence of the H3K4me3 mark, it is obvious that for some genes the RbBP5 shRNA (e.g. ZEB1, FOXC2) shows stronger effect and for some other the FOXQ1 shRNA (e.g. SIX2). This can be discussed in more detail, as it shows that the impact of silencing one or the other protein of the same complex is not functionally equivalent.

Response: FOXQ1 recruits RbBP5, but it does not exclude that other TFs may regulate these genes as well. For differences in the expression at individual gene loci, the upstream TFs may be different. Therefore, knockdown of RbBP5 or FOXQ1 could have other impacts on the different downstream gene expressions.

3. All experiments with shRNA knockdown need to show the degree of silencing of the respective protein (Fig. 3, FOXQ1, RbBP5).

Response: All the gene expression level before and after gene knockdown was determined by western blot and/or q-PCR in Supplementary Fig. 3.

4. The MDA-MB-231 mammosphere morphology is rather inconsistent between experiments: compare the NT cells in Suppl. Fig. 6h, 6j and Suppl. Fig. 7f: aggregates in 6h, 7f, acinar-like organoids in 6j. Why is it so? Are the culture conditions the same in all mammosphere experiments?

Response: We have replaced all these Figures with clean new images. The primary reason for the morphological differences could be due to the way to generate stable knockdown cells. We used pGIPZ shRNA constructs to generate the FOXQ1 knockdown cells, and we used pLKO shRNA constructs to generate the RbBP5 and other genes constructs.

5. Fig. 8b, c: it would be more appropriate if the three images (V5, vimentin, H&E, Fig. 8b) and (V5, H&E, Fig. 8c)

were from serial sections so that the same area of the tumor was stained in order to make sure that the same tumor cells are being analysed.

Response: The old Figure 8 was removed to Supplementary Figure 8. We performed a new *in vivo* study and made a new Figure 8 with the images taken from serial sections.

Reviewer #2 (Remarks to the Author):

Authors of the manuscript provide a mechanistic study which shows FOXQ1's direct interaction with RbBP5, a core subunit of KMT2/MLL H3K4 methyltransferase complexes, follow by the recruitment of the KMT2/MLL complex to EMT promoters, which in turn facilitates the deposition of H3K4 tri-methylation (H3K4me3) within the target promoters and activates EMT gene expression programs. They found that FOXQ1 binds to RbBP5 through its Forkhead domain and further identified key amino acid residues for this interaction. Authors then utilized genetic manipulations and pharmacologic approaches to elucidate the molecular mechanism where the disruption of FOXQ1-RbBP5 interaction (OICR-9429 treatment) or the failure of the KMT2/MLL complex recruitment (FOXQ1-I132S/A129S mutants) attenuates FOXQ1-dependnet gene expression, EMT, and cancer progression. Overall, the manuscript is well written, organized, and focused on an interesting topic with data presented support for authors' claims; however, few concerns remain and need to be addressed, thus, more studies and data would be required for publication in Nature Communications.

Response: We thank this reviewer's positive comment that "the manuscript is well written, organized, and data presented support for authors' claims." We have provided a point-by-point response to the concerns raised as follows.

General comments:

1. Given that KMT2/MLL family members share the core subunits RbBP5, ASH2L and WDR5, and they play distinct or redundant roles in different cancer contexts, a major problem is that the present study failed to identify the specific KMT2 member(s) that contribute(s) to FOXQ1 transactivation activity in breast cancer. I wonder if the authors identified any KMT2 members and/or unique subunits of the KMT2/MLL complexes in the TAP-MS analysis. Specific knockdown experiments for individual members of the KMT2/MLL family, such as H3K4 trimethyltransferases KMT2A, 2B, 2F, and 2G, would need to be performed in the HMLE/FOXQ1 and/or MDA-MB-231 cells and check for the effects of specific KMT2 loss on EMT promoter-associated H3K4me3. Although KMT2C and KMT2D function as H3K4 monomethyltransferases and mainly exert their function in enhancer regulation, they could also contribute to the regulation in promoter-associated H3K4me3 (Lee et al. PNAS 2008, Dhar et al. Genes & Development 2012, Kim et al. Cancer Research 2014). Thus, the effects of KMT2C/2D loss on EMT promoter-associated H3K4me3 would be worth to be checked as well.

Response: This is a great question. Actually, we did not detect any KMT2 members in the mass spectrometry. The reason we believe is that we may not have collected all the proteins based on the procedure for gel preparation and digestion requested by the proteomics core facility in Wayne State University. The KMT2 members are all very large proteins. Most of the KMT2 protein likely could not run into the gel with the 30mins gel electrophoresis we performed in the sample preparation.

To further clarify which KMT2 member is involved in FOXQ1-promoted gene regulation, we tested the interaction of FOXQ1 with a total of six KMT2 members, including SET1A, SET1B, KMT2A, KMT2B, KMT2C, and KMT2D. We only detected KMT2A in a complex with FOXQ1 (Fig. 1h). We further knocked down KMT2A in MDA-MB231 cells and observed significant inhibition of fibronectin expression, reversed mesenchymal morphology, decreased cell migration and invasion, and a reduced stem cell-like population (Figure 8a-d, and Supplementary 8e, f). We also implanted these cells in mice and found no significant change in tumor growth but a dramatic inhibition of tumor metastasis to the lung (Figure 8e-h).

2. As RbBP5 could also be shared by the KMT2C/2D complexes that catalyze H3K4 monomethylation on enhancer regions as well as facilitate enhancer activation, it is unclear whether the major role of FOXQ1 in regulating the activation of EMT transcription programs is exclusively through the promoter-associated H3K4me3. The ChIP-seq results of FOXQ1 and RbBP5 (Figure 2) also imply that the major occupancies of them both are not in the proximal promoter regions. Instead, the majority of FOXQ1 and RbBP5 seem to enrich in the distal intergenic and intron regions, where often contain enhancer DNA sequences. Thus, the authors should emphasize more clearly on the rationale to focus on the proximal promoter regions for the role of the FOXQ1-RbBP5-KMT2 axis in EMT gene regulation. Additional ChIP-Seq data for H3K4me1 may provide better distinctions between the FOXQ1 and RbBP5-targeted regions (whether promoters only or enhancers as well), and more clues for identifying the specific KMT2/MLL family member involved (mono- or trimethyltransferase).

Response: This is a good point we should clarify. Initially, We found FOXQ1 bound to both the proximal promoter regions and distal intergenic regions by ChIP-seq. However, when both the ChIP-seq and RNA-seq data are considered together, we found that 92% of FOXQ1-bound transcriptionally active promoters (622/791 genes) were also occupied by RbBP5 (<10 kb upstream TSS), and 73% were marked by H3K4me3 (Fig. 2e). We found this to be a functionally significant fraction that warranted characterization.

Based upon the helpful suggestions by this peer review, we are now able to address this question even more thoroughly since we found that KMT2A/MLL1 to be the likely enzymatic subunit that associates with FOXQ1 to regulate transcription of EMT in our study. As this reviewer likely knows, KMT2A/B is the MLL members that are thought to predominately regulate H3K4me3 within the promoter regions at specific gene targets. We do not exclude the likelihood that FOXQ1 has regulatory roles within enhancer regions and would like to explore these questions in more details in a future study.

We provided the rationale for focusing on the proximal promoter regions for the role of the FOXQ1-RbBP5-KMT2 axis in EMT gene regulation (page 8, line 193-197). We also added one paragraph about FOXQ1 enhancer activity as a future direction in the discussion section (page 18, lines 450-455).

3. The introduction is too short. There is a need to have a general introduction of the KMT2/MLL family of H3K4

methyltransferase complexes, especially on how they contribute to H3K4 methylation and target gene transcription.

Response: We provided more information about the research background about FOXQ1, including the upstream and downstream signaling pathways and biological function of FOXQ1 in cancer progression (Page 3, lines 74-81). We also added one paragraph in the introduction section providing general information about KMT2/MLL methyltransferase complexes and how they contribute to H3K4 methylation and target gene transcription. (page 3-4, Lines 86-101).

Specific comments:

1. Figure 2-- The display for the genomic distribution and colocalization of FOXQ1 and RbBP5 is not clear. In lines 118-122, the authors mentioned “our data indicate that FOXQ1 displays equivalent enrichment within proximal promoter region (<10 kb upstream) and distal intergenic regions (8% enrichment each for 13,513 peaks) (Figure 2A). RbBP5 binding peaks were enriched within proximal promoter regions in HMLE/FOXQ1 cells (10.8% of 25,866 peaks), and a fraction were enriched in distal intergenic regions (<2%)(Figure 2A)”. However, Figure 2A showed that FOXQ1 and RbBP5 occupied 58.5% and 51.9%, respectively, of distal intergenic regions. If it is true that FOXQ1 displays equivalent enrichment within proximal promoter region (<10 kb upstream) and distal intergenic regions (8% enrichment each for 13,513 peaks), with the enrichment of FOXQ1 in proximal promoter and intergenic regions only occupying 16% of the total FOXQ1 peaks, where would be the major distribution of FOXQ1 in the whole genome? Similarly, what is the percentage of FOXQ1–RbBP5–H3K4me3 co-targeted regions (1,397 peaks) that are within the promoter regions? A critical issue to address is how important the promoter-associated FOXQ1 function is in the scenario of EMT and breast cancer progression.

Response: We have agreed this paragraph was not clearly worded and we revised this portion of the manuscript (page 6, line 149-154). Basically, FOXQ1 binding to proximal promoter regions was calculated as the difference between FOXQ1 bound regions (8.1%) and the genomic distribution of promoter regions (3.5%). This is because the genomic distribution of these regions is what you would expect to find by random chance and are therefore serve as the expected control values. Therefore, FOXQ1 binding to proximal promoter regions is 4.6% above this background. Similarly, FOXQ1 bound within intergenic regions at 8.5% above background (58.5% minus 50%).

Based on our analysis, the portion of FOXQ1 binding within proximal promoter regions is lower than in the distal intergenic regions. However, additional pieces of evidence influenced our choice to focus on promoter regions for this study. The importance of the promoter-associated FOXQ1 function in the scenario of EMT and breast cancer progression is highlighted by several pieces of data: 1) more than 30% FOXQ1 upregulated genes displayed FOXQ1 binding within the associated promoter; 2) majority FOXQ1-bound transcriptionally active promoters (63%) were also occupied by RbBP5 and H3K4me3, 3) these FOXQ1-RbBP5 co-occupied promoters were significantly enriched for pathways critical to the EMT program and 4) in our new data we found that MLL1/KMT2 is involved

in the FOXQ1 complex. These results prompt us to investigate the promoter-associated FOXQ1 functions in EMT and breast cancer progression and the role of the MLL complex as a transcriptional coactivator. However, we agree that other mechanisms, including FOXQ1 function through the enhancer or different biological pathways in addition to EMT, could be equally important. These will serve as our future directions.

2. Figure 2C -- It is noticed that a significant number of FOXQ1 and RbBP5 co-targeted proximal promoters was covered by broad H3K4me3. Essentially, the major enrichment of FOXQ1 and RbBP5 within proximal promoter regions were broad and away from the target promoters (>1000 bp, ≤10,000 bp) (Figure 2A). Since broad H3K4me3 domains have been recently associated with maintenance of cell identity and cancer (Chen et al. Nature Genetics 2015, Cao et al. Scientific Reports 2017, Thibodeau et al. Scientific Reports 2017, Dhar et al. Molecular Cell 2018), it is unclear whether the FOXQ1-RbBP5-KMT2 axis is associated with the establishment of broad H3K4me3 domains in EMT-related genes. Additional discussion of this aspect in the manuscript is suggested.

Response: We agree the implications of broad H3K4me3 domains for the regulation of cell identity programs is interesting to be considered. When we analyzed H3K4me3 broad domains, we found only 50 differentially expressed genes with both FOXQ1 and H3K4 broad trimethylation in the promoter. However, we do find a similar overlap in the global chromatin distribution of FOXQ1 with broad H3K4me3 regions (**See attached Figure 2**). Due to the limitation of space and scope of the current study, we would like to focus on the promoter activity in the EMT context in this manuscript. We will follow this exciting avenue in the future.

Figure 2: Overlap of FOXQ1 binding peaks and H3K4me3 broad domains in HMLE/FOXQ1 cells. Venn diagram depicts the ChIP-seq peaks that overlap by at least 1 bp between FOXQ1 peaks and H3K4me3 broad domains. FOXQ1 peaks were called as described in the manuscript. H3K4me3 broad domains were called using MACS2 bdgbroadcall (Galaxy Version 2.1.1.20160309.0) with q = 0.1.

-Furthermore, it would be better to indicate the genomic locations of qPCR primers for the target promoters in the study, such as SIX2, TWIST1, ZEB1, FOXC2 and so on.

Response: Due to the scale of the image, it would be challenging to add the primers' locations and make the panel visible in this figure. We thus use the red arrows to indicate the ~150bp amplicon to the promoter regions on a scale of 2kb-10kb. The genomic regions of the amplicons analyzed by each of our ChIP-qPCR primer sets have been added to the supplementary tables.

3. Figure 5D and 5F -- Occupancies of RbBP5 and H3K4me3 in the FOXQ1/RbBP5 target promoter regions would need to be confirmed if the FOXQ1 mutants affect the recruitment of KMT2/MLL complex to these target promoters.

Response: We have already shown the FOXQ1 mutants do not interact with RbBP5 (Fig. 4g) and that these mutants do not activate the expression of FOXQ1-RbBP5 target genes (Fig.5d and Supplementary Fig.5e) despite being able to bind within the gene promoter (Fig. 5f). Therefore, we would expect the RbBP5-binding to be negative and correlative with the data that we already generated.

4. Figure 6E, 6J and 7E -- It is worth noting that RbBP5-knockdown or OICR-9429 treatment had modest effects on stem cell-like phenotype (CD44+/CD24-) when compared to the HMLE/FOXQ1 mutants. Please explain and discuss in the discussion section.

Response: These experiments were performed under different conditions and addressed different aspects of the EMT process. For HMLE/FOXQ1-WT and mutants, all the FOXQ1 variants are overexpressed in HMLE cells. The mutant FOXQ1 cannot induce EMT since it lacks the capability to interact with MLL core complex, and therefore these cells do not acquire the EMT-induced stem cell-like properties. For RbBP5-knockdown or OICR-9429 treatment, both target the MLL complex and try to reverse WT-FOXQ1 promoted stem cell-like properties. The effect could be impacted by many facets, such as knockdown efficacy and drug treatment dose. Therefore, in most cases, these approaches can only achieve a partial effect.

The above phenomenon observed *in vitro* was further validated *in vivo*. We demonstrated that ectopic expression of mutant FOXQ1 showed marked weak capability in tumor initiation and driving tumor progression compared to WT-FOXQ1 (Supplementary Fig. 8a-d). However, knockdown of MLL1 in MDA MB231 cells significantly inhibited tumor metastasis without significantly impacting tumor initiation and progression (Fig. 8e-h).

5. The representative pictures are missing for the morphological changes after knockdown of RbBP5 (Figure 6C), treatment of OICR-9429 (Figure 6H), introduction of FOXQ1 mutants (Figure 7A), and FOXQ1 manipulation in MDA-MB-231 cells (Figure 7F).

Response: We have shown the morphological changes in MDA MB231 cells in new Supplementary Fig. S6a, S6e, and S7e. We did not present new picture for S7a because the epithelial feature was already shown by increased IF staining of epithelial marker E-cadherin and b-catenin.

6. In lines 278-285, as the function of OICR-9429 is to disrupt the binding of RbBP5 to the KMT2/MLL complex, which should mimic the RbBP5 knockdown condition. It is not clear why the OICR-9429 treatment had effects on

cell proliferation (Figure S6E) but RbBP5-knockdown showed no significant effects on cell proliferation (Figure S6B).

Response: Sorry for the typo. Treatment of OICR had no significant effect on cell proliferation. This is corrected in the revised manuscript (See page 13, lines 333-335).

7. The basal level expression of FOXQ1 in HMLE/LacZ cells was missing. As this cell line serves as a control for many experiments, this could impact many observable results. Thus, it is important to show the basal level of FOXQ1 in the parental and HMLE/LacZ cells.

Response: The FOXQ1 expression level in HMLE/LacZ is very low, as shown by qRT-PCR in Supplementary Fig. 1a.

8. The immunoblotting results for the knockdown of FOXQ1 and RbBP5, and overexpression of FOXQ1 variants in HMLE/FOXQ1 or MDA-MB-231 cells were missing. Moreover, it would need to be concerned if the knockdown of FOXQ1 affects the levels of RbBP5 in these cells.

Response: We have shown knockdown of FOXQ1 and RbBP5 and overexpression of FOXQ1 variants in HMLE/FOXQ1 and MDA231 cells in Supplementary Fig. 3a, b, 3e, f, 3h, i, 3k, l, and 3n, o. The overexpression of FOXQ1 variants was shown in Supplementary Fig. 5a and 5e. We also determined that FOXQ1 knockdown did not affect the RBBP5 expression in Supplementary Fig. 3t, u.

Reviewer #3 (Remarks to the Author):

In this manuscript, Mitchel et al. investigated the requirement of FOXQ1 binding to RbBP5 subunit of KMT2/MLL histone methyltransferase to activate epithelial to mesenchymal transition (EMT) in breast cancer. The authors concluded that the disruption of RbBP5 binding to FOXQ1 causes decreased expression of EMT genes/proteins, inhibits the cell migration in vitro and prolongs survival of mice with xenograft tumors in vivo. While the in vitro mechanistic studies are interesting, most of the conclusions are limited to a one or two cell lines. The proposed molecular mechanism has very little relevance to triple-negative breast cancer in vivo. The in vivo orthotopic injection of the modified cell lines failed to demonstrate the convincing metastatic growth or changes in EMT that can support this mechanism. Clinical relevance of the conclusions to human breast cancer is also unclear. Major points are outlined below.

Response: We thank the reviewer for pointing out the weakness we need to address. We have provided more data based on more TNBC cell lines and performed another *in vivo* study using a more reliable metastatic model. Taken together with our previous results, the new evidence highlights the clinical relevance for the FOXQ1-MLL epigenetic axis in TNBC tumor progression and insights for potential therapeutic interventions. The detailed information is shown below in our point-by-point response.

Major concerns:

1. Although the main focus of the manuscript is the mechanisms of breast cancer metastasis, all the major conclusions are based on the data from a single mammary epithelial cell line with ectopic expression of FOXQ1 (HMLE/ FOXQ1). Mass spectrometry was done using embryonic kidney cells. The biological relevance of these two cell lines to the metastatic breast cancer is not clear.

Response: Thank you for pointing out an area we need to clarify. We have added a more thorough review of the literature surrounding the biological contexts and the significance of FOXQ1 as a critical regulator of EMT and metastasis in the introduction. We have included additional breast cancer cell lines and also performed additional *in vivo* xenograft experiments (Fig.8) using breast cancer cells to provide more evidence for the impact of this mechanism on metastasis.

The focus of the study was to understand the molecular mechanism that is utilized by FOXQ1 to promote gene expression, which we believe is important due to the demonstrated implications of EMT on cancer progression. The mass spectrometry experiments serve as the hypothesis-generating work in this line of inquiry. HEK293 cells were used because easy to transfect, and a lot of input protein is needed for the tandem affinity purification process. HEK cell line has been frequently used for interaction studies and protein-MS ⁷. We then validated our findings in HMLE/FOXQ1 cells as an EMT model and endogenously in breast cancer cell lines. We have further conducted additional experiments to confirm the endogenous interaction of FOXQ1-RBPP5 in various human tumor samples (Fig. 1g).

The HMLE cell line was selected because well-characterized model that is frequently used to study EMT^{8,9}. The HMLE cells are helpful to assess factors that are sufficient to induce EMT without the contribution of additional alterations that are found in breast cancer cells. Indeed, we have previously shown that ectopic expression of FOXQ1 is sufficient to induce EMT in HMLE cells and spontaneous distant metastasis in vivo Ras-transformed HMLE (HMLER) cells^{8,9}. These model experiments are routinely performed in the EMT field.

The HMLE/FOXQ1 cell is an EMT model, and we are exploring the transcriptional mechanism of FOXQ1 on EMT². Therefore, the results of the specific protein interaction of interest (FOXQ1 with the MLL complex) identified in HEK293T were confirmed in this FOXQ1-promoted EMT cell model. Moreover, EMT has been well recognized as a driving factor of cancer cell invasiveness, stemness, and metastatic potential. Therefore, we next validated this FOXQ1-MLL core complex interaction in multiple TNBC cell lines and human tumor samples.

2. All the conclusions are limited to analysis of cell lines, HMLE or a single breast cancer cell line MDA231, the latter of which was used to verify some of HMLE data. Authors should use transgenic mouse models of breast cancer to verify the results. Several transgenic mouse models of breast cancer are available and should be used in these studies.

Response: We have expanded our cell line models. We verified FOXQ1 and RbBP5 interaction in at least three TNBC cell lines. We also confirmed this interaction in multiple patient tumor samples, including lung cancer, Ovarian cancer, and TNBC tumors. The transgenic mouse model is a very useful model to study gene function under physiological conditions. However, to study how the FOXQ1-RBBP5 interaction drives tumor progression using transgenic mice, we need to generate mice with wild-type and mutant FOXQ1, and then cross these mice with other TNBC mouse models. This will be a huge project and will be our future direction.

3. The clinical relevance of the data to human breast cancer is unclear. Human tissue samples were not used in these studies. Is the proposed mechanism of FOXQ1-RbBP5 interaction important for all types of breast cancer or only for TNBC and why? For breast cancer patients, is there correlation between metastatic potential and FOXQ1/MLL expression?

Response: Thanks for bringing up several points we needed to clarify. We have revised the introduction to make these aspects of the study much clearer.

FOXQ1 expression is correlated with poor prognosis in breast cancer patients¹⁰, as well as in other solid tumors¹. In breast cancer, the expression of FOXQ1 in the primary tumor is correlated with the incidence of metastasis. More specifically, FOXQ1 overexpression is observed almost exclusively in the TNBC subtype of breast cancer.

On the other hand, MLL1 is a histone methyltransferase, which acts as a transcriptional co-activator, found to play an essential role in regulating gene expression during early development and hematopoiesis⁵. The novelty of our study is that overexpression of FOXQ1 could lead to recruitment (“Hijack”) of MLL core complex – a typical epigenetic mechanism for transcriptional gene activation.

In this revision, we have addressed the concern of the clinical relevance of our study to breast cancer by providing several new lines of new evidence. 1) We expanded TNBC cell models to confirm FOXQ1 and RbBP5 interaction (Fig. 1d); 2) We examined FOXQ1-RbBP5 interaction in multiple human tumor samples, including breast cancers (Fig. 1g); 3) We analyzed the prognostic value of core gene signatures that regulated by FOXQ1-RbBP5 interaction and identified several genes showed worse overall prognosis in breast cancer (Supplementary Fig. 2f-h); and 4) We performed MLL1 knockdown experiment in MDA-MB231 cells and demonstrated that knockdown of MLL1 in MDA231 cells significantly decreased cell migration and invasion without affecting cell proliferation in vitro (Fig. 8a-d). Consistent with this observation, knockdown of MLL1 in MDA-MB-231cells significantly decreased tumor metastasis but did not change tumor growth (Fig. 8e-h).

4. All the main experiments are missing the HMLE/ LacZ control. What are the levels of FOXQ1 in the normal breast epithelial cells in culture and in vivo?

Response: In adult tissue, the human FOXQ1 gene is expressed predominantly in the stomach, trachea, bladder, and salivary gland. Additionally, overexpression of human FOXQ1 was shown in many cancer cell lines including non-small cell lung cancer, colorectal cancer, hepatocellular carcinoma, gastric cancer, and thyroid carcinoma ¹¹. For breast cancer, FOXQ1 is highly expressed in TNBC ^{2,10}. FOXQ1 expression in normal mammary epithelial cells is also very low. We have shown the expression of FOXQ1 in HMLE/LacZ cells along with a panel of breast cancer cell lines by qRT-PCR in supplemental Fig. 1a.

How different is the ectopically expressed levels of FOXQ1 in cell lines compared to endogenous levels of FOXQ1 in human breast cancer tissues?

Response: This is essential information that we ought to show the readers. We have assessed this question previously and found that HMLE/FOXQ1 cells express FOXQ1 to a level that is observed in other models of TNBC cell lines. We have added this qRT-PCR data to supplemental Fig. 1a.

5. During assessment of migration/invasion of HMLE/FOXQ1 cells + shRbBP5, the authors did not include an important control of HMLE cells + shRbBP5.

Response: We performed the migration and invasion assay based on HMLE/LacZ cells with or without RbBP5 knockdown and observed no significant difference (See attached Figure 3). This result was mentioned in the manuscript without the figure shown (page 13, lines 328-329).

Figure 3. Cell migration and invasion analysis on HMLE/LacZ cells with or without RbBP5 knockdown. NS: no statistical significance.

6. Although, the authors intended to verify their data in vivo, their metastasis model is not very convincing. Based

on histological images provided by the authors, the V5-antibody staining is non-specific and labels lung airway epithelial cells the most, making it difficult to quantify the metastases. To evaluate whether the distal metastases are developed in this model and to quantify them, the authors should label tumor cells with the reporter and track the development of the metastases with this reporter.

Response: We agree with the reviewer's comments and removed the metastatic results of HMLER model from the main figures in this revision. Instead, we performed another *in vivo* study using MDA-MB231 cell model with MLL1 knockdown, which forms much more overt lung metastases. The results showed that knockdown of MLL1 in MDA-MB231 cells did not change cell proliferation but significantly changed cell migration and invasion (new Fig. 8a-c), as well as cell morphology *in vitro* (new Supplementary Fig. 8e, f). In line with these observations, knockdown of MLL1 in MDA-MB 231 cells did not change the primary tumor growth but significantly decreased tumor metastasis to the lung (new Fig. 8e-h).

7. Some of the conclusions are not supported by the data. For example, using FOXQ1 ChIP-seq analysis, the authors state that FOXQ1 binds equally to proximal promoter regions (8%) and distal intergenic regions (Fig.2A). However the data in Fig.2A shows the majority of FOXQ1 binding peaks being in intron-distal intergenic regions (~90%). How the analysis was done is not explained in the manuscript.

Response: Thanks for the reviewer's comments. We agree that the previous statement is not correct. We revised the paragraph as follow: In HMLE cells, we observed FOXQ1 bound to proximal promoter regions (<10 kb upstream; 8.1% of 13,513 total peaks, 4.6% above the genome average of 3.5%) and distal intergenic regions (58.5% of the total, 8.5% above the genome average of 50%) (Fig. 2a). RbBP5 binding peaks also fell within proximal promoter regions (10.8% of 25,866 total peaks, 7.3% above the genome average) and in distal intergenic regions (51.9% of the total, <2% above the genome average) (Fig. 2a) (Page 6, lines 149-154). This analysis approach was added to the ChIP-data analysis section in the method in this revision (page 31, lines 874-884).

-The other example is Fig.S5B, which uses the principle component analysis (PCA) for RNAseq. The authors state that overexpression of mutant FOXQ1 in HMLE/LacZ cells make the cells more similar to HMLE/LacZ than to HMLE/FOXQ1 cells. That is not what the PCA shows.

Response: In our PCA plot, we have PC1 labeled on the x-axis, which accounts for 75% of the variance in the data and is the more heavily weighted principal component. When we consider the distribution of the samples first on the x-axis, we find that the HMLE/FOXQ1 mutant samples are closer to the HMLE/LACZ samples than they are to the HMLE/FOXQ1 WT samples. PC2 is plotted on the Y-axis and approximates 21% of the sample variance. In this case, we do see that HMLE/FOXQ1 MUT and HMLE/FOXQ1 WT samples are more closely distributed along the y-axis. However, since PC1 is weighted more heavily than PC2 we interpret this result as stated in the manuscript.

8. The figure legends are abbreviated. It is hard to understand how the data were generated and to assess the quality of the analysis.

Response: The figure legends for all figures have been modified to explain how the data were generated. See attached manuscript highlighted for review only.

References

- 1 Cui, X. *et al.* Prognostic value of FOXQ1 in patients with malignant solid tumors: a meta-analysis. *OncoTargets and therapy* **10**, 1777-1781, doi:10.2147/ott.s130905 (2017).
- 2 Zhang, H. *et al.* Forkhead transcription factor foxq1 promotes epithelial-mesenchymal transition and breast cancer metastasis. *Cancer research* **71**, 1292-1301, doi:10.1158/0008-5472.can-10-2825 (2011).
- 3 Christensen, J., Bentz, S., Sengstag, T., Shastri, V. P. & Anderle, P. FOXQ1, a novel target of the Wnt pathway and a new marker for activation of Wnt signaling in solid tumors. *PloS one* **8**, e60051, doi:10.1371/journal.pone.0060051 (2013).
- 4 Peng, X. *et al.* FOXQ1 mediates the crosstalk between TGF-beta and Wnt signaling pathways in the progression of colorectal cancer. *Cancer biology & therapy* **16**, 1099-1109, doi:10.1080/15384047.2015.1047568 (2015).
- 5 Rao, R. C. & Dou, Y. Hijacked in cancer: the KMT2 (MLL) family of methyltransferases. *Nat Rev Cancer* **15**, 334-346, doi:10.1038/nrc3929 (2015).
- 6 McDonald, O. G., Wu, H., Timp, W., Doi, A. & Feinberg, A. P. Genome-scale epigenetic reprogramming during epithelial-to-mesenchymal transition. *Nature structural & molecular biology* **18**, 867-874, doi:10.1038/nsmb.2084 (2011).
- 7 Thomas, P. & Smart, T. G. HEK293 cell line: a vehicle for the expression of recombinant proteins. *J Pharmacol Toxicol Methods* **51**, 187-200, doi:10.1016/j.vascn.2004.08.014 (2005).
- 8 Mani, S. A. *et al.* The epithelial-mesenchymal transition generates cells with properties of stem cells. *Cell* **133**, 704-715, doi:10.1016/j.cell.2008.03.027 (2008).
- 9 Mani, S. A. *et al.* Mesenchyme Forkhead 1 (FOXQ2) plays a key role in metastasis and is associated with aggressive basal-like breast cancers. *Proceedings of the National Academy of Sciences of the United States of America* **104**, 10069-10074, doi:10.1073/pnas.0703900104 (2007).
- 10 Qiao, Y. *et al.* FOXQ1 regulates epithelial-mesenchymal transition in human cancers. *Cancer research* **71**, 3076-3086, doi:10.1158/0008-5472.CAN-10-2787 (2011).
- 11 Li, Y. *et al.* Forkhead box Q1: A key player in the pathogenesis of tumors (Review). *Int J Oncol* **49**, 51-58, doi:10.3892/ijo.2016.3517 (2016).

REVIEWER COMMENTS

Reviewer #2 (Remarks to the Author):

The authors answered all my questions to my complete satisfaction. I would only like to suggest some smaller changes and additions.

In Figure 3, the effects of MLL1 knockdown on the H3K4me3 ChIP to those FOXQ1-targeted genes were missing. This is an important result to support that MLL1 is the key histone methyltransferase for the implementation of H3K4me3 on FOXQ1-targeted loci.

In Figure 7e, an important control for shFOXQ1 in MDA-MB-231 cells was missing. Because MDA-MB-231 cells express high levels of FOXQ1, it would be important to show whether knockdown of FOXQ1 in this metastatic breast cancer cell line could switch the expression of mesenchymal markers to epithelial markers while only wild-type FOXQ1 (but not A129S and I132S mutants) could rescue the loss of FOXQ1 effect. The effect of A129S on Fibronectin seems to be much modest than that of I132S. The reviewer is questioning why to detect Occludin but not E-cadherin or Claudin in this experiment. Moreover, the Fibronectin immunoblot did not align to other immunoblots well.

There are typos in line 98: within punitive enhancer regions, and line 328: shRbBP5 knockdown. There might be more typos and/or grammar errors in the current manuscript.

Reviewer #3 (Remarks to the Author):

Some of my points were addressed. The manuscript is based on strong in vitro mechanistic studies which comprehensively demonstrated the direct connection between FOXQ1 and MLL. However, verification of the in vitro data using the in vivo models is still weak.

I still think that labeling of tumor cells in your xenograft model with fluorescent or luciferase reporters will make the in vivo results more convincing and the lung metastases more visible, and easier to quantify.

Of note, the new Fig. 8F does not look right. There are no differences in the immunostaining between panels. Everything is brown, even H&E staining. Maybe there was an error during uploading?

Reviewer #4 (Remarks to the Author):

In this manuscript, Mitchel AV and colleagues set to examine the molecular mechanisms of FOXQ1 mediated transcriptional induction of the EMT program. FOXQ1 has been previously shown to activate EMT in mammary epithelial cell lines by the senior author of this manuscript (ref. #10 - Zhang H et al, Cancer Research, 2011) and others (ref. #9 - Qiao Y et al, Cancer Research, 2011). Interestingly, the authors used the same cellular model, HMLE cells with FOXQ1 overexpression as in their paper decade ago, as well as HMLER cells used by the other research group in 2011. HMLE cells have proven to be a good exploratory model; however, its relevance to breast cancer is limited. As a short note, the authors refer to HMLE as "human mammary luminal epithelial cells" on page 4, line 109. It should be corrected that these cells are of basal, not luminal origin. In addition, in functional assays, i.e. knockdown of endogenous FOXQ1, the authors utilized MDA231 breast cancer cells, which are commonly used as a metastasis model. One caveat is that these cells have undergone full EMT and lost typical epithelial characteristics. In this regard, it could be a good model of metastasis but it does not represent vast majority of basal/triple negative breast cancers. Some of the functional assays in this study extend the data obtained in same MDA231 cell line reported in the Qiao et al paper (effects of FOXQ1 on EMT, invasion, 3D growth, cell proliferation).

Despite these limitations with cellular models, the manuscript presents enormous amount of work and novel findings, which will be of great interest to the field. The most significant part of this study is the identification and comprehensive characterization of FOXQ1 co-activator complex, consisting of direct interactor RbBP5 and other subunits of the KMT2A/MLL1 chromatin-modifying complex. Identification of RbBP5 and ASH2L as FOXQ1 interacting proteins by TAP mass spectrometry is consistent with their presence in FOXQ1 complex reported in high throughput proteomic study of 56 human TFs, including FOXQ1 (Li X et al, Molecular Systems Biology, 2015). Mapping of the interaction domains of FOXQ1 and RbBP5 and generation and use of the interaction-deficient point mutants in functional assays provides new mechanistic insight for FOXQ1 transactivation function. Cross-analysis of FOXQ1 OX RNA-Seq data coupled with ChIP-Seq for FOXQ1, RbBP5 and H3K4me3 nicely identifies 791 direct FOXQ1 target genes (by proximal promoter binding data), most of which are co-occupied by RbBP5 and H3K4me3. Multiple knockdown experiments confirm functional significance and downstream nature of RbBP5 and MLL1/H3K4me3 vis-a-vis FOXQ1. Among these targets are several EMT master regulator TFs like ZEB1, TWIST1, SIX2 and FOXC2, activation of which is likely a mechanism for FOXQ1 induced EMT.

The biochemical, genomic/transcriptomic and cellular functional parts of this work are quite solid. At least two shRNAs are used per gene, multiple gene targets and point mutants are analyzed, and some data are confirmed in other cell lines. The manuscript is well written and follows defined logical narrative. The authors addressed most of the concerns in the revised version. Thus, few minor concerns outlined below remain to be addressed before the manuscript could be accepted for publication in Nature Communications.

One of my conceptual concerns is about the authors' proposed model of FOXQ1 induced EMT. Four EMT TFs (FOXC2, TWIST1, ZEB1, SIX2) were identified by ChIP-Seq and RNA-Seq as direct FOXQ1-RbBP5 downstream targets. However, alongside these four, several other EMT TFs were similarly upregulated in HMLE-FOXQ1 cells: ZEB1 – 106-fold, ZEB2 – 96-fold, TWIST1 – 63-fold, TWIST2 – 23-fold, SIX1 – 20-fold, SIX2 – 1342-fold, FOXC2 – 122-fold, FOXF1 – 2876-fold (Supplementary dataset 2). Upregulation of multiple EMT TFs and specifically ZEBs and TWISTs is consistent with previous observation of cooperative activation of the EMT/stem-like program in HMLE and MDA231 cells (Addison JB et al, Molecular Cancer Research, 2021). Could the authors discuss these implications? Could it also be possible that FOXQ1 binds to enhancers rather than promoters of these other EMT TFs? It has been shown that ZEB2 is essential for FOXQ1-mediated HCC metastasis, with FOXQ1 transactivating ZEB2 gene by directly binding to the ZEB2 promoter in HCC cells (Xia L et al, Hepatology, 2014). In this regard, the discussion section focuses on the biochemical/mechanistic findings of the FOXQ1-RbBP5-MLL1 axis, which are novel and very interesting, but completely skips the implications of FOXQ1 on the regulation of the EMT program. Is the main mechanism for FOXQ1 mediated induction of EMT by transactivation of EMT-TF master regulators, ZEBs and TWISTs?

Since the authors have successfully generated two sublines with knockdowns of each FOXQ1, RbBP5 (Suppl. Fig.3), and MLL1 (Fig.8) in MDA231 cells, their most relevant cellular model, it would be interesting to compare their individual effects on EMT. These cellular lysates could be examined on the same blot and probed with antibodies to multiple epithelial and mesenchymal markers that the authors already have, and also for ZEB1, E-cadherin, and others. In addition, the same figure should feature Western blot for FOXQ1, RbBP5 and MLL1, confirming their knockdown. Although MDA231/shMLL1-2 & -3 cells are used in experiments described in Fig.8, no protein or qPCR data is presented on the efficiency of MLL1 knockdown anywhere in the manuscript.

I also agree that the word "hijacks" in the manuscript title might be misleading. "Recruits" instead could be a better alternative.

Other comments:

1. There are several inconsistencies with Western blot figures.

a. Vimentin, which is a protein of ~57 kDa (>51 kDa by the datasheet of the vimentin antibody used by the authors), is identified at ~45 kDa on all full blots (see Suppl. Fig.9 full blots for Fig.6C, 6H, 7A, 7F, Suppl. Fig.1B). However, the same bands are labeled at ~57 kDa on cut-outs in Fig.6C, 6H and 7F. On Fig. 7A it is shown at ~45 kDa. How the authors could explain these discrepancies?

b. Claudin-1, a protein of ~25 kDa, is labeled as ~60 kDa in Fig. 6C and 6H, while their corresponding full blots show it as ~50 kDa and 25 kDa, respectively.

c. Occludin is labeled at ~60 kDa in Fig.7F and at ~90 kDa on the corresponding full blot.

d. Fibronectin IB has 8 lanes in Fig.7F and its corresponding full blot, while this figure analyzes only 7 lanes for all other proteins.

2. Both FOXQ1 shRNA target sites are inside FOXQ1 ORF. Why wouldn't they target exogenous FOXQ1 constructs that are reintroduced into MDA231/shFOXQ1 cells?

3. Supplemental Table 1 "Sequences of gene-specific shRNAs" lists two shRNAs for each FOXQ1, RbBP5 and MLL1, some encoded in pLKO vector, others in pGIPZ. However, no control shRNAs are listed for either vector. Nontarget (NT) control cells are used alongside knockdown sublines in all experiments, but there is no description what these cells are. Are these parental cells, cells infected with an empty vector or with a vector with control shRNA? If so, which shRNA?

4. Supplementary datasets 2 & 3 reporting differential gene expression between HMLE/FOXQ1 wild type vs HMLE/LacZ and HMLE/FOXQ1 I132S mutant vs HMLE/LacZ, respectively, should to be reformatted and standardized.

a. A column with gene IDs/identifiers should to be added.

b. Genes with zero values should to be filtered out/removed from the analysis. For example, E-Cadherin/CDH1 and many other epithelial markers seem to be below detection level, showing 0 value.

c. The output of the analysis for both datasets should to be standardized/unified. For example, Gene ID, Gene Name, log₂FC, FC, replicates TMM (weighted trimmed mean of M) values, p-values, q-values columns.

Typos:

Figure 2D legend has a typo "Axes show log₁₀ fold-change versus log₂ p-value." It should be the opposite: Axes show log₂ fold-change versus log₁₀ p-value.

Line 334 in the main text should refer to Supplementary Fig.6e.

Line 371 in the main text, did the author mean a delay in tumor initiation (not progression)?

Responses to reviewers' comments:

Reviewer #2 (Remarks to the Author):

Q1. In Figure 3, the effects of MLL1 knockdown on the H3K4me3 ChIP to those FOXQ1-targeted genes were missing. This is an important result to support that MLL1 is the key histone methyltransferase for the implementation of H3K4me3 on FOXQ1-targeted loci.

Response: We performed this experiment as requested. The results showed that MLL1 knockdown decreased H3K4me3 binding to the panel of FOXQ1-targeted genes (new Supplementary Figure 3t) on page 9, lines 222-224. This result is consistent with Supplementary Figure 3r, which showed the expression of the panel of FOXQ1-targeted genes was inhibited due to the MLL1 knockdown in MDA-MB231 cells.

Q2. In Figure 7e, an important control for shFOXQ1 in MDA-MB-231 cells was missing. Because MDA-MB-231 cells express high levels of FOXQ1, it would be important to show whether knockdown of FOXQ1 in this metastatic breast cancer cell line could switch the expression of mesenchymal markers to epithelial markers while only wild-type FOXQ1 (but not A129S and I132S mutants) could rescue the loss of FOXQ1 effect. The effect of A129S on Fibronectin seems to be much modest than that of I132S. The reviewer is questioning why to detect Occludin but not E-cadherin or Claudin in this experiment. Moreover, the Fibronectin immunoblot did not align to other immunoblots well.

Response: Thank you for bringing our attention to a point that need clarification. Previous work by our lab, and others, have shown that knockdown of FOXQ1 in TNBC cells, and specifically in MDA-MB-231 and 4T1 cells, disrupts EMT and leads to a decrease in mesenchymal marker expression and increase in epithelial markers (1, 2). However, MDA-MB-231 cells do not express E-cadherin due to methylation of the DNA promoter (3). Therefore, we rely on other markers such as claudin and occludin as additional surrogate markers for epithelial character, along with the function biological changes we observe such as invasion, migration, mammosphere formation, etc.

The specific differences observed when FOXQ1-A129S or FOXQ1-I132S are ectopically expressed could be attributed to the impact that each of these point mutations has on disrupting the protein-protein interaction between FOXQ1 and RbBP5. While both of the point mutations have a dramatic decrease in the FOXQ1-RbBP5 protein interaction, experiments we performed using purified protein show that each mutant still sustains some level of binding to RbBP5, with A129S potentially maintaining a slightly higher affinity than I132S (Supplemental figure 4D). These potential biochemical differences could become more pronounced when introduced into a biological system. More detailed biochemical and structural analysis is needed to further interrogate molecular dynamics of the FOXQ1-RbBP5 protein interaction to determine if this is the case.

We found out that the Fibronectin immunoblot did not align well to other immunoblots because we included one extra blank lane on the right of the immunoblot. We have deleted the blank lane and align it well with other immunoblots (Figure 7F).

Q3. There are typos in line 98: within punitive enhancer regions, and line 328: shRbBP5 knockdown. There might be more typos and/or grammar errors in the current manuscript.

Response: We corrected these typos in this revision. For page 4, line 98: within putative enhancer regions. For page 13, line 327: RbBP5 knockdown. We also corrected several other typos identified by reviewer 4. Moreover, we carefully proofreading the manuscript several times ensuing there are no other typos.

References

1. Qiao Y, Jiang X, Lee ST, Karuturi RK, Hooi SC, Yu Q. FOXQ1 regulates epithelial-mesenchymal transition in human cancers. *Cancer Res.* 2011;71(8):3076-86.
2. Zhang H, Meng F, Liu G, Zhang B, Zhu J, Wu F, Ethier SP, Miller F, Wu G. Forkhead transcription factor foxq1 promotes epithelial-mesenchymal transition and breast cancer metastasis. *Cancer Res.* 2011;71(4):1292-301.
3. Liu Y, Lee W, Wang C, Chao T, Chen Y, Chen J. Regulatory mechanisms controlling human E-cadherin gene expression. *Oncogene* 2005; 24:8277-8290.

Reviewer #3 (Remarks to the Author):

Q1. Some of my points were addressed. The manuscript is based on strong in vitro mechanistic studies which comprehensively demonstrated the direct connection between FOXQ1 and MLL. However, verification of the in vitro data using the in vivo models is still weak.

Response: We appreciate the reviewer point out this problem. As a future direction, we will generate a genetically modified TNBC cell model with mutant FOXQ1 deficiency in binding with RbBP5 to study the function in vivo. We also have a plan to develop a split-luciferase system to screen small molecular inhibitor to target FOXQ1/RbBP5 interaction.

Q2. I still think that labeling of tumor cells in your xenograft model with fluorescent or luciferase reporters will make the in vivo results more convincing and the lung metastases more visible, and easier to quantify.

Response: We agree that it would be easier to quantify the lung metastatic lesion with fluorescent or luciferase reporter labelled tumor cells. However, since the knockdown tumor cell barely showed any metastatic lesion in the lung, it would be not necessary for us to repeat this experiment. We appreciate the editor agree with this argument.

Q3. Of note, the new Fig. 8F does not look right. There are no differences in the immunostaining between panels. Everything is brown, even H&E staining. Maybe there was an error during uploading?

Response: We have repeat the IHC experiment and new pictures were uploaded in revised Fig.8F.

Reviewer #4 (Remarks to the Author):

We sincerely appreciate the reviewer's comments on our revised manuscript that "The biochemical, genomic/transcriptomic and cellular functional parts of this work are quite solid"; "The manuscript is well written and follows defined logical narrative", and "The authors addressed most of the concerns in the revised version". Some comments and few minor concerns of this reviewer were discussed and addressed as follows.

Q1. In this manuscript, Mitchel AV and colleagues set to examine the molecular mechanisms of FOXQ1 mediated transcriptional induction of the EMT program. FOXQ1 has been previously shown to activate EMT in mammary epithelial cell lines by the senior author of this manuscript (ref. #10 - Zhang H et al, Cancer Research, 2011) and others (ref. #9 - Qiao Y et al, Cancer Research, 2011). Interestingly, the authors used the same cellular model, HMLE cells with FOXQ1 overexpression as in their paper decade ago, as well as HMLER cells used by the other research group in 2011. HMLE cells have proven to be a good exploratory model; however, its relevance to breast cancer is limited. As a short note, the authors refer to HMLE as "human mammary luminal epithelial cells" on page 4, line 109. It should be corrected that these cells are of basal, not luminal origin.

Response: We agree with reviewer's comment that HMLE is only the EMT model and its relevance to breast cancer is limited. We thus validated the results using multiple TNBC cell models including MDA231, SUM159 and MDA468 cells.

We corrected the definition of HMLE as human mammary epithelial cells on page 4, line 109.

Q2. In addition, in functional assays, i.e. knockdown of endogenous FOXQ1, the authors utilized MDA231 breast cancer cells, which are commonly used as a metastasis model. One caveat is that these cells have undergone full EMT and lost typical epithelial characteristics. In this regard, it could be a good model of metastasis but it does not represent vast majority of basal/triple negative breast cancers. Some of the functional assays in this study extend the data obtained in same MDA231 cell line reported in the Qiao et al paper (effects of FOXQ1 on EMT, invasion, 3D growth, cell proliferation).

Response: We fully agree with the comments of MDA231 does not represent majority of basal/Triple negative breast cancer. In this revision, we have also validated some data in SUM159 and MDA468 cells. We hope in the future we can validate and confirm the results in more TNBC cells and human tumor samples.

Q3. One of my conceptual concerns is about the authors' proposed model of FOXQ1 induced EMT. Four EMT TFs (FOXC2, TWIST1, ZEB1, SIX2) were identified by ChIP-Seq and RNA-Seq as direct FOXQ1-RbBP5 downstream targets. However, alongside these four, several other EMT TFs were similarly upregulated in HMLE-FOXQ1 cells: ZEB1 – 106-fold, ZEB2 – 96-fold, TWIST1 – 63-fold, TWIST2 – 23-fold, SIX1 – 20-fold, SIX2 – 1342-fold, FOXC2 – 122-fold, FOXF1 – 2876-fold (Supplementary dataset 2). Upregulation of multiple EMT TFs and specifically ZEBs and TWISTs is consistent with previous observation of cooperative activation of the EMT/stem-like program in HMLE and MDA231 cells (Addison JB et al, Molecular Cancer Research, 2021). Could the authors discuss these implications? Could it also be possible that FOXQ1 binds to enhancers rather than promoters of these other EMT TFs?

Response: We appreciate the reviewer suggestion and added this viewpoint into our discussion section in line 448 to 456 on page 18.

Q4. It has been shown that ZEB2 is essential for FOXQ1-mediated HCC metastasis, with FOXQ1 transactivating ZEB2 gene by directly binding to the ZEB2 promoter in HCC cells (Xia L et al, Hepatology, 2014). In this regard, the discussion section focuses on the biochemical/mechanistic findings of the FOXQ1-RbBP5-MLL1 axis, which are novel and very interesting, but completely skips the implications of FOXQ1 on the regulation of the EMT program. Is the main mechanism for FOXQ1 mediated induction of EMT by transactivation of EMT-TF master regulators, ZEBs and TWISTs?

Response: We agree with reviewer's comments. Our current provided a global view of FOXQ1 transcriptional profile and identified many known and unknown, direct and indirect downstream targets including multiple EMT master regulators such as ZEBs and TWISTs. This result is consistent with another new manuscript from our laboratory showing FOXQ1 and SNAI1 are independent EMT regulators with common and distinct transcriptional profiles (data not shown). Given the importance of ZEBs and TWISTs in regulation of EMT and cancer progression, we believe FOXQ1 could mediate induction of EMT by transactivation of EMT-TF master regulators, ZEBs and TWISTs. A systematic loss of function approach could be used to test this hypothesis. This could serve as a future direction for my laboratory. In current manuscript, we are focusing on identification and validating the interaction of FOXQ1 and MLL complex, and exploring the effect of this complex in promoting transcription of EMT and tumor progression.

Q5. Since the authors have successfully generated two sublimes with knockdowns of each FOXQ1, RbBP5 (Suppl. Fig.3), and MLL1 (Fig.8) in MDA231 cells, their most relevant cellular model, it would be interesting to compare their individual effects on EMT. These cellular lysates could be examined on the same blot and probed with antibodies to multiple epithelial and mesenchymal markers that the authors already have, and also for ZEB1, E-cadherin, and others. In addition, the same figure should feature Western blot for FOXQ1, RbBP5 and MLL1, confirming their knockdown.

Response: We performed some western blotting analysis according to reviewer's suggestion. The EMT markers showed various degree of alterations in different knockdown models (Figure 1). However, due to the uneven level of knockdown for different molecules, it is hard to compare these WB results across different cell models. Therefore, we believe that, at this moment, the only conclusion we can summarize is that knockdown of FOXQ1, RbBP5 and MLL1 all impact the FOXQ1 transcription on EMT program.

Figure1. Western blots for different EMT related genes in MDA231 cells with knockdown of FOXQ1, RbBP5 or MLL1. B-actin was a loading control. Note: We have technical difficulty in western blot analysis to present MLL1 protein expression because of its large size. The MLL1 knockdown expression level was shown using qPCR in Supplementary Figure 3q.

Q6. Although MDA231/shMLL1-2 & -3 cells are used in experiments described in Fig.8, no protein or qPCR data is presented on the efficiency of MLL1 knockdown anywhere in the manuscript.

Response: The MLL1 knockdown expression level was shown using qPCR in Supplementary Figure 3q.

Q7. I also agree that the word “hijacks” in the manuscript title might be misleading. “Recruits” instead could be a better alternative.

Response: We agree to change the title to “FOXQ1 Recruits the MLL Complex to Activate Transcription of the EMT Program and Promote Breast Cancer Metastatic Progression”.

Other comments:

1. There are several inconsistencies with Western blot figures.

a. Vimentin, which is a protein of ~57 kDa (>51 kDa by the datasheet of the vimentin antibody used by the authors), is identified at ~45 kDa on all full blots (see Suppl. Fig.9 full blots for Fig.6C, 6H, 7A, 7F, Suppl. Fig.1B). However,

the same bands are labeled at ~57 kDa on cut-outs in Fig.6C, 6H and 7F. On Fig. 7A it is shown at ~45 kDa. How the authors could explain these discrepancies?

Response: These are mis-labels. We have corrected all Vimentin labels to around 57 KDa on all full blots for Fig.6C, 6H, 7A, 7F and Supplementary Fig.1B. We also changed it on Fig.7A.

b. Claudin-1, a protein of ~25 kDa, is labeled as ~60 kDa in Fig. 6C and 6H, while their corresponding full blots show it as ~50 kDa and 25 kDa, respectively.

Response: We have changed all labels to around 25 kDa in Fig.6C and 6H.

c. Occludin is labeled at ~60 kDa in Fig.7F and at ~90 kDa on the corresponding full blot.

Response: According to the information from the company, the right size for the Occludin ab we used is around 90 KDa. We added the correct labels to both Fig.7F and original full blot.

d. Fibronectin IB has 8 lanes in Fig.7F and its corresponding full blot, while this figure analyzes only 7 lanes for all other proteins.

Response: We appreciate the reviewer point out this problem. The protein loading in this immunoblot should be 7 lanes. We included one extra blank lane. We have changed it by deleting the blank lane on the right. See new Figure 7F and the original full blot in supplementary Figure.

2. Both FOXQ1 shRNA target sites are inside FOXQ1 ORF. Why wouldn't they target exogenous FOXQ1 constructs that are reintroduced into MDA231/shFOXQ1 cells?

Response: Yes, they do target both endogenous and exogenous FOXQ1. We selected clones for these experiments that displayed sufficient ectopic expression of FOXQ1 to overwhelm the shRNA.

3. Supplemental Table 1 "Sequences of gene-specific shRNAs" lists two shRNAs for each FOXQ1, RbBP5 and MLL1, some encoded in pLKO vector, others in pGIPZ. However, no control shRNAs are listed for either vector. Nontarget (NT) control cells are used alongside knockdown sublines in all experiments, but there is no description what these cells are. Are these parental cells, cells infected with an empty vector or with a vector with control shRNA? If so, which shRNA?

Response: We apologize for missing this critical information, these have been included in the revised supplemental table 1.

4. Supplementary datasets 2 & 3 reporting differential gene expression between HMLE/FOXQ1 wild type vs HMLE/LacZ and HMLE/FOXQ1 I132S mutant vs HMLE/LacZ, respectively, should to be reformatted and standardized.

a. A column with gene IDs/identifiers should to be added.

b. Genes with zero values should to be filtered out/removed from the analysis. For example, E-Cadherin/CDH1 and many other epithelial markers seem to be below detection level, showing 0 value.

c. The output of the analysis for both datasets should to be standardized/unified. For example, Gene ID, Gene Name, log2FC, FC, replicates TMM (weighted trimmed mean of M) values, p-values, q-values columns.

Response: We have made sure to include the standardized data files as suggested by the reviewer. See revised Supplementary datasets 2 & 3.

Typos:

Figure 2D legend has a typo “Axes show log10 fold-change versus log2 p-value.” It should be the opposite: Axes show log2 fold-change versus log10 p-value.

Response: We corrected the mistake accordingly. See Page 42, line 1264.

Line 334 in the main text should refer to Supplementary Fig.6e.

Response: We have changed it to Supplementary Fig.6e. See page 13, Line 333.

Line 371 in the main text, did the author mean a delay in tumor initiation (not progression)?

Response: The HMLER/FOXQ1-WT group had a median time to the endpoint of 8 days (23-15), while FOXQ1-A129S and FOXQ1-I132S groups had a median time to the endpoint of 10 (52-42) days and 18 (75-57) days (Supplementary Fig. 8a-c). Therefore, we believe FOXQ1-mutant groups had a delay in tumor progression (page 15, line 370).

We also indicated the effect on tumor initiation on page 14-15, lines 362-367.

REVIEWER COMMENTS

Reviewer #2 (Remarks to the Author):

The authors have already addressed my questions and concerns. I think the manuscript is in a good shape to be published in Nature Communications.

Reviewer #4 (Remarks to the Author):

The authors answered most of my questions and addressed most of my comments.

Thank you for adding the clarification to Suppl. Table 1: "pLKO.1 empty vector was used to match RbBP5 and MLL1 shRNA. pGipZ empty vector was used to match FOXQ1 shRNA." This reviewer strongly recommends the authors to use non-targeting shRNAs as controls rather than empty vectors. Ectopic expression of shRNAs may overwhelm cellular miRNA/shRNA processing machinery in experimental cell lines compared to empty vector control cell lines potentially leading to non-specific effects unrelated to gene specificity of the shRNAs.

Minor comments:

In the revised version, the authors included the sentence:

222 In line with this, we showed that MLL1

223 knockdown decreased H3K4me3 binding to the panel of FOXQ1-targeted genes

(Supplementary

224 Fig.3t).

I am not sure that the word "binding" is appropriate here. Perhaps, better wording would be "MLL1 knockdown decreased H3K4me3 levels on the panel of FOXQ1-targeted gene promoters".

Supplementary files with RNA-Seq data are labeled Supplementary Dataset 2 & 3 in the main text. The excel files have labels Extended Data. Please reconcile. It would be useful to add description tab to each excel spreadsheet describing the dataset.

Typo on line 450:

450 This can be partially explained by possible FOXQ1 binding to the enhancer, rath than promoter

451 regions of these gene targets.

There are few other typos throughout the manuscript. Please proofread.

Responses to reviewers' comments:

Reviewer #2 (Remarks to the Author):

The authors have already addressed my questions and concerns. I think the manuscript is in a good shape to be published in Nature Communications.

Response: Thanks for the comments that this manuscript is in a good shape and can be published in Nature Communications.

Reviewer #4 (Remarks to the Author):

The authors answered most of my questions and addressed most of my comments.

Thank you for adding the clarification to Suppl. Table 1: “pLKO.1 empty vector was used to match RbBP5 and MLL1 shRNA. pGipZ empty vector was used to match FOXQ1 shRNA.” This reviewer strongly recommends the authors to use non-targeting shRNAs as controls rather than empty vectors. Ectopic expression of shRNAs may overwhelm cellular miRNA/shRNA processing machinery in experimental cell lines compared to empty vector control cell lines potentially leading to non-specific effects unrelated to gene specificity of the shRNAs.

Response: I checked with the students working on this project and found that my previous answer is not accurate. We have several plasmids serve as control to different shRNAs. The empty vectors were designated as EV and vectors expressing scramble shRNA were designated as NT. We have corrected the mistakes in the Supplementary table 1. “pLKO.1 vector expressing scramble shRNA (designated as NT) was used to match RbBP5 and MLL1 shRNA. pGipZ vector expressing scramble shRNA (designated as NT) was used to match FOXQ1 shRNA”.

Minor comments:

In the revised version, the authors included the sentence:

222 In line with this, we showed that MLL1

223 knockdown decreased H3K4me3 binding to the panel of FOXQ1-targeted genes (Supplementary
224 Fig.3t).

I am not sure that the word “binding” is appropriate here. Perhaps, better wording would be “MLL1 knockdown decreased H3K4me3 levels on the panel of FOXQ1-targeted gene promoters”.

Response: We have revised the sentence and changed it to “MLL1 knockdown decreased H3K4me3 levels on the FOXQ1-targeted genes”. (Line 222-223, Page 9)

Supplementary files with RNA-Seq data are labeled Supplementary Dataset 2 & 3 in the main text. The excel files have labels Extended Data. Please reconcile. It would be useful to add description tab to each excel spreadsheet describing the dataset.

Response: We have changed the labels in Source data excel file from “Extended Data” to “Supplementary dataset”. Now, the labels of the datasets were now aligned between the main text and Source data.

Typo on line 450:

450 This can be partially explained by possible FOXQ1 binding to the enhancer, rath than promoter
451 regions of these gene targets.

There are few other typos throughout the manuscript. Please proofread.

Response: We have revised the sentence and changed it to “This observation can be partially explained by FOXQ1 binding to the possible enhancer rather than promoter regions of these gene targets”. (Line 449-451, Page 18).

We also performed proofreading of the whole manuscript and identified multiple other typos or grammatical mistakes. The corrections were all highlighted in the “manuscript file with highlights for review only”.